# UNDERSTANDING WHY GENERALIZED REWEIGHTING DOES NOT IMPROVE OVER ERM

**Runtian Zhai, Chen Dan, Zico Kolter, Pradeep Ravikumar**
Carnegie Mellon University
Pittsburgh, PA, USA 15213
`{rzhai,cdan,zkolter,pradeepr}@cs.cmu.edu`

## ABSTRACT

Empirical risk minimization (ERM) is known to be non-robust in practice to distributional shift where the training and the test distributions are different. A suite of approaches, such as importance weighting, and variants of distributionally robust optimization (DRO), have been proposed to solve this problem. But a line of recent work has empirically shown that these approaches do not significantly improve over ERM in real applications with distribution shift. The goal of this work is to obtain a comprehensive theoretical understanding of this intriguing phenomenon. We first posit the class of Generalized Reweighting (GRW) algorithms, as a broad category of approaches that iteratively update model parameters based on iterative reweighting of the training samples. We show that when overparameterized models are trained under GRW, the resulting models are close to that obtained by ERM. We also show that adding small regularization which does not greatly affect the empirical training accuracy does not help. Together, our results show that a broad category of what we term GRW approaches are not able to achieve distributionally robust generalization. Our work thus has the following sobering takeaway: to make progress towards distributionally robust generalization, we either have to develop non-GRW approaches, or perhaps devise novel classification/regression loss functions that are adapted to GRW approaches.

## 1 INTRODUCTION

It has now been well established that empirical risk minimization (ERM) can empirically achieve high test performance on a variety of tasks, particularly with modern overparameterized models where the number of parameters is much larger than the number of training samples. This strong performance of ERM however has been shown to degrade under *distributional shift*, where the training and test distributions are different (Hovy & Søgaard, 2015; Blodgett et al., 2016; Tatman, 2017). There are two broad categories of distribution shift: *domain generalization*, defined as the scenario where the test distribution contains samples from new domains that did not appear during training; and *subpopulation shift*, defined as the scenario where the training set contains several subgroups and the testing distribution weighs these subgroups differently, like in fair machine learning.

People have proposed various approaches to learn models robust to distributional shift. The most classical one is importance weighting (IW) (Shimodaira, 2000; Fang et al., 2020), which reweights training samples; for subpopulation shift these weights are typically set so that each subpopulation has the same overall weight in the training objective. The approach most widely used today is Distributional Robust Optimization (DRO) (Duchi & Namkoong, 2018; Hashimoto et al., 2018), which assumes that the test distribution belongs to a certain *uncertainty set* of distributions that are close to the training distribution, and train on the worst distribution in that set. Many variants of DRO have been proposed and are used in practice (Sagawa et al., 2020a; Zhai et al., 2021a;b).

While these approaches have been developed for the express purpose of improving ERM for distribution shift, a line of recent work has empirically shown the negative result that when used to train overparameterized models, these methods do not improve over ERM. For IW, Byrd & Lipton (2019) observed that its effect under stochastic gradient descent (SGD) diminishes over training epochs, and finally does not improve over ERM. For variants of DRO, Sagawa et al. (2020a) found that these methods overfit very easily, i.e. their test performances will drop to the same low level as ERM after sufficiently many epochs if no regularization is applied. Gulrajani & Lopez-Paz (2021); Koh et al.

(2021) compared these methods with ERM on a number of real-world applications, and found that in most cases none of these methods improves over ERM.

This line of empirical results has also been bolstered by some recent theoretical results. Sagawa et al. (2020b) constructed a synthetic dataset where a linear model trained with IW is provably not robust to subpopulation shift. Xu et al. (2021) further proved that under gradient descent (GD) with a sufficiently small learning rate, a linear classifier trained with either IW or ERM converges to the same max-margin classifier, and thus upon convergence, are no different. These previous theoretical results are limited to linear models and specific approaches such as IW where sample weights are fixed during training. They are not applicable to more complex models, and more general approaches where the sample weights could iteratively change, including most DRO variants.

Towards placing the empirical results on a stronger theoretical footing, we define the class of *generalized reweighting* (GRW), which dynamically assigns weights to the training samples, and iteratively minimizes the weighted average of the sample losses. By allowing the weights to vary with iterations, we cover not just static importance weighting, but also DRO approaches outlined earlier; though of course, the GRW class is much broader than just these instances.

**Main contributions.** We prove that GRW and ERM have (almost) equivalent implicit biases, in the sense that **the points they converge to are very close to each other**, under a much more general setting than those used in previous work. Thus, GRW cannot improve over ERM because it does not yield a significantly different model. We are the first to extend this line of theoretical results (i) to wide neural networks, (ii) to reweighting methods with dynamic weights, (iii) to regression tasks, and (iv) to methods with $L_2$ regularization. We note that these extensions are non-trivial technically as they require the result that wide neural networks can be approximated by their linearized counterparts to hold *uniformly throughout the iterative process* of GRW algorithms. Moreover, we fix the proof in a previous paper (Lee et al., 2019) (see Appendix E) which is also a great contribution.

Overall, the important takeaway is that *distributionally robust generalization* (DRG) cannot be directly achieved by the broad class of GRW algorithms (which includes popular approaches such as importance weighting and most DRO variants). Progress towards this important goal thus requires either going beyond GRW algorithms, or devising novel loss functions that are adapted to GRW approaches. In Section 6 we will discuss some promising future directions and the case with non-overparameterized models and early stopping. Finally, we want to emphasize that while the models we use in our results (linear models and wide neural networks) are different from practical models, they are general models most widely used in existing theory papers, and our results based on these models provide explanations to the baffling observations made in previous empirical work, as well as valuable insights into how to improve distributionally robust generalization.

## 2 PRELIMINARIES

Let the input space be $\mathcal{X} \subseteq \mathbb{R}^d$ and the output space be $\mathcal{Y} \subseteq \mathbb{R}$.[1] We assume that $\mathcal{X}$ is a subset of the unit $L_2$ ball of $\mathbb{R}^d$, so that any $\boldsymbol{x} \in \mathcal{X}$ satisfies $\|\boldsymbol{x}\|_2 \leq 1$. We have a training set $\{\boldsymbol{z}_i = (\boldsymbol{x}_i, y_i)\}_{i=1}^n$ *i.i.d.* sampled from an underlying distribution $P$ over $\mathcal{X} \times \mathcal{Y}$. Denote $\boldsymbol{X} = (\boldsymbol{x}_1, \cdots, \boldsymbol{x}_n) \in \mathbb{R}^{d \times n}$, and $\boldsymbol{Y} = (y_1, \cdots, y_n) \in \mathbb{R}^n$. For any function $g : \mathcal{X} \mapsto \mathbb{R}^m$, we overload notation and use $g(\boldsymbol{X}) = (g(\boldsymbol{x}_1), \cdots, g(\boldsymbol{x}_n)) \in \mathbb{R}^{m \times n}$ (except when $m = 1$, $g(\boldsymbol{X})$ is defined as a column vector). Let the loss function be $\ell : \mathcal{Y} \times \mathcal{Y} \to [0, 1]$. ERM trains a model by minimizing its *expected risk* $\mathcal{R}(f; P) = \mathbb{E}_{\boldsymbol{z} \sim P}[\ell(f(\boldsymbol{x}), y)]$ via minimizing the *empirical risk* $\hat{\mathcal{R}}(f) = \frac{1}{n} \sum_{i=1}^n \ell(f(\boldsymbol{x}_i), y_i)$.

In distributional shift, the model is evaluated not on the training distribution $P$, but a different test distribution $P_{\text{test}}$, so that we care about the expected risk $\mathcal{R}(f; P_{\text{test}})$. A large family of methods designed for such distributional shift is *distributionally robust optimization* (DRO), which minimizes the expected risk over the worst-case distribution $Q \ll P$[2] in a ball w.r.t. divergence $D$ around the training distribution $P$. Specifically, DRO minimizes the *expected DRO risk* defined as:

$$\mathcal{R}_{D,\rho}(f; P) = \sup_{Q \ll P} \{\mathbb{E}_Q[\ell(f(\boldsymbol{x}), y)] : D(Q \parallel P) \leq \rho\} \tag{1}$$

for $\rho > 0$. Examples include CVaR, $\chi^2$-DRO (Hashimoto et al., 2018), and DORO (Zhai et al., 2021a), among others.

---

[1]Our results can be easily extended to the multi-class scenario (see Appendix B).

[2]For distributions $P$ and $Q$, $Q$ is *absolute continuous* to $P$, or $Q \ll P$, means that for any event $A$, $P(A) = 0$ implies $Q(A) = 0$.

A common category of distribution shift is known as subpopulation shift. Let the data domain contain $K$ *groups* $\mathcal{D}_1, \cdots, \mathcal{D}_K$. The training distribution $P$ is the distribution over all groups, and the test distribution $P_{\text{test}}$ is the distribution over one of the groups. Let $P_k(\boldsymbol{z}) = P(\boldsymbol{z} \mid \boldsymbol{z} \in \mathcal{D}_k)$ be the conditional distribution over group $k$, then $P_{\text{test}}$ can be any one of $P_1, \cdots, P_k$. The goal is to train a model $f$ that performs well over every group. There are two common ways to achieve this goal: one is minimizing the *balanced empirical risk* which is an unweighted average of the empirical risk over each group, and the other is minimizing the *worst-group risk* defined as

$$\mathcal{R}_{\max}(f; P) = \max_{k=1,\cdots,K} \mathcal{R}(f; P_k) = \max_{k=1,\cdots,K} \mathbb{E}_{\boldsymbol{z} \sim P}[\ell(f(\boldsymbol{x}), y) | z \in \mathcal{D}_k] \tag{2}$$

## 3   GENERALIZED REWEIGHTING (GRW)

Various methods have been proposed towards learning models that are robust to distributional shift. In contrast to analyzing each of these individually, we instead consider a large class of what we call Generalized Reweighting (GRW) algorithms that includes the ones mentioned earlier, but potentially many others more. Loosely, GRW algorithms iteratively assign each sample a weight during training (that could vary with the iteration) and iteratively minimize the weighted average risk. Specifically, at iteration $t$, GRW assigns a weight $q_i^{(t)}$ to sample $\boldsymbol{z}_i$, and minimizes the weighted empirical risk:

$$\hat{\mathcal{R}}_{\boldsymbol{q}^{(t)}}(f) = \sum_{i=1}^{n} q_i^{(t)} \ell(f(\boldsymbol{x}_i), y_i) \tag{3}$$

where $\boldsymbol{q}^{(t)} = (q_1^{(t)}, \cdots, q_n^{(t)})$ and $q_1^{(t)} + \cdots + q_n^{(t)} = 1$.

*Static GRW* assigns to each $\boldsymbol{z}_i = (\boldsymbol{x}_i, y_i)$ a fixed weight $q_i$ that does not change during training, i.e. $q_i^{(t)} \equiv q_i$. A classical method is *importance weighting* (IW) (Shimodaira, 2000), where if $\boldsymbol{z}_i \in \mathcal{D}_k$ and the size of $\mathcal{D}_k$ is $n_k$, then $q_i = (Kn_k)^{-1}$. Under IW, (3) becomes the balanced empirical risk in which each group has the same weight. Note that ERM is also a special case of static GRW.

On the other hand, in *dynamic GRW*, $\boldsymbol{q}^{(t)}$ changes with $t$. For instance, any approach that iteratively upweights samples with high losses in order to help the model learn "hard" samples, such as DRO, is an instance of GRW. When estimating the population DRO risk $\mathcal{R}_{D,\rho}(f; P)$ in Eqn. (1), if $P$ is set to the empirical distribution over the training samples, then $Q \ll P$ implies that $Q$ is also a distribution over the training samples. Thus, DRO methods belong to the broad class of GRW algorithms. There are two common ways to implement DRO. One uses Danskin's theorem and chooses $Q$ as the maximizer of $\mathbb{E}_Q[\ell(f(\boldsymbol{x}), y)]$ in each epoch. The other one formulates DRO as a bi-level optimization problem, where the lower level updates the model to minimize the expected risk over $Q$, and the upper level updates $Q$ to maximize it. Both can be seen as instances of GRW. As one popular instance of the latter, *Group DRO* was proposed by Sagawa et al. (2020a) to minimize (2). Denote the empirical risk over group $k$ by $\hat{\mathcal{R}}_k(f)$, and the model at time $t$ by $f^{(t)}$. Group DRO iteratively sets $q_i^{(t)} = g_k^{(t)}/n_k$ for all $\boldsymbol{z}_i \in \mathcal{D}_k$ where $g_k^{(t)}$ is the group weight that is updated as

$$g_k^{(t)} \propto g_k^{(t-1)} \exp\left(\nu \hat{\mathcal{R}}_k(f^{(t-1)})\right) \ (\forall k = 1, \cdots, K) \tag{4}$$

for some $\nu > 0$, and then normalized so that $q_1^{(t)} + \cdots + q_n^{(t)} = 1$. Sagawa et al. (2020a) then showed (in their Proposition 2) that for convex settings, the Group DRO risk of iterates converges to the global minimum with the rate $O(t^{-1/2})$ if $\nu$ is sufficiently small.

## 4   THEORETICAL RESULTS FOR REGRESSION

In this section, we will study GRW for regression tasks that use the squared loss

$$\ell(\hat{y}, y) = \frac{1}{2}(\hat{y} - y)^2. \tag{5}$$

We will prove that for both linear models and sufficiently wide fully-connected neural networks, the implicit bias of GRW is equivalent to ERM, which means that starting from the same initial point, GRW and ERM will converge to the same point when trained for an infinitely long time. Thus, GRW cannot improve over ERM as it produces the exact same model as ERM. We will further show that while regularization can affect this implicit bias, it must be large enough to *significantly lower the training performance*, or the final model will still be similar to the unregularized ERM model.

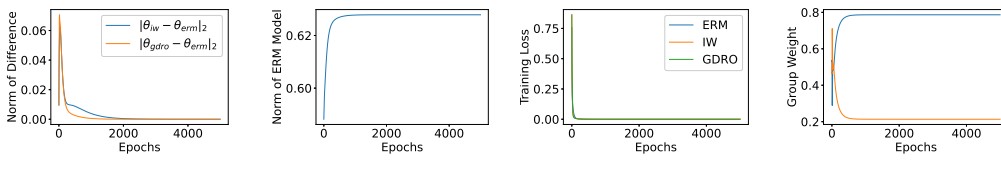

| (a) Weight Difference | (b) Norm of ERM Model | (c) Training Loss | (d) Group Weights |

Figure 1: Experimental results of ERM, importance weighting (IW) and Group DRO (GDRO) with the squared loss on six MNIST images with a linear model. All norms are $L_2$ norms.

## 4.1 LINEAR MODELS

We first demonstrate our result on simple linear models to provide our readers with a key intuition which we will later apply to neural networks. This key intuition draws from results of Gunasekar et al. (2018). Let the linear model be denoted by $f(\boldsymbol{x}) = \langle \theta, \boldsymbol{x} \rangle$, where $\theta \in \mathbb{R}^d$. We consider the overparameterized setting where $d > n$. The weight update rule of GRW under GD is the following:

$$\theta^{(t+1)} = \theta^{(t)} - \eta \sum_{i=1}^{n} q_i^{(t)} \nabla_\theta \ell(f^{(t)}(\boldsymbol{x}_i), y_i) \tag{6}$$

where $\eta > 0$ is the learning rate. For a linear model with the squared loss, the update rule is

$$\theta^{(t+1)} = \theta^{(t)} - \eta \sum_{i=1}^{n} q_i^{(t)} \boldsymbol{x}_i(f^{(t)}(\boldsymbol{x}_i) - y_i) \tag{7}$$

For this training scheme, we can prove that if the training error converges to zero, then the model converges to an interpolator $\theta^*$ (*s.t.* $\forall i, \langle \theta^*, \boldsymbol{x}_i \rangle = y_i$) independent of $q_i^{(t)}$ (proofs in Appendix D):

**Theorem 1.** *If $\boldsymbol{x}_1, \cdots, \boldsymbol{x}_n$ are linearly independent, then under the squared loss, for any GRW such that the empirical training risk $\hat{\mathcal{R}}(f^{(t)}) \to 0$ as $t \to \infty$, it holds that $\theta^{(t)}$ converges to an interpolator $\theta^*$ that only depends on $\theta^{(0)}$ and $\boldsymbol{x}_1, \cdots, \boldsymbol{x}_n$, but does not depend on $q_i^{(t)}$.*

The proof is based on the following key intuition regarding the update rule (7): $\theta^{(t+1)} - \theta^{(t)}$ is a linear combination of $\boldsymbol{x}_1, \cdots, \boldsymbol{x}_n$ for all $t$, so $\theta^{(t)} - \theta^{(0)}$ always lies in the linear subspace span$\{\boldsymbol{x}_1, \cdots, \boldsymbol{x}_n\}$, which is an $n$-dimensional linear subspace if $\boldsymbol{x}_1, \cdots, \boldsymbol{x}_n$ are linearly independent. By Cramer's rule, there is exactly one $\tilde{\theta}$ in this subspace such that we get interpolation of all the data $\langle \tilde{\theta} + \theta^{(0)}, \boldsymbol{x}_i \rangle = y_i$ for all $i \in \{1, \ldots, n\}$. In other words, the parameter $\theta^* = \tilde{\theta} + \theta^{(0)}$ in this subspace that interpolates all the data is unique. Thus the proof would follow if we were to show that $\theta^{(t)} - \theta^{(0)}$, which lies in the subspace, also converges to interpolating the data. Moreover, this proof works for any first-order optimization method such that the training risk converges to 0.

We have essentially proved the following sobering result: *any GRW algorithm that achieves zero training error exactly produces the ERM model, so it does not improve over ERM*. While the various distributional shift methods discussed in the introduction have been shown to satisfy the precondition of convergence to zero training error with overparameterized models and linearly independent inputs (Sagawa et al., 2020a), we provide the following theorem that shows this for the broad class of GRW methods. Specifically, we show this result for any GRW method that satisfies the following assumption with a sufficiently small learning rate:

**Assumption 1.** There are constants $q_1, \cdots, q_n$ *s.t.* $\forall i, q_i^{(t)} \to q_i$ as $t \to \infty$. And $\min_i q_i = q^* > 0$.

**Theorem 2.** *If $\boldsymbol{x}_1, \cdots, \boldsymbol{x}_n$ are linearly independent, then there exists $\eta_0 > 0$ such that for any GRW satisfying Assumption 1 with the squared loss, and any $\eta \leq \eta_0$, the empirical training risk $\hat{\mathcal{R}}(f^{(t)}) \to 0$ as $t \to \infty$.*

Finally, we use a simple experiment to demonstrate the correctness of this result. The experiment is conducted on a training set of six MNIST images, five of which are digit 0 and one is digit 1. We use a 784-dimensional linear model and run ERM, importance weighting and group DRO. The results are presented in Figure 1, and they show that the training loss of each method converges to 0, and the gap between the model weights of importance weighting, Group DRO and ERM converges to 0, meaning that all three model weights converge to the same point, whose $L_2$ norm is about 0.63. Figure 1d also shows that the group weights in Group DRO empirically satisfy Assumption 1.

## 4.2 WIDE NEURAL NETWORKS (WIDE NNS)

Now we study *sufficiently wide fully-connected neural networks*. We extend the analysis in Lee et al. (2019) in the neural tangent kernel (NTK) regime (Jacot et al., 2018). In particular we study

the following network:

$$\boldsymbol{h}^{l+1} = \frac{W^l}{\sqrt{d_l}}\boldsymbol{x}^l + \beta\boldsymbol{b}^l \qquad \text{and} \qquad \boldsymbol{x}^{l+1} = \sigma(\boldsymbol{h}^{l+1}) \qquad (l = 0, \cdots, L) \qquad (8)$$

where $\sigma$ is a non-linear activation function, $W^l \in \mathbb{R}^{d_{l+1} \times d_l}$ and $W^L \in \mathbb{R}^{1 \times d_L}$. Here $d_0 = d$. The parameter vector $\theta$ consists of $W^0, \cdots, W^L$ and $b^0, \cdots, b^L$ ($\theta$ is the concatenation of all flattened weights and biases). The final output is $f(\boldsymbol{x}) = \boldsymbol{h}^{L+1}$. And let the neural network be initialized as

$$\begin{cases} W_{i,j}^{l(0)} \sim \mathcal{N}(0,1) \\ \boldsymbol{b}_j^{l(0)} \sim \mathcal{N}(0,1) \end{cases} (l = 0, \cdots, L-1) \qquad \text{and} \qquad \begin{cases} W_{i,j}^{L(0)} = 0 \\ \boldsymbol{b}_j^{L(0)} \sim \mathcal{N}(0,1) \end{cases} \qquad (9)$$

We also need the following assumption on the wide neural network:

**Assumption 2.** $\sigma$ is differentiable everywhere. Both $\sigma$ and its first-order derivative $\dot{\sigma}$ are Lipschitz.[3]

**Difference from Jacot et al. (2018).** Our initialization (9) differs from the original one in Jacot et al. (2018) in the last (output) layer, where we use the zero initialization $W_{i,j}^{L(0)} = 0$ instead of the Gaussian initialization $W_{i,j}^{L(0)} \sim \mathcal{N}(0,1)$. This modification permits us to accurately approximate the NN with its linearized counterpart (11), as we notice that the proofs in Lee et al. (2019) (particularly the proofs of their Theorem 2.1 and their Lemma 1 in Appendix G) are flawed. In Appendix E we will explain what went wrong in their proofs and how we fix it with this modification.

Denote the neural network at time $t$ by $f^{(t)}(\boldsymbol{x}) = f(\boldsymbol{x}; \theta^{(t)})$ which is parameterized by $\theta^{(t)} \in \mathbb{R}^p$ where $p$ is the number of parameters. We use the shorthand $\nabla_\theta f^{(0)}(\boldsymbol{x}) := \nabla_\theta f(\boldsymbol{x}; \theta)\big|_{\theta=\theta_0}$. The *neural tangent kernel* (NTK) of this model is $\Theta^{(0)}(\boldsymbol{x}, \boldsymbol{x}') = \nabla_\theta f^{(0)}(\boldsymbol{x})^\top \nabla_\theta f^{(0)}(\boldsymbol{x}')$, and the *Gram matrix* is $\Theta^{(0)} = \Theta^{(0)}(\boldsymbol{X}, \boldsymbol{X}) \in \mathbb{R}^{n \times n}$. For this wide NN, we still have the following NTK theorem:

**Lemma 3.** *If $\sigma$ is Lipschitz and $d_l \to \infty$ for $l = 1, \cdots, L$ sequentially, then $\Theta^{(0)}(\boldsymbol{x}, \boldsymbol{x}')$ converges in probability to a non-degenerate[4] deterministic limiting kernel $\Theta(\boldsymbol{x}, \boldsymbol{x}')$.*

The *kernel Gram matrix* $\Theta = \Theta(\boldsymbol{X}, \boldsymbol{X}) \in \mathbb{R}^{n \times n}$ is a positive semi-definite symmetric matrix. Denote its largest and smallest eigenvalues by $\lambda^{\max}$ and $\lambda^{\min}$. Note that $\Theta$ is non-degenerate, so we can assume that $\lambda^{\min} > 0$ (which is almost surely true when $d_L \gg n$). Then we have:

**Theorem 4.** *Let $f^{(t)}$ be a wide fully-connected neural network that satisfies Assumption 2 and is trained by any GRW satisfying Assumption 1 with the squared loss. Let $f_{\text{ERM}}^{(t)}$ be the same model trained by ERM from the same initial point. If $d_1 = \cdots = d_L = \tilde{d}$, $\nabla_\theta f^{(0)}(\boldsymbol{x}_1), \cdots, \nabla_\theta f^{(0)}(\boldsymbol{x}_n)$ are linearly independent, and $\lambda^{\min} > 0$, then there exists a constant $\eta_1 > 0$ such that: if $\eta \leq \eta_1$[5], then for any $\delta > 0$, there exists $\tilde{D} > 0$ such that as long as $\tilde{d} \geq \tilde{D}$, with probability at least $(1 - \delta)$ over random initialization we have: for any test point $\boldsymbol{x} \in \mathbb{R}^d$ such that $\|\boldsymbol{x}\|_2 \leq 1$, as $\tilde{d} \to \infty$,*

$$\limsup_{t \to \infty} \left| f^{(t)}(\boldsymbol{x}) - f_{\text{ERM}}^{(t)}(\boldsymbol{x}) \right| = O(\tilde{d}^{-1/4}) \to 0 \qquad (10)$$

Essentially this theorem implies that on any test point $\boldsymbol{x}$ in the unit ball, the GRW model and the ERM model produce almost the same output, so they have almost the same performance. Note that for simplicity, we only prove for $d_1 = \cdots = d_L = \tilde{d} \to \infty$, but the result can be very easily extended to the case where $d_l/d_1 \to \alpha_l$ for $l = 2, \cdots, L$ for some constants $\alpha_2, \cdots, \alpha_L$, and $d_1 \to \infty$. The key to proving this theorem is to consider the *linearized neural network* of $f^{(t)}(\boldsymbol{x})$:

$$f_{\text{lin}}^{(t)}(\boldsymbol{x}) = f^{(0)}(\boldsymbol{x}) + \langle \theta^{(t)} - \theta^{(0)}, \nabla_\theta f^{(0)}(\boldsymbol{x}) \rangle \qquad (11)$$

which is a linear model *w.r.t.* $\nabla_\theta f^{(0)}(\boldsymbol{x})$. If $\nabla_\theta f^{(0)}(\boldsymbol{x}_1), \cdots, \nabla_\theta f^{(0)}(\boldsymbol{x}_n)$ are linearly independent (which is almost surely true when the model is overparameterized so that $\theta$ has a very high dimension), then our key intuition tells us that the linearized NN will converge to the unique interpolator. Then we show that the wide NN can be approximated by its linearized counterpart *uniformly throughout training*, which is considerably more subtle in our case due to the GRW dynamics. Here we prove the upper bound $O(\tilde{d}^{-1/4})$, but in fact the upper bound can be $O(\tilde{d}^{-1/2+\epsilon})$ for any $\epsilon > 0$:

---

[3] $f$ is *Lipschitz* if there exists a constant $L > 0$ such that for any $\boldsymbol{x}_1, \boldsymbol{x}_2$, $|f(\boldsymbol{x}_1) - f(\boldsymbol{x}_2)| \leq L \|\boldsymbol{x}_1 - \boldsymbol{x}_2\|_2$.

[4] *Non-degenerate* means that $\Theta(\boldsymbol{x}, \boldsymbol{x}')$ depends on $\boldsymbol{x}$ and $\boldsymbol{x}'$ and is not a constant.

[5] For ease of understanding, later we will write this condition as "with a sufficiently small learning rate".

**Lemma 5** (Approximation Theorem). *For a wide fully-connected neural network $f^{(t)}$ satisfying Assumption 2 and is trained by any GRW satisfying Assumption 1 with the squared loss, let $f_{\text{lin}}^{(t)}$ be its linearized neural network trained by the same GRW (i.e. $q_i^{(t)}$ are the same for both networks for any $i$ and $t$). Under the conditions of Theorem 4, with a sufficiently small learning rate, for any $\delta > 0$, there exist constants $\tilde{D} > 0$ and $C > 0$ such that as long as $\tilde{d} \geq \tilde{D}$, with probability at least $(1 - \delta)$ over random initialization we have: for any test point $\boldsymbol{x} \in \mathbb{R}^d$ such that $\|\boldsymbol{x}\|_2 \leq 1$,*

$$\sup_{t \geq 0} \left| f_{\text{lin}}^{(t)}(\boldsymbol{x}) - f^{(t)}(\boldsymbol{x}) \right| \leq C \tilde{d}^{-1/4} \tag{12}$$

This lemma essentially says that throughout the GRW training process, on any test point $\boldsymbol{x}$ in the unit ball, the linear NN and the wide NN produce almost the same output. So far, we have shown that in a regression task, for both linear and wide NNs, GRW does not improve over ERM.

### 4.3 WIDE NEURAL NETWORKS, WITH $L_2$ REGULARIZATION

Previous work such as Sagawa et al. (2020a) proposed to improve GRW by adding $L_2$ penalty to the objective function. In this section, we thus study adding $L_2$ regularization to GRW algorithms:

$$\hat{\mathcal{R}}_{\boldsymbol{q}^{(t)}}^{\mu}(f) = \sum_{i=1}^{n} q_i^{(t)} \ell(f(\boldsymbol{x}_i), y_i) + \frac{\mu}{2} \left\| \theta - \theta^{(0)} \right\|_2^2 \tag{13}$$

At first sight, adding regularization seems to be a natural approach and should make a difference. Indeed, from the outset, we can easily show that with $L_2$ regularization, the GRW model and the ERM model are different unlike the case without regularization. As an concrete example, when $f$ is a linear model, $\ell$ is convex and smooth, then $\hat{\mathcal{R}}_{\boldsymbol{q}^{(t)}}^{\mu}(f)$ with static GRW is a convex smooth objective function, so under GD with a sufficiently small learning rate, the model will converge to the global minimizer (see Appendix D.1). Moreover, the global optimum $\theta^*$ satisfies $\nabla_\theta \hat{\mathcal{R}}_{\boldsymbol{q}^{(t)}}^{\mu}(f(\boldsymbol{x}; \theta^*)) = 0$, solving which yields $\theta^* = \theta^{(0)} + (\boldsymbol{X}\boldsymbol{Q}\boldsymbol{X}^\top + \mu\boldsymbol{I})^{-1}\boldsymbol{X}\boldsymbol{Q}(\boldsymbol{Y} - f^{(0)}(\boldsymbol{X}))$, which depends on $\boldsymbol{Q} = \text{diag}(q_1, \cdots, q_n)$, so adding $L_2$ regularization at least seems to yield different results from ERM (albeit whether it improves over ERM might depend on $q_1, \cdots, q_n$).

However, the following result shows that this regularization must be large enough to *significantly lower the training performance*, or the final model would still be close to the unregularized ERM model. We still denote the largest and smallest eigenvalues of the kernel Gram matrix $\Theta$ by $\lambda^{\max}$ and $\lambda^{\min}$. We use the subscript "reg" to refer to a regularized model (trained by minimizing (13)).

**Theorem 6.** *Suppose there exists $M_0 > 0$ s.t. $\left\| \nabla_\theta f^{(0)}(\boldsymbol{x}) \right\|_2 \leq M_0$ for all $\|\boldsymbol{x}\|_2 \leq 1$. If $\lambda^{\min} > 0$ and $\mu > 0$, then for a wide NN satisfying Assumption 2, and any GRW minimizing the squared loss with a sufficiently small learning rate $\eta$, if $d_1 = d_2 = \cdots = d_L = \tilde{d}$, $\nabla_\theta f^{(0)}(\boldsymbol{x}_1), \cdots, \nabla_\theta f^{(0)}(\boldsymbol{x}_n)$ are linearly independent, and the empirical training risk of $f_{\text{reg}}^{(t)}$ satisfies*

$$\limsup_{t \to \infty} \hat{\mathcal{R}}(f_{\text{reg}}^{(t)}) < \epsilon \tag{14}$$

*for some $\epsilon > 0$, then with a sufficiently small learning rate, as $\tilde{d} \to \infty$, with probability close to 1 over random initialization, for any $\boldsymbol{x}$ such that $\|\boldsymbol{x}\|_2 \leq 1$ we have*

$$\limsup_{t \to \infty} \left| f_{\text{reg}}^{(t)}(\boldsymbol{x}) - f_{\text{ERM}}^{(t)}(\boldsymbol{x}) \right| = O(\tilde{d}^{-1/4} + \sqrt{\epsilon}) \to O(\sqrt{\epsilon}) \tag{15}$$

*where $f_{\text{reg}}^{(t)}$ is trained by regularized GRW and $f_{\text{ERM}}^{(t)}$ by unregularized ERM from same initial points.*

This theorem essentially says that if the regularization is too small and the training error is close to zero, then the regularized GRW model still produces an output very close to that of the ERM model on any test point $\boldsymbol{x}$ in the unit ball. Thus, a small regularization makes little difference.

The proof again starts from analyzing linearized NNs, and showing that regularization does not help there (Appendix D.4.2). Then, we prove a new approximation theorem for $L_2$ regularized GRW connecting wide NNs to linearized NNs uniformly throughout training (Appendix D.4.1). With regularization, we no longer need Assumption 1 to prove the new approximation theorem which was used to prove the convergence of GRW, because with regularization GRW naturally converges.

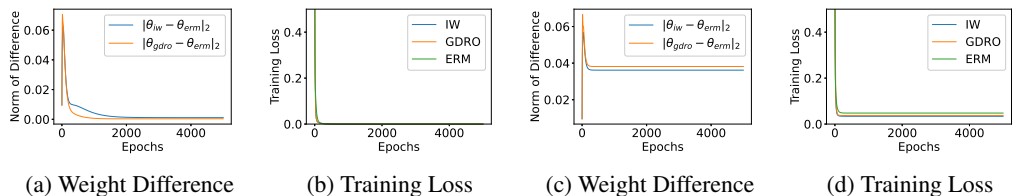

(a) Weight Difference      (b) Training Loss      (c) Weight Difference      (d) Training Loss

Figure 2: Experimental results of ERM, importance weighting (IW) and Group DRO (GDRO) with $L_2$ regularization with the squared loss. Left two: $\mu = 0.1$; Right two: $\mu = 10$.

To empirically demonstrate this result, we run the same experiment as in Section 4.1 but with $L_2$ regularization. The results are presented in Figure 2. We can see that when the regularization is small, the training losses still converge to 0, and the three model weights still converge to the same point. On the contrary, with a large regularization, the training loss does not converge to 0, and the three model weights converge to different points. This shows that the regularization must be large enough to lower the training performance to make a significant difference to the implicit bias.

## 5 THEORETICAL RESULTS FOR CLASSIFICATION

Now we consider classification where $\mathcal{Y} = \{+1, -1\}$. The big difference is that *classification losses don't have finite minimizers*. A classification loss converging to zero means that the model weight "explodes" to infinity instead of converging to a finite point. We focus on the canonical logistic loss:

$$\ell(\hat{y}, y) = \log(1 + \exp(-\hat{y}y)) \tag{16}$$

### 5.1 LINEAR MODELS

We first consider training the linear model $f(\boldsymbol{x}) = \langle \theta, \boldsymbol{x} \rangle$ with GRW under gradient descent with the logistic loss. As noted earlier, in this setting, Byrd & Lipton (2019) made the empirical observation that importance weighting does not improve over ERM. Then, Xu et al. (2021) proved that for importance weighting algorithms, as $t \to \infty$, $\|\theta^{(t)}\|_2 \to \infty$ and $\theta^{(t)}/\|\theta^{(t)}\|_2$ converges to a unit vector that does not depend on the sample weights, so it does not improve over ERM. To extend this theoretical result to the broad class of GRW algorithms, we will prove two results. First, in Theorem 7 we will show that for the logistic loss and any GRW algorithm satisfying the weaker assumption:

**Assumption 3.** For all $i$, $\liminf_{t \to \infty} q_i^{(t)} > 0$,

if the training error converges to 0, and the direction of the model weight converges to a fixed unit vector, then this unit vector must be the *max-margin classifier* defined as

$$\hat{\theta}_{\mathrm{MM}} = \arg\max_{\theta : \|\theta\|_2 = 1} \left\{ \min_{i=1,\cdots,n} y_i \cdot \langle \theta, \boldsymbol{x}_i \rangle \right\} \tag{17}$$

Second, Theorem 8 shows that for any GRW satisfying Assumption 1, the training error converges to 0 and the direction of the model weight converges, so it does not improve over ERM.

**Theorem 7.** *If $\boldsymbol{x}_1, \cdots, \boldsymbol{x}_n$ are linearly independent, then for the logistic loss, we have: for any GRW satisfying Assumption 3, if as $t \to \infty$ the empirical training risk $\hat{\mathcal{R}}(f^{(t)})$ converges to 0 and $\theta^{(t)}/\|\theta^{(t)}\|_2 \to \boldsymbol{u}$ for some unit vector $\boldsymbol{u}$, then $\boldsymbol{u} = \hat{\theta}_{MM}$.*

This result is an extension of Soudry et al. (2018), and says that all GRW methods (including ERM) make the model converge to the same point $\hat{\theta}_{\mathrm{MM}}$ that does not depend on $q_i^{(t)}$. In other words, the samples weights do not affect the implicit bias. Thus, for any GRW method that only satisfies the weak Assumption 3, as long as the training error converges to 0 and the model weight direction converges, GRW does not improve over ERM. We next show that any GRW satisfying Assumption 1 does have its model weight direction converge, and its training error converge to 0.

**Theorem 8.** *For any loss $\ell$ that is convex, $L$-smooth in $\hat{y}$ and strictly monotonically decreasing to zero as $y\hat{y} \to +\infty$, and GRW satisfying Assumption 1, denote $F(\theta) = \sum_{i=1}^{n} q_i \ell(\langle \theta, \boldsymbol{x}_i \rangle, y_i)$. If $\boldsymbol{x}_1, \cdots, \boldsymbol{x}_n$ are linearly independent, then with a sufficiently small learning rate $\eta$, we have:*

*(i)*    $F(\theta^{(t)}) \to 0$ *as $t \to \infty$.*          *(ii)*    $\|\theta^{(t)}\|_2 \to \infty$ *as $t \to \infty$.*

*(iii)* *Let $\theta_R = \arg\min_\theta \{F(\theta) : \|\theta\|_2 \le R\}$. $\theta_R$ is unique for any $R$ such that $\min_{\|\theta\|_2 \le R} F(\theta) < \min_i q_i \ell(0, y_i)$. And if $\lim_{R \to \infty} \frac{\theta_R}{R}$ exists, then $\lim_{t \to \infty} \frac{\theta^{(t)}}{\|\theta^{(t)}\|_2}$ also exists and they are equal.*

This result is an extension of Theorem 1 of Ji et al. (2020). For the logistic loss, it is easy to show that it satisfies the conditions of the above theorem and $\lim_{R \to \infty} \frac{\theta_R}{R} = \hat{\theta}_{\text{MM}}$. Thus, Theorems 8 and 7 together imply that all GRW satisfying Assumption 1 (including ERM) have the same implicit bias (see Appendix D.5.3). We also have empirical verification for these results (see Appendix C).

**Remark.** It is impossible to extend these results to wide NNs like Theorem 4 because for a neural network, if $\|\theta^{(t)}\|_2$ goes to infinity, then $\|\nabla_\theta f\|_2$ will also go to infinity. However, for a linear model, the gradient is a constant. Consequently, the gap between the neural networks and its linearized counterpart will "explode" under gradient descent, so there can be no approximation theorem like Lemma 5 that can connect wide NNs to their linearized counterparts. Thus, we consider regularized GRW, for which $\theta^{(t)}$ converges to a finite point and there is an approximation theorem.

## 5.2 WIDE NEURAL NETWORKS, WITH $L_2$ REGULARIZATION

Consider minimizing the regularized weighted empirical risk (13) with $\ell$ being the logistic loss. As in the regression case, with $L_2$ regularization, GRW methods have different implicit biases than ERM for the same reasons as in Section 4.3. And similarly, we can show that in order for GRW methods to be sufficiently different from ERM, the regularization needs to be large enough to significantly lower the training performance. Specifically, in the following theorem we show that if the regularization is too small to lower the training performance, then a wide neural network trained with regularized GRW and the logistic loss will still be very close to the *max-margin linearized neural network*:

$$f_{\text{MM}}(\boldsymbol{x}) = \langle \hat{\theta}_{\text{MM}}, \nabla_\theta f^{(0)}(\boldsymbol{x}) \rangle \quad \text{where} \quad \hat{\theta}_{\text{MM}} = \arg\max_{\|\theta\|_2=1} \left\{ \min_{i=1,\cdots,n} y_i \cdot \langle \theta, \nabla_\theta f^{(0)}(\boldsymbol{x}_i) \rangle \right\} \quad (18)$$

Note that $f_{\text{MM}}$ does not depend on $q_i^{(t)}$. Moreover, using the result in the previous section we can show that a linearized neural network trained with unregularized ERM will converge to $f_{\text{MM}}$:

**Theorem 9.** *Suppose there exists $M_0 > 0$ such that $\left\|\nabla_\theta f^{(0)}(\boldsymbol{x})\right\|_2 \leq M_0$ for all test point $\boldsymbol{x}$. For a wide NN satisfying Assumption 2, and for any GRW satisfying Assumption 1 with the logistic loss, if $d_1 = d_2 = \cdots = d_L = \tilde{d}$ and $\nabla_\theta f^{(0)}(\boldsymbol{x}_1), \cdots, \nabla_\theta f^{(0)}(\boldsymbol{x}_n)$ are linearly independent and the learning rate is sufficiently small, then for any $\delta > 0$ there exists a constant $C > 0$ such that: with probability at least $(1 - \delta)$ over random initialization, as $\tilde{d} \to \infty$ we have: for any $\epsilon \in (0, \frac{1}{4})$, if the empirical training error satisfies $\limsup_{t \to \infty} \hat{\mathcal{R}}(f_{\text{reg}}^{(t)}) < \epsilon$, then for any test point $\boldsymbol{x}$ such that $|f_{MM}(\boldsymbol{x})| > C \cdot (-\log 2\epsilon)^{-1/2}$, $f_{\text{reg}}^{(t)}(\boldsymbol{x})$ has the same sign as $f_{MM}(\boldsymbol{x})$ when $t$ is sufficiently large.*

This result says that at any test point $\boldsymbol{x}$ on which the max-margin linear classifier classifies with a margin of $\Omega((-\log 2\epsilon)^{-1/2})$, the neural network has the same prediction. And as $\epsilon$ decreases, the confidence threshold also becomes lower. Similar to Theorem 6, this theorem provides the scaling of the gap between the regularized GRW model and the unregularized ERM model *w.r.t.* $\epsilon$.

This result justifies the empirical observation in Sagawa et al. (2020a) that with large regularization, some GRW algorithms can maintain a high worst-group test performance, with the cost of suffering a significant drop in training accuracy. On the other hand, if the regularization is small and the model can achieve nearly perfect training accuracy, then its worst-group test performance will still significantly drop.

## 6 DISCUSSION

### 6.1 DISTRIBUTIONALLY ROBUST GENERALIZATION AND FUTURE DIRECTIONS

A large body of prior work focused on distributionally robust optimization, but we show that these methods have (almost) equivalent implicit biases as ERM. In other words, **distributionally robust optimization (DRO) does not necessarily achieve better distributionally robust generalization (DRG).** Our results pinpoint a critical bottleneck in the current distribution shift research, and we argue that a deeper understanding in DRG is crucial for developing better distributionally robust training algorithms. Here we discuss three promising future directions to improving DRG.

The first approach is data augmentation and pretraining on large datasets. Our theoretical findings suggest that the implicit bias of GRW is determined by the training samples and the initial point, but not the sample weights. Thus, to improve DRG, we can either obtain more training samples, or start from a better initial point, as proposed in two recent papers (Wiles et al., 2022; Sagawa et al., 2022).

The second approach (for classification) is to go beyond the class of (iterative) sample reweighting based GRW algorithms, for instance via *logit adjustment* (Menon et al., 2021), which makes a classifier *have larger margins on smaller groups* to improve its generalization on smaller groups. An early approach by Cao et al. (2019) proposed to add an $O(n_k^{-1/4})$ additive adjustment term to the logits output by the classifier. Following this spirit, Menon et al. (2021) proposed the LA-loss which also adds an additive adjustment term to the logits. Ye et al. (2020) proposed the CDT-loss which adds a multiplicative adjustment term to the logits by dividing the logits of different classes with different temperatures. Kini et al. (2021) proposed the VS-loss which includes both additive and multiplicative adjustment terms, and they showed that only the multiplicative adjustment term affects the implicit bias, while the additive term only affects optimization, a fact that can be easily derived from our Theorem 8. Finally, Li et al. (2021a) proposed AutoBalance which optimizes the adjustment terms with a bi-level optimization framework.

The third approach is to stay within the class of GRW algorithms, but to change the classification/regression loss function to be suited to GRW. A recent paper (Wang et al., 2022) showed that for linear classifiers, one can make the implicit bias of GRW dependent on the sample weights by replacing the exponentially-tailed logistic loss with the following *polynomially-tailed loss*:

$$\ell_{\alpha,\beta}(\hat{y}, y) = \begin{cases} \ell_{\text{left}}(\hat{y}y) & \text{, if } \hat{y}y < \beta \\ \dfrac{1}{[\hat{y}y - (\beta - 1)]^\alpha} & \text{, if } \hat{y}y \geq \beta \end{cases} \tag{19}$$

And this result can be extended to GRW satisfying Assumption 1 using our Theorem 8. The reason why loss (19) works is that it changes $\lim_{R \to \infty} \frac{\theta_R}{R}$, and the new limit depends on the sample weights.

## 6.2 LIMITATIONS

Like most theory papers, our work makes some strong assumptions. The two main assumptions are:

  (i) The model is a linear model or a sufficiently wide fully-connected neural network.
 (ii) The model is trained for sufficiently long time, *i.e.* without early stopping.

Regarding (i), Chizat et al. (2019) argued that NTK neural networks fall in the "lazy training" regime and results might not be transferable to general neural networks. However, this class of neural networks has been widely studied in recent years and has provided considerable insights into the behavior of general neural networks, which is hard to analyze otherwise. Regarding (ii), in some easy tasks, when early stopping is applied, existing algorithms for distributional shift can do better than ERM (Sagawa et al., 2020a). However, as demonstrated in Gulrajani & Lopez-Paz (2021); Koh et al. (2021), in real applications these methods still cannot significantly improve over ERM even with early stopping, so early stopping is not the ultimate universal solution. Thus, though inevitably our results rely on some strong assumptions, we believe that they provide important insights into the problems of existing methods and directions for future work, which are significant contributions to the study of distributional shift problems.

## 7 CONCLUSION

In this work, we posit a broad class of what we call Generalized Reweighting (GRW) algorithms that include popular approaches such as importance weighting, and Distributionally Robust Optimization (DRO) variants, that were designed towards the task of learning models that are robust to distributional shift. We show that when used to train overparameterized linear models or wide NN models, even this very broad class of GRW algorithms does not improve over ERM, because they have the same implicit biases. We also showed that regularization does not help if it is not large enough to significantly lower the average training performance. Our results thus suggest to make progress towards learning models that are robust to distributional shift, we have to either go beyond this broad class of GRW algorithms, or design new losses specifically targeted to this class.

ACKNOWLEDGMENTS

We acknowledge the support of NSF via OAC-1934584, IIS-1909816, DARPA via HR00112020006, and ARL.

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

# A    RELATED WORK

## A.1    SUBPOPULATION SHIFT

In this work, we mainly focus on the subpopulation shift problem, which has two main applications: group fairness and long-tailed learning (learning with class imbalance). In both applications, the dataset can be divided into several subgroups, and this work considers minimizing the worst-group risk, defined as the maximum risk over any group.

**Group Fairness.**    Group fairness refers to the scenario where the dataset contains several "groups" (such as demographic groups), and a model is considered fair if its per-group performances meet certain criteria (a "fairness" notion). Group fairness in machine learning was first studied in Hardt et al. (2016) and Zafar et al. (2017), where they required the model to perform equally well over all groups. Many previous papers proposed a number of fairness notions, such as equal opportunity, statistical parity, etc. Among them, Hashimoto et al. (2018) studied another type of group fairness called Rawlsian max-min fairness Rawls (2001), which does not require equal performance but rather requires high performance on the worst-off group. The subpopulation shift problem we study in this paper is naturally connected to the Rawlsian max-min fairness. A large body of recent work have studied how to improve this worst-group performance Duchi & Namkoong (2018); Oren et al. (2019); Liu et al. (2021); Zhai et al. (2021a). Recent work however observe that these approaches, when used with modern overparameterized models, easily overfit Sagawa et al. (2020a;b). Apart from group fairness, there are also other notions of fairness, such as individual fairness Dwork et al. (2012); Zemel et al. (2013) and counterfactual fairness Kusner et al. (2017), which we do not study in this work.

**Long-tailed Learning.**    Long-tailed learning refers to the scenario where different classes have different sizes, and usually there are some "minority classes" with extremely few samples that are much more difficult to learn than the other classes. Using GRW such as importance weighting for long-tailed learning is a very old idea which dates back to Xie & Manski (1989). However, recently Byrd & Lipton (2019) found that the effect of importance weighting for long-tailed learning diminishes as training proceeds, which leads to a line of recent work on how to improve the generalization in long-tailed learning (Cao et al., 2019; Menon et al., 2021; Ye et al., 2020; Kim & Kim, 2020; Kini et al., 2021). Most of these papers share a common idea: Forcing the model to have larger margins on smaller groups, so that its generalization on smaller groups can be better. Self-supervised learning is also used in long-tailed learning. For instance, Liu et al. (2022) found that self-supervised learning can achieve good performances in long-tailed learning, Wang et al. (2021) used contrastive learning for long-tailed learning, and Li et al. (2021b) used self-distillation.

## A.2    DOMAIN GENERALIZATION

Domain generalization is the second common type of distribution shift. In domain generalization and the related domain adaptation, a model is tested on a different domain than what it is trained on. The most common idea in domain generalization is *invariant learning*, which learns a feature extractor that is invariant across domains, usually by matching the feature distribution of different domains. Since we have no access to the target domain, in invariant learning we assume that we have access to multiple domains in the training set, and we learn a feature extractor with a small variance across these domains. Examples include CORAL (Sun & Saenko, 2016), DANN (Ganin et al., 2016), MMD (Li et al., 2018) and IRM (Arjovsky et al., 2019). However, Gulrajani & Lopez-Paz (2021); Koh et al. (2021) empirically showed that most of these methods cannot do better than standard ERM, and Rosenfeld et al. (2021) theoretically proved that IRM cannot do better than ERM unless the number of training domains is greater than the number of independent features.

One problem of invariant learning methods is that they do not necessarily align the classes. For a source domain $P$ and a target domain $Q$, even if we have successfully learned a feature extractor $\Phi$ such that $\Phi(P) \approx \Phi(Q)$, there is no guarantee that $\Phi$ can map the samples in $P$ and $Q$ from the same class to the same location in the feature space. In the worst case, $\Phi$ can map the positive samples in $P$ and the negative samples in $Q$ to the same location and vice versa, in which case 100% accuracy over $P$ means 0% accuracy over $Q$. The goal of class alignment is to make sure that samples from the same class are mapped together, and far away from the other classes. For

example, Tzeng et al. (2015) used soft labels to align the classes, Long et al. (2016) minimized the class-based cross entropy on the target domain while keeping the source and target classifiers close with a residual block, and Motiian et al. (2017) adopted a similarity penalty to keep samples from different classes away from each other.

### A.3 IMPLICIT BIAS UNDER THE OVERPARAMETERIZED SETTING

For overparameterized models, there could be many model parameters which all minimize the training loss. In such cases, it is of interest to study the implicit bias of specific optimization algorithms such as gradient descent i.e. to what minimizer the model parameters will converge to Du et al. (2019); Allen-Zhu et al. (2019). Our results use the NTK formulation of wide neural networks Jacot et al. (2018), and specifically we use linearized neural networks to approximate such wide neural networks following Lee et al. (2019). There is some criticism of this line of work, e.g. Chizat et al. (2019) argued that infinitely wide neural networks fall in the "lazy training" regime and results might not be transferable to general neural networks. Nonetheless such wide neural networks are being widely studied in recent years, since they provide considerable insights into the behavior of more general neural networks, which are typically intractable to analyze otherwise.

### A.4 COMPARISON WITH HU ET AL. (2018)

A prior work Hu et al. (2018) also proved that GRW is equivalent to ERM under certain conditions. However, we would like to point out that this work is *substantially different* from Hu et al. (2018). Hu et al. (2018) proved that in classification that uses the zero-one loss, GRW methods such as DRSL are equivalent to ERM, in the sense that the minimizer of the DRSL risk is also the minimizer of the average risk. However, this does not mean that DRSL and ERM will always converge to the *same point*, as there could be multiple minimizers. Their result relies on the zero-one loss, which leads to a monotonic linear relationship between the DRSL risk and the average risk. Moreover, their result is only about the relationship between two minimizers, and they did not prove that DRSL and ERM can actually converge to these global minima.

On the other hand, in our results, we first show that without regularization, GRW and ERM will *converge to the exact same point*, so that they have equivalent implicit biases, which is a much stronger result. Then we show that even with regularization, if the regularization is not large enough, GRW will still converge to a point that is very close to the point ERM converges to. Our results do not depend on the loss function, and work for both the squared loss for regression and the logistic loss for classification (and can be extended to other losses). Instead, our results depend on the optimization method (must be first-order or gradient-based) as well as the model architecture (linear or wide NN), since we need to explicitly prove that both GRW and ERM can reach the global minima if trained under a small learning rate for sufficiently long. In a word, Hu et al. (2018) proves the equivalence between GRW and ERM under the zero-one loss with a monotonic relationship between the two risk functions, while our results focus on the optimization and training dynamics, and prove that GRW and ERM have almost equivalent implicit biases.

## B EXTENSION TO MULTI-DIMENSIONAL REGRESSION / MULTI-CLASS CLASSIFICATION

In our results, we assume that $f : \mathbb{R}^d \to \mathbb{R}$ for simplicity, but our results can be very easily extended to the case where $f : \mathbb{R}^d \to \mathbb{R}^k$. For most of our results, the proof consists of two major components: (i) The linearized neural network will converge to some point (interpolator, max-margin classifier, etc.); (ii) The wide fully-connected neural network can be approximated by its linearized counterpart. For both components, the extension is very simple and straightforward. For (i), the proof only relies on the smoothness of the objective function and the upper quadratic bound it entails, and the function is still smooth when its output becomes multi-dimensional; For (ii), we can prove that $\sup_t \|f(\boldsymbol{x}) - f_{\text{lin}}(\boldsymbol{x})\|_2 = O(\tilde{d}^{-1/4})$ in exactly the same way. Thus, all of our results hold for multi-dimensional regression and multi-class classification.

Particularly, for the multi-class cross-entropy loss, using Theorem 8 we can show that under any GRW satisfying Assumption 1, the direction of the weight of a linear classifier will converge to the

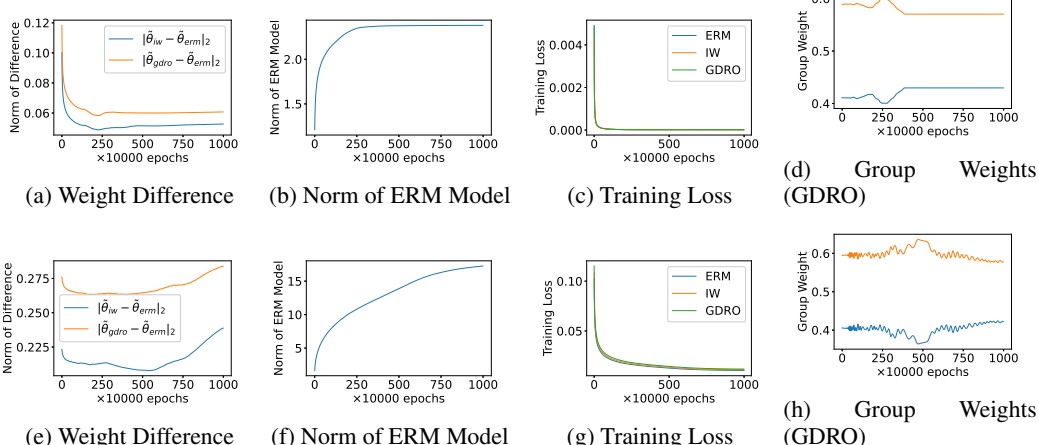

Figure 3: Experimental results of ERM, importance weighting (IW) and Group DRO (GDRO) with the logistic loss and the polynomially-tailed loss. First row: Logistic loss; Second row: Polynomially-tailed loss. All norms are $L_2$ norms. $\tilde{\theta}$ is a unit vector which is the direction of $\theta$.

following max-margin classifier:

$$\hat{\theta}_{\text{MM}} = \arg\min_{\theta} \left\{ \min_{i=1,\cdots,n} \left[ f(\boldsymbol{x}_i)_{y_i} - \max_{y' \neq y_i} f(\boldsymbol{x}_i)_{y'} \right] : \|\theta\|_2 = 1 \right\} \tag{20}$$

which is still independent of $q_i$.

## C  MORE EXPERIMENTS

We run ERM, importance weighting and Group DRO on the training set with 6 MNIST images which we used in Section 4.1 with the logistic loss and the polynomially-tailed loss (Eqn. (19), with $\alpha = 1$, $\beta = 0$ and $\ell_{\text{left}}$ being the logistic loss shifted to make the overall loss function continuous) on this dataset for 10 million epochs (note that we run for much more epochs because the convergence is very slow). The results are shown in Figure 3. From the plots we can see that:

- For either loss function, the training loss of each method converges to 0.
- In contrast to the theory that the norm of the ERM model will go to infinity and all models will converge to the max-margin classifier, the weight of the ERM model gets stuck at some point, and the norms of the gaps between the normalized model weights also get stuck. The reason is that the training loss has got so small that it becomes zero in the floating number representation, so the gradient also becomes zero and the training halts due to limited computational precision.
- However, we can still observe a fundamental difference between the logistic loss and the polynomially-tailed loss. For the logistic loss, the norm of the gap between importance weighting (or Group DRO) and ERM will converge to around 0.06 when the training stops, while for the polynomially-tailed loss, the norm will be larger than 0.22 and will keep growing, which shows that for the polynomially-tailed loss the normalized model weights do not converge to the same point.
- For either loss, the group weights of Group DRO still empirically satisfy Assumption 1.

## D  PROOFS

In this paper, for any matrix $\boldsymbol{A}$, we will use $\|\boldsymbol{A}\|_2$ to denote its spectral norm and $\|\boldsymbol{A}\|_F$ to denote its Frobenius norm.

### D.1 BACKGROUND ON SMOOTHNESS

A first-order differentiable function $f$ over $\mathcal{D}$ is called *L-smooth* for $L > 0$ if

$$f(\boldsymbol{y}) \le f(\boldsymbol{x}) + \langle \nabla f(\boldsymbol{x}), \boldsymbol{y} - \boldsymbol{x} \rangle + \frac{L}{2} \|\boldsymbol{y} - \boldsymbol{x}\|_2^2 \qquad \forall \boldsymbol{x}, \boldsymbol{y} \in \mathcal{D} \tag{21}$$

which is also called the *upper quadratic bound*. If $f$ is second-order differentiable and $\mathcal{D}$ is a convex set, then $f$ is $L$-smooth is equivalent to

$$\boldsymbol{v}^\top \nabla^2 f(\boldsymbol{x}) \boldsymbol{v} \le L \qquad \forall \|\boldsymbol{v}\|_2 = 1, \forall \boldsymbol{x} \in \mathcal{D} \tag{22}$$

A classical result in convex optimization is the following:

**Theorem 10.** *If $f(\boldsymbol{x})$ is convex and $L$-smooth with a unique finite minimizer $\boldsymbol{x}^*$, and is minimized by gradient descent $\boldsymbol{x}_{t+1} = \boldsymbol{x}_t - \eta \nabla f(\boldsymbol{x}_t)$ starting from $\boldsymbol{x}_0$ where the learning rate $\eta \le \frac{1}{L}$, then we have*

$$f(\boldsymbol{x}_T) \le f(\boldsymbol{x}^*) + \frac{1}{\eta T} \|\boldsymbol{x}_0 - \boldsymbol{x}^*\|_2^2 \tag{23}$$

*which also implies that $\boldsymbol{x}_T$ converges to $\boldsymbol{x}^*$ as $T \to \infty$.*

### D.2 PROOFS FOR SUBSECTION 4.1

#### D.2.1 PROOF OF THEOREM 1

Using the key intuition, the weight update rule (7) implies that $\theta^{(t+1)} - \theta^{(t)} \in \text{span}\{\boldsymbol{x}_1, \cdots, \boldsymbol{x}_n\}$ for all $t$, which further implies that $\theta^{(t)} - \theta^{(0)} \in \text{span}\{\boldsymbol{x}_1, \cdots, \boldsymbol{x}_n\}$ for all $t$. By Cramer's rule, in this $n$-dimensional subspace there exists one and only one $\theta^*$ such that $\theta^* - \theta^{(0)} \in \text{span}\{\boldsymbol{x}_1, \cdots, \boldsymbol{x}_n\}$ and $\langle \theta^*, \boldsymbol{x}_i \rangle$ for all $i$. Then we have

$$\left\| \boldsymbol{X}^\top (\theta^{(t)} - \theta^*) \right\|_2 = \left\| (\boldsymbol{X}^\top \theta^{(t)} - \boldsymbol{Y}) - (\boldsymbol{X}^\top \theta^* - \boldsymbol{Y}) \right\|_2 \le \left\| \boldsymbol{X}^\top \theta^{(t)} - \boldsymbol{Y} \right\|_2 + \left\| \boldsymbol{X}^\top \theta^* - \boldsymbol{Y} \right\|_2 \to 0 \tag{24}$$

because $\left\| \boldsymbol{X}^\top \theta - \boldsymbol{Y} \right\|_2^2 = 2n\hat{\mathcal{R}}(f(\boldsymbol{x}; \theta))$. On the other hand, let $s^{\min}$ be the smallest singular value of $\boldsymbol{X}$. Since $\boldsymbol{X}$ is full-rank, $s^{\min} > 0$, and $\left\| \boldsymbol{X}^\top (\theta^{(t)} - \theta^*) \right\|_2 \ge s^{\min} \left\| \theta^{(t)} - \theta^* \right\|_2$. This shows that $\left\| \theta^{(t)} - \theta^* \right\|_2 \to 0$. Thus, $\theta^{(t)}$ converges to this unique $\theta^*$. $\qquad\square$

#### D.2.2 PROOF OF THEOREM 2

To help our readers understand the proof more easily, we will first prove the result for static GRW where $q_i^{(t)} = q_i$ for all $t$, and then we will prove the result for dynamic GRW that satisfy $q_i^{(t)} \to q_i$ as $t \to \infty$.

**Static GRW.** We first prove the result for all static GRW such that $\min_i q_i = q^* > 0$.

We will use smoothness introduce in Appendix D.1. Denote $A = \sum_{i=1}^n \|\boldsymbol{x}_i\|_2^2$. The empirical risk of the linear model $f(\boldsymbol{x}) = \langle \theta, \boldsymbol{x} \rangle$ is

$$F(\theta) = \sum_{i=1}^n q_i (\boldsymbol{x}_i^\top \theta - y_i)^2 \tag{25}$$

whose Hessian is

$$\nabla_\theta^2 F(\theta) = 2 \sum_{i=1}^n q_i \boldsymbol{x}_i \boldsymbol{x}_i^\top \tag{26}$$

So for any unit vector $\boldsymbol{v} \in \mathbb{R}^d$, we have (since $q_i \in [0, 1]$)

$$\boldsymbol{v}^\top \nabla_\theta^2 F(\theta) \boldsymbol{v} = 2 \sum_{i=1}^n q_i (\boldsymbol{x}_i^\top \boldsymbol{v})^2 \le 2 \sum_{i=1}^n q_i \|\boldsymbol{x}_i\|_2^2 \le 2A \tag{27}$$

which implies that $F(\theta)$ is $2A$-smooth. Thus, we have the following upper quadratic bound: for any $\theta_1, \theta_2 \in \mathbb{R}^d$,

$$F(\theta_2) \leq F(\theta_1) + \langle \nabla_\theta F(\theta_1), \theta_2 - \theta_1 \rangle + A \|\theta_2 - \theta_1\|_2^2 \tag{28}$$

Denote $g(\theta^{(t)}) = (\boldsymbol{X}^\top \theta^{(t)} - \boldsymbol{Y}) \in \mathbb{R}^n$. We can see that $\left\| \sqrt{\boldsymbol{Q}} g(\theta^{(t)}) \right\|_2^2 = F(\theta^{(t)})$, where where $\sqrt{\boldsymbol{Q}} = \text{diag}(\sqrt{q_1}, \cdots, \sqrt{q_n})$. Thus, $\nabla F(\theta^{(t)}) = 2\boldsymbol{X} \boldsymbol{Q} g(\theta^{(t)})$. The update rule of a static GRW with gradient descent and the squared loss is:

$$\theta^{(t+1)} = \theta^{(t)} - \eta \sum_{i=1}^n q_i \boldsymbol{x}_i (f^{(t)}(\boldsymbol{x}_i) - y_i) = \theta^{(t)} - \eta \boldsymbol{X} \boldsymbol{Q} g(\theta^{(t)}) \tag{29}$$

Substituting $\theta_1$ and $\theta_2$ in (28) with $\theta^{(t)}$ and $\theta^{(t+1)}$ yields

$$F(\theta^{(t+1)}) \leq F(\theta^{(t)}) - 2\eta g(\theta^{(t)})^\top \boldsymbol{Q}^\top \boldsymbol{X}^\top \boldsymbol{X} \boldsymbol{Q} g(\theta^{(t)}) + A \left\| \eta \boldsymbol{X} \boldsymbol{Q} g(\theta^{(t)}) \right\|_2^2 \tag{30}$$

Since $\boldsymbol{x}_1, \cdots, \boldsymbol{x}_n$ are linearly independent, $\boldsymbol{X}^\top \boldsymbol{X}$ is a positive definite matrix. Denote the smallest eigenvalue of $\boldsymbol{X}^\top \boldsymbol{X}$ by $\lambda^{\min} > 0$. And $\left\| \boldsymbol{Q} g(\theta^{(t)}) \right\|_2 \geq \sqrt{q^*} \left\| g(\theta^{(t)}) \right\|_2 = \sqrt{q^* F(\theta^{(t)})}$, so we have $g(\theta^{(t)})^\top \boldsymbol{Q}^\top \boldsymbol{X}^\top \boldsymbol{X} \boldsymbol{Q} g(\theta^{(t)}) \geq q^* \lambda^{\min} F(\theta^{(t)})$. Thus,

$$\begin{aligned} F(\theta^{(t+1)}) &\leq F(\theta^{(t)}) - 2\eta q^* \lambda^{\min} F(\theta^{(t)}) + A\eta^2 \left\| \boldsymbol{X} \sqrt{\boldsymbol{Q}} \right\|_2^2 \left\| \sqrt{\boldsymbol{Q}} g(\theta^{(t)}) \right\|_2^2 \\ &\leq F(\theta^{(t)}) - 2\eta q^* \lambda^{\min} F(\theta^{(t)}) + A\eta^2 \left\| \boldsymbol{X} \sqrt{\boldsymbol{Q}} \right\|_F^2 F(\theta^{(t)}) \\ &\leq F(\theta^{(t)}) - 2\eta q^* \lambda^{\min} F(\theta^{(t)}) + A\eta^2 \|\boldsymbol{X}\|_F^2 F(\theta^{(t)}) \\ &= (1 - 2\eta q^* \lambda^{\min} + A^2 \eta^2) F(\theta^{(t)}) \end{aligned} \tag{31}$$

Let $\eta_0 = \frac{q^* \lambda^{\min}}{A^2}$. For any $\eta \leq \eta_0$, we have $F(\theta^{(t+1)}) \leq (1 - \eta q^* \lambda^{\min}) F(\theta^{(t)})$ for all $t$, which implies that $\lim_{t \to \infty} F(\theta^{(t)}) = 0$. This implies that the empirical training risk must converge to 0.

**Dynamic GRW.** Now we prove the result for all dynamic GRW satisfying Assumption 1. By Assumption 1, for any $\epsilon > 0$, there exists $t_\epsilon$ such that for all $t \geq t_\epsilon$ and all $i$,

$$q_i^{(t)} \in (q_i - \epsilon, q_i + \epsilon) \tag{32}$$

This is because for all $i$, there exists $t_i$ such that for all $t \geq t_i$, $q_i^{(t)} \in (q_i - \epsilon, q_i + \epsilon)$. Then, we can define $t_\epsilon = \max\{t_1, \cdots, t_n\}$. Denote the largest and smallest eigenvalues of $\boldsymbol{X}^\top \boldsymbol{X}$ by $\lambda^{\max}$ and $\lambda^{\min}$, and because $\boldsymbol{X}$ is full-rank, we have $\lambda^{\min} > 0$. Define $\epsilon = \min\{\frac{q^*}{3}, \frac{(q^* \lambda^{\min})^2}{12 \lambda^{\max 2}}\}$, and then $t_\epsilon$ is also fixed.

We still denote $\boldsymbol{Q} = \text{diag}(q_1, \cdots, q_n)$. When $t \geq t_\epsilon$, the update rule of a dynamic GRW with gradient descent and the squared loss is:

$$\theta^{(t+1)} = \theta^{(t)} - \eta \boldsymbol{X} \boldsymbol{Q}_\epsilon^{(t)} (\boldsymbol{X}^\top \theta^{(t)} - \boldsymbol{Y}) \tag{33}$$

where $\boldsymbol{Q}_\epsilon^{(t)} = \boldsymbol{Q}^{(t)}$, and we use the subscript $\epsilon$ to indicate that $\left\| \boldsymbol{Q}_\epsilon^{(t)} - \boldsymbol{Q} \right\|_2 < \epsilon$. Then, note that we can rewrite $\boldsymbol{Q}_\epsilon^{(t)}$ as $\boldsymbol{Q}_\epsilon^{(t)} = \sqrt{\boldsymbol{Q}_{3\epsilon}^{(t)}} \cdot \sqrt{\boldsymbol{Q}}$ as long as $\epsilon \leq q^*/3$. This is because $q_i + \epsilon \leq \sqrt{(q_i + 3\epsilon) q_i}$ and $q_i - \epsilon \geq \sqrt{(q_i - 3\epsilon) q_i}$ for all $\epsilon \leq q_i/3$, and $q_i \geq q^*$. Thus, we have

$$\theta^{(t+1)} = \theta^{(t)} - \eta \boldsymbol{X} \sqrt{\boldsymbol{Q}_{3\epsilon}^{(t)}} \sqrt{\boldsymbol{Q}} g(\theta^{(t)}) \quad \text{where } \boldsymbol{Q}_\epsilon^{(t)} = \sqrt{\boldsymbol{Q}_{3\epsilon}^{(t)}} \cdot \sqrt{\boldsymbol{Q}} \tag{34}$$

Again, substituting $\theta_1$ and $\theta_2$ in (28) with $\theta^{(t)}$ and $\theta^{(t+1)}$ yields

$$F(\theta^{(t+1)}) \leq F(\theta^{(t)}) - 2\eta g(\theta^{(t)})^\top \boldsymbol{Q}^\top \boldsymbol{X}^\top \boldsymbol{X} \sqrt{\boldsymbol{Q}_{3\epsilon}^{(t)}} \sqrt{\boldsymbol{Q}} g(\theta^{(t)}) + A \left\| \eta \boldsymbol{X} \sqrt{\boldsymbol{Q}_{3\epsilon}^{(t)}} \sqrt{\boldsymbol{Q}} g(\theta^{(t)}) \right\|_2^2 \tag{35}$$

Then, note that

$$
\begin{aligned}
&\left| g(\theta^{(t)})^\top \boldsymbol{Q}^\top \boldsymbol{X}^\top \boldsymbol{X} \left( \sqrt{\boldsymbol{Q}_{3\epsilon}^{(t)}} - \sqrt{\boldsymbol{Q}} \right) \sqrt{\boldsymbol{Q}} g(\theta^{(t)}) \right| \\
&\leq \left\| \sqrt{\boldsymbol{Q}}^\top \boldsymbol{X}^\top \boldsymbol{X} \left( \sqrt{\boldsymbol{Q}_{3\epsilon}^{(t)}} - \sqrt{\boldsymbol{Q}} \right) \right\|_2 \left\| \sqrt{\boldsymbol{Q}} g(\theta^{(t)}) \right\|_2^2 \\
&\leq \left\| \sqrt{\boldsymbol{Q}} \right\|_2 \left\| \boldsymbol{X}^\top \boldsymbol{X} \right\|_2 \left\| \sqrt{\boldsymbol{Q}_{3\epsilon}^{(t)}} - \sqrt{\boldsymbol{Q}} \right\|_2 \left\| \sqrt{\boldsymbol{Q}} g(\theta^{(t)}) \right\|_2^2 \\
&\leq \lambda^{\max} \sqrt{3\epsilon} F(\theta^{(t)})
\end{aligned}
\tag{36}
$$

where the last step comes from the following fact: for all $\epsilon < q_i/3$,

$$
\sqrt{q_i + 3\epsilon} - \sqrt{q_i} \leq \sqrt{3\epsilon} \quad \text{and} \quad \sqrt{q_i} - \sqrt{q_i - 3\epsilon} \leq \sqrt{3\epsilon}
\tag{37}
$$

And as proved before, we also have

$$
g(\theta^{(t)})^\top \boldsymbol{Q}^\top \boldsymbol{X}^\top \boldsymbol{X} \boldsymbol{Q} g(\theta^{(t)}) \geq q^* \lambda^{\min} F(\theta^{(t)})
\tag{38}
$$

Since $\epsilon \leq \frac{(q^* \lambda^{\min})^2}{12 \lambda^{\max 2}}$, we have

$$
g(\theta^{(t)})^\top \boldsymbol{Q}^\top \boldsymbol{X}^\top \boldsymbol{X} \sqrt{\boldsymbol{Q}_{3\epsilon}^{(t)}} \sqrt{\boldsymbol{Q}} g(\theta^{(t)}) \geq \left( q^* \lambda^{\min} - \lambda^{\max} \sqrt{3\epsilon} \right) F(\theta^{(t)}) \geq \frac{1}{2} q^* \lambda^{\min} F(\theta^{(t)})
\tag{39}
$$

Thus,

$$
\begin{aligned}
F(\theta^{(t+1)}) &\leq F(\theta^{(t)}) - \eta q^* \lambda^{\min} F(\theta^{(t)}) + A\eta^2 \left\| \boldsymbol{X} \sqrt{\boldsymbol{Q}_{3\epsilon}^{(t)}} \right\|_2^2 \left\| \sqrt{\boldsymbol{Q}} g(\theta^{(t)}) \right\|_2^2 \\
&\leq (1 - \eta q^* \lambda^{\min} + A^2 \eta^2 (1 + 3\epsilon)) F(\theta^{(t)}) \\
&\leq (1 - \eta q^* \lambda^{\min} + 2A^2 \eta^2) F(\theta^{(t)})
\end{aligned}
\tag{40}
$$

for all $\epsilon \leq 1/3$. Let $\eta_0 = \frac{q^* \lambda^{\min}}{4A^2}$. For any $\eta \leq \eta_0$, we have $F(\theta^{(t+1)}) \leq (1 - \eta q^* \lambda^{\min}/2) F(\theta^{(t)})$ for all $t \geq t_\epsilon$, which implies that $\lim_{t \to \infty} F(\theta^{(t)}) = 0$. Thus, the empirical training risk converges to 0. $\qquad \square$

### D.3 PROOFS FOR SUBSECTION 4.2

#### D.3.1 PROOF OF LEMMA 3

Note that the first $l$ layers (except the output layer) of the original NTK formulation and our new formulation are the same, so we still have the following proposition:

**Proposition 11** (Proposition 1 in Jacot et al. (2018)). *If $\sigma$ is Lipschitz and $d_l \to \infty$ for $l = 1, \cdots, L$ sequentially, then for all $l = 1, \cdots, L$, the distribution of a single element of $\boldsymbol{h}^l$ converges in probability to a zero-mean Gaussian process of covariance $\Sigma^l$ that is defined recursively by:*

$$
\begin{aligned}
\Sigma^1(\boldsymbol{x}, \boldsymbol{x}') &= \frac{1}{d_0} \boldsymbol{x}^\top \boldsymbol{x}' + \beta^2 \\
\Sigma^l(\boldsymbol{x}, \boldsymbol{x}') &= \mathbb{E}_f [\sigma(f(\boldsymbol{x})) \sigma(f(\boldsymbol{x}'))] + \beta^2
\end{aligned}
\tag{41}
$$

*where $f$ is sampled from a zero-mean Gaussian process of covariance $\Sigma^{(l-1)}$.*

Now we show that for an infinitely wide neural network with $L \geq 1$ hidden layers, $\Theta^{(0)}$ converges in probability to the following non-degenerated deterministic limiting kernel

$$
\Theta = \mathbb{E}_{f \sim \Sigma^L} [\sigma(f(\boldsymbol{x})) \sigma(f(\boldsymbol{x}'))] + \beta^2
\tag{42}
$$

Consider the output layer $\boldsymbol{h}^{L+1} = \frac{W^L}{\sqrt{d}} \sigma(\boldsymbol{h}^L) + \beta \boldsymbol{b}^L$. We can see that for any parameter $\theta_i$ before the output layer,

$$
\nabla_{\theta_i} \boldsymbol{h}^{L+1} = \text{diag}(\dot{\sigma}(\boldsymbol{h}^L)) \frac{W^{L\top}}{\sqrt{d_L}} \nabla_{\theta_i} \boldsymbol{h}^L = 0
\tag{43}
$$

And for $W^L$ and $\boldsymbol{b}^L$, we have

$$\nabla_{W^L}\boldsymbol{h}^{L+1} = \frac{1}{\sqrt{d_L}}\sigma(\boldsymbol{h}^L) \qquad \text{and} \qquad \nabla_{\boldsymbol{b}^L}\boldsymbol{h}^{L+1} = \beta \tag{44}$$

Then we can achieve (42) by the law of large numbers. $\qquad\square$

### D.3.2 PROOF OF LEMMA 5

We will use the following short-hand in the proof:

$$\begin{cases} g(\theta^{(t)}) = f^{(t)}(\boldsymbol{X}) - \boldsymbol{Y} \\ J(\theta^{(t)}) = \nabla_\theta f(\boldsymbol{X}; \theta^{(t)}) \in \mathbb{R}^{p \times n} \\ \Theta^{(t)} = J(\theta^{(t)})^\top J(\theta^{(t)}) \end{cases} \tag{45}$$

For any $\epsilon > 0$, there exists $t_\epsilon$ such that for all $t \geq t_\epsilon$ and all $i$, $q_i^{(t)} \in (q_i - \epsilon, q_i + \epsilon)$. Like what we have done in (34), we can rewrite $\boldsymbol{Q}^{(t)} = \boldsymbol{Q}_\epsilon^{(t)} = \sqrt{\boldsymbol{Q}_{3\epsilon}^{(t)}} \cdot \sqrt{\boldsymbol{Q}}$, where $\boldsymbol{Q} = \mathrm{diag}(q_1, \cdots, q_n)$.

The update rule of a GRW with gradient descent and the squared loss for the wide neural network is:

$$\theta^{(t+1)} = \theta^{(t)} - \eta J(\theta^{(t)})\boldsymbol{Q}^{(t)}g(\theta^{(t)}) \tag{46}$$

and for $t \geq t_\epsilon$, it can be rewritten as

$$\theta^{(t+1)} = \theta^{(t)} - \eta J(\theta^{(t)})\sqrt{\boldsymbol{Q}_{3\epsilon}^{(t)}}\left[\sqrt{\boldsymbol{Q}}g(\theta^{(t)})\right] \tag{47}$$

First, we will prove the following theorem:

**Theorem 12.** *There exist constants $M > 0$ and $\epsilon_0 > 0$ such that for all $\epsilon \in (0, \epsilon_0]$, $\eta \leq \eta^*$ and any $\delta > 0$, there exist $R_0 > 0$, $\tilde{D} > 0$ and $B > 1$ such that for any $\tilde{d} \geq \tilde{D}$, the following (i) and (ii) hold with probability at least $(1 - \delta)$ over random initialization when applying gradient descent with learning rate $\eta$:*

*(i) For all $t \leq t_\epsilon$, there is*

$$\left\|g(\theta^{(t)})\right\|_2 \leq B^t R_0 \tag{48}$$

$$\sum_{j=1}^t \left\|\theta^{(j)} - \theta^{(j-1)}\right\|_2 \leq \eta M R_0 \sum_{j=1}^t B^{j-1} < \frac{MB^{t_\epsilon}R_0}{B-1} \tag{49}$$

*(ii) For all $t \geq t_\epsilon$, we have*

$$\left\|\sqrt{\boldsymbol{Q}}g(\theta^{(t)})\right\|_2 \leq \left(1 - \frac{\eta q^*\lambda^{\min}}{3}\right)^{t-t_\epsilon}B^{t_\epsilon}R_0 \tag{50}$$

$$\sum_{j=t_\epsilon+1}^t \left\|\theta^{(j)} - \theta^{(j-1)}\right\|_2 \leq \eta\sqrt{1+3\epsilon}MB^{t_\epsilon}R_0 \sum_{j=t_\epsilon+1}^t \left(1 - \frac{\eta q^*\lambda^{\min}}{3}\right)^{j-t_\epsilon}$$
$$< \frac{3\sqrt{1+3\epsilon}MB^{t_\epsilon}R_0}{q^*\lambda^{\min}} \tag{51}$$

*Proof.* The proof is based on the following lemma:

**Lemma 13** (Local Lipschitzness of the Jacobian). *Under Assumption 2, there is a constant $M > 0$ such that for any $C_0 > 0$ and any $\delta > 0$, there exists a $\tilde{D}$ such that: If $\tilde{d} \geq \tilde{D}$, then with probability*

*at least $(1 - \delta)$ over random initialization, for any $\boldsymbol{x}$ such that $\|\boldsymbol{x}\|_2 \leq 1$,*

$$
\begin{cases}
\left\|\nabla_\theta f(\boldsymbol{x}; \theta) - \nabla_\theta f(\boldsymbol{x}; \tilde{\theta})\right\|_2 \leq \dfrac{M}{\sqrt[4]{\tilde{d}}} \left\|\theta - \tilde{\theta}\right\|_2 \\[2mm]
\|\nabla_\theta f(\boldsymbol{x}; \theta)\|_2 \leq M \\[2mm]
\left\|J(\theta) - J(\tilde{\theta})\right\|_F \leq \dfrac{M}{\sqrt[4]{\tilde{d}}} \left\|\theta - \tilde{\theta}\right\|_2 \\[2mm]
\|J(\theta)\|_F \leq M
\end{cases}, \qquad \forall \theta, \tilde{\theta} \in B(\theta^{(0)}, C_0) \tag{52}
$$

*where $B(\theta^{(0)}, R) = \{\theta : \left\|\theta - \theta^{(0)}\right\|_2 < R\}$.*

The proof of this lemma can be found in Appendix D.3.3. Note that for any $\boldsymbol{x}$, $f^{(0)}(\boldsymbol{x}) = \beta \boldsymbol{b}^L$ where $\boldsymbol{b}^L$ is sampled from the standard Gaussian distribution. Thus, for any $\delta > 0$, there exists a constant $R_0$ such that with probability at least $(1 - \delta/3)$ over random initialization,

$$
\left\|g(\theta^{(0)})\right\|_2 < R_0 \tag{53}
$$

And by Proposition 3, there exists $D_2 \geq 0$ such that for any $\tilde{d} \geq D_2$, with probability at least $(1 - \delta/3)$,

$$
\left\|\Theta - \Theta^{(0)}\right\|_F \leq \frac{q^* \lambda^{\min}}{3} \tag{54}
$$

Let $M$ be the constant in Lemma 13. Let $\epsilon_0 = \frac{(q^* \lambda^{\min})^2}{108 M^4}$. Let $B = 1 + \eta^* M^2$, and $C_0 = \frac{M B^{t_\epsilon} R_0}{B - 1} + \frac{3\sqrt{1 + 3\epsilon} M B^{t_\epsilon} R_0}{q^* \lambda^{\min}}$. By Lemma 13, there exists $D_1 > 0$ such that with probability at least $(1 - \delta/3)$, for any $\tilde{d} \geq D_1$, (52) is true for all $\theta, \tilde{\theta} \in B(\theta^{(0)}, C_0)$.

By union bound, with probability at least $(1 - \delta)$, (52), (53) and (54) are all true. Now we assume that all of them are true, and prove (48) and (49) by induction. (48) is true for $t = 0$ due to (53), and (49) is always true for $t = 0$. Suppose (48) and (49) are true for $t$, then for $t + 1$ we have

$$
\begin{aligned}
\left\|\theta^{(t+1)} - \theta^{(t)}\right\|_2 &\leq \eta \left\|J(\theta^{(t)}) \boldsymbol{Q}^{(t)}\right\|_2 \left\|g(\theta^{(t)})\right\|_2 \leq \eta \left\|J(\theta^{(t)}) \boldsymbol{Q}^{(t)}\right\|_F \left\|g(\theta^{(t)})\right\|_2 \\
&\leq \eta \left\|J(\theta^{(t)})\right\|_F \left\|g(\theta^{(t)})\right\|_2 \leq M \eta B^t R_0
\end{aligned} \tag{55}
$$

So (49) is also true for $t + 1$. And we also have

$$
\begin{aligned}
\left\|g(\theta^{(t+1)})\right\|_2 &= \left\|g(\theta^{(t+1)}) - g(\theta^{(t)}) + g(\theta^{(t)})\right\|_2 \\
&= \left\|J(\tilde{\theta}^{(t)})^\top (\theta^{(t+1)} - \theta^{(t)}) + g(\theta^{(t)})\right\|_2 \\
&= \left\|-\eta J(\tilde{\theta}^{(t)})^\top J(\theta^{(t)}) \boldsymbol{Q}^{(t)} g(\theta^{(t)}) + g(\theta^{(t)})\right\|_2 \\
&\leq \left\|\boldsymbol{I} - \eta J(\tilde{\theta}^{(t)})^\top J(\theta^{(t)}) \boldsymbol{Q}^{(t)}\right\|_2 \left\|g(\theta^{(t)})\right\|_2 \\
&\leq \left(1 + \left\|\eta J(\tilde{\theta}^{(t)})^\top J(\theta^{(t)}) \boldsymbol{Q}^{(t)}\right\|_2\right) \left\|g(\theta^{(t)})\right\|_2 \\
&\leq \left(1 + \eta \left\|J(\tilde{\theta}^{(t)})\right\|_F \left\|J(\theta^{(t)})\right\|_F\right) \left\|g(\theta^{(t)})\right\|_2 \\
&\leq (1 + \eta^* M^2) \left\|g(\theta^{(t)})\right\|_2 \leq B^{t+1} R_0
\end{aligned} \tag{56}
$$

Therefore, (48) and (49) are true for all $t \leq t_\epsilon$, which implies that $\left\|\sqrt{\boldsymbol{Q}} g(\theta^{(t_\epsilon)})\right\|_2 \leq \left\|g(\theta^{(t_\epsilon)})\right\|_2 \leq B^{t_\epsilon} R_0$, so (50) is true for $t = t_\epsilon$. And (51) is obviously true for $t = t_\epsilon$. Now, let us prove (ii) by induction. Note that when $t \geq t_\epsilon$, we have the alternative update rule (47). If (50) and (51) are true for $t$, then for $t + 1$, there is

$$\left\|\theta^{(t+1)} - \theta^{(t)}\right\|_2 \leq \eta \left\|J(\theta^{(t)})\sqrt{\boldsymbol{Q}_{3\epsilon}^{(t)}}\right\|_2 \left\|\sqrt{\boldsymbol{Q}}g(\theta^{(t)})\right\|_2 \leq \eta \left\|J(\theta^{(t)})\sqrt{\boldsymbol{Q}_{3\epsilon}^{(t)}}\right\|_F \left\|\sqrt{\boldsymbol{Q}}g(\theta^{(t)})\right\|_2$$

$$\leq \eta\sqrt{1+3\epsilon}\left\|J(\theta^{(t)})\right\|_F \left\|\sqrt{\boldsymbol{Q}}g(\theta^{(t)})\right\|_2 \leq M\eta\sqrt{1+3\epsilon}\left(1 - \frac{\eta q^* \lambda^{\min}}{3}\right)^{t-t_\epsilon} B^{t_\epsilon} R_0$$

$$(57)$$

So (51) is true for $t+1$. And we also have

$$\left\|\sqrt{\boldsymbol{Q}}g(\theta^{(t+1)})\right\|_2 = \left\|\sqrt{\boldsymbol{Q}}g(\theta^{(t+1)}) - \sqrt{\boldsymbol{Q}}g(\theta^{(t)}) + \sqrt{\boldsymbol{Q}}g(\theta^{(t)})\right\|_2$$

$$= \left\|\sqrt{\boldsymbol{Q}}J(\tilde{\theta}^{(t)})^\top(\theta^{(t+1)} - \theta^{(t)}) + \sqrt{\boldsymbol{Q}}g(\theta^{(t)})\right\|_2$$

$$= \left\|-\eta\sqrt{\boldsymbol{Q}}J(\tilde{\theta}^{(t)})^\top J(\theta^{(t)})\boldsymbol{Q}^{(t)}g(\theta^{(t)}) + \sqrt{\boldsymbol{Q}}g(\theta^{(t)})\right\|_2 \qquad (58)$$

$$\leq \left\|\boldsymbol{I} - \eta\sqrt{\boldsymbol{Q}}J(\tilde{\theta}^{(t)})^\top J(\theta^{(t)})\sqrt{\boldsymbol{Q}_{3\epsilon}^{(t)}}\right\|_2 \left\|\sqrt{\boldsymbol{Q}}g(\theta^{(t)})\right\|_2$$

$$\leq \left\|\boldsymbol{I} - \eta\sqrt{\boldsymbol{Q}}J(\tilde{\theta}^{(t)})^\top J(\theta^{(t)})\sqrt{\boldsymbol{Q}_{3\epsilon}^{(t)}}\right\|_2 \left(1 - \frac{\eta q^* \lambda^{\min}}{3}\right)^t R_0$$

where $\tilde{\theta}^{(t)}$ is some linear interpolation between $\theta^{(t)}$ and $\theta^{(t+1)}$. Now we prove that

$$\left\|\boldsymbol{I} - \eta\sqrt{\boldsymbol{Q}}J(\tilde{\theta}^{(t)})^\top J(\theta^{(t)})\sqrt{\boldsymbol{Q}_{3\epsilon}^{(t)}}\right\|_2 \leq 1 - \frac{\eta q^* \lambda^{\min}}{3} \qquad (59)$$

For any unit vector $\boldsymbol{v} \in \mathbb{R}^n$, we have

$$\boldsymbol{v}^\top(\boldsymbol{I} - \eta\sqrt{\boldsymbol{Q}}\Theta\sqrt{\boldsymbol{Q}})\boldsymbol{v} = 1 - \eta\boldsymbol{v}^\top\sqrt{\boldsymbol{Q}}\Theta\sqrt{\boldsymbol{Q}}\boldsymbol{v} \qquad (60)$$

$\left\|\sqrt{\boldsymbol{Q}}\boldsymbol{v}\right\|_2 \in [\sqrt{q^*}, 1]$, so for any $\eta \leq \eta^*$, $\boldsymbol{v}^\top(\boldsymbol{I} - \eta\sqrt{\boldsymbol{Q}}\Theta\sqrt{\boldsymbol{Q}})\boldsymbol{v} \in [0, 1 - \eta\lambda^{\min}q^*]$, which implies that $\left\|\boldsymbol{I} - \eta\sqrt{\boldsymbol{Q}}\Theta\sqrt{\boldsymbol{Q}}\right\|_2 \leq 1 - \eta\lambda^{\min}q^*$. Thus,

$$\left\|\boldsymbol{I} - \eta\sqrt{\boldsymbol{Q}}J(\tilde{\theta}^{(t)})^\top J(\theta^{(t)})\sqrt{\boldsymbol{Q}}\right\|_2$$

$$\leq \left\|\boldsymbol{I} - \eta\sqrt{\boldsymbol{Q}}\Theta\sqrt{\boldsymbol{Q}}\right\|_2 + \eta\left\|\sqrt{\boldsymbol{Q}}(\Theta - \Theta^{(0)})\sqrt{\boldsymbol{Q}}\right\|_2 + \eta\left\|\sqrt{\boldsymbol{Q}}(J(\theta^{(0)})^\top J(\theta^{(0)}) - J(\tilde{\theta}^{(t)})^\top J(\theta^{(t)}))\sqrt{\boldsymbol{Q}}\right\|_2$$

$$\leq 1 - \eta\lambda^{\min}q^* + \eta\left\|\sqrt{\boldsymbol{Q}}(\Theta - \Theta^{(0)})\sqrt{\boldsymbol{Q}}\right\|_F + \eta\left\|\sqrt{\boldsymbol{Q}}(J(\theta^{(0)})^\top J(\theta^{(0)}) - J(\tilde{\theta}^{(t)})^\top J(\theta^{(t)}))\sqrt{\boldsymbol{Q}}\right\|_F$$

$$\leq 1 - \eta\lambda^{\min}q^* + \eta\left\|\Theta - \Theta^{(0)}\right\|_F + \eta\left\|J(\theta^{(0)})^\top J(\theta^{(0)}) - J(\tilde{\theta}^{(t)})^\top J(\theta^{(t)})\right\|_F$$

$$\leq 1 - \eta\lambda^{\min}q^* + \frac{\eta q^* \lambda^{\min}}{3} + \frac{\eta M^2}{\sqrt[4]{\tilde{d}}}\left(\left\|\theta^{(t)} - \theta^{(0)}\right\|_2 + \left\|\tilde{\theta}^{(t)} - \theta^{(0)}\right\|_2\right) \leq 1 - \frac{\eta q^* \lambda^{\min}}{2}$$

$$(61)$$

for all $\tilde{d} \geq \max\left\{D_1, D_2, \left(\frac{12M^2 C_0}{q^* \lambda^{\min}}\right)^4\right\}$, which implies that

$$\left\|\boldsymbol{I} - \eta\sqrt{\boldsymbol{Q}}J(\tilde{\theta}^{(t)})^\top J(\theta^{(t)})\sqrt{\boldsymbol{Q}_{3\epsilon}^{(t)}}\right\|_2$$

$$\leq 1 - \frac{\eta q^* \lambda^{\min}}{2} + \left\|\eta\sqrt{\boldsymbol{Q}}J(\tilde{\theta}^{(t)})^\top J(\theta^{(t)})\left(\sqrt{\boldsymbol{Q}_{3\epsilon}^{(t)}} - \sqrt{\boldsymbol{Q}}\right)\right\|_2 \qquad (62)$$

$$\leq 1 - \frac{\eta q^* \lambda^{\min}}{2} + \eta M^2\sqrt{3\epsilon} \leq 1 - \frac{\eta q^* \lambda^{\min}}{3} \qquad \text{(due to (37))}$$

for all $\epsilon \leq \epsilon_0$. Thus, (50) is also true for $t+1$. In conclusion, (50) and (51) are true with probability at least $(1 - \delta)$ for all $\tilde{d} \geq \tilde{D} = \max\left\{D_1, D_2, \left(\frac{12M^2 C_0}{q^* \lambda^{\min}}\right)^4\right\}$. $\qquad\square$

Returning back to the proof of Lemma 5. Choose and fix an $\epsilon$ such that $\epsilon <$ $\min\{\epsilon_0, \frac{1}{3}\left(\frac{q^*\lambda^{\min}}{3\lambda^{\max}+q^*\lambda^{\min}}\right)^2\}$, where $\epsilon_0$ is defined by Theorem 12. Then, $t_\epsilon$ is also fixed. There exists $\tilde{D} \geq 0$ such that for any $\tilde{d} \geq \tilde{D}$, with probability at least $(1 - \delta)$, Theorem 12 and Lemma 13 are true and

$$\left\|\Theta - \Theta^{(0)}\right\|_F \leq \frac{q^*\lambda^{\min}}{3} \tag{63}$$

which immediately implies that

$$\left\|\Theta^{(0)}\right\|_2 \leq \|\Theta\|_2 + \left\|\Theta - \Theta^{(0)}\right\|_F \leq \lambda^{\max} + \frac{q^*\lambda^{\min}}{3} \tag{64}$$

We still denote $B = 1 + \eta^* M^2$ and $C_0 = \frac{MB^{t_\epsilon}R_0}{B-1} + \frac{3\sqrt{1+3\epsilon}MB^{t_\epsilon}R_0}{q^*\lambda^{\min}}$. Theorem 12 ensures that for all $t$, $\theta^{(t)} \in B(\theta^{(0)}, C_0)$. Then we have

$$\begin{aligned}
\left\|\boldsymbol{I} - \eta\sqrt{\boldsymbol{Q}}\Theta^{(0)}\sqrt{\boldsymbol{Q}}\right\|_2 &\leq \left\|\boldsymbol{I} - \eta\sqrt{\boldsymbol{Q}}\Theta\sqrt{\boldsymbol{Q}}\right\|_2 + \eta\left\|\sqrt{\boldsymbol{Q}}(\Theta - \Theta^{(0)})\sqrt{\boldsymbol{Q}}\right\|_2 \\
&\leq 1 - \eta\lambda^{\min}q^* + \frac{\eta q^*\lambda^{\min}}{3} = 1 - \frac{2\eta q^*\lambda^{\min}}{3}
\end{aligned} \tag{65}$$

so it follows that

$$\begin{aligned}
\left\|\boldsymbol{I} - \eta\sqrt{\boldsymbol{Q}}\Theta^{(0)}\sqrt{\boldsymbol{Q}_{3\epsilon}^{(t)}}\right\|_2 &\leq \left\|\boldsymbol{I} - \eta\sqrt{\boldsymbol{Q}}\Theta^{(0)}\sqrt{\boldsymbol{Q}}\right\|_2 + \left\|\eta\sqrt{\boldsymbol{Q}}\Theta^{(0)}\left(\sqrt{\boldsymbol{Q}_{3\epsilon}^{(t)}} - \sqrt{\boldsymbol{Q}}\right)\right\|_2 \\
&\leq 1 - \frac{2\eta q^*\lambda^{\min}}{3} + \eta(\lambda^{\max} + \frac{q^*\lambda^{\min}}{3})\sqrt{3\epsilon}
\end{aligned} \tag{66}$$

Thus, for all $\epsilon < \frac{1}{3}\left(\frac{q^*\lambda^{\min}}{3\lambda^{\max}+q^*\lambda^{\min}}\right)^2$, there is

$$\left\|\boldsymbol{I} - \eta\sqrt{\boldsymbol{Q}}\Theta^{(0)}\sqrt{\boldsymbol{Q}_{3\epsilon}^{(t)}}\right\|_2 \leq 1 - \frac{\eta q^*\lambda^{\min}}{3} \tag{67}$$

The update rule of the GRW for the linearized neural network is:

$$\theta_{\text{lin}}^{(t+1)} = \theta_{\text{lin}}^{(t)} - \eta J(\theta^{(0)})\boldsymbol{Q}^{(t)}g_{\text{lin}}(\theta^{(t)}) \tag{68}$$

where we use the subscript "lin" to denote the linearized neural network, and with a slight abuse of notion denote $g_{\text{lin}}(\theta^{(t)}) = g(\theta_{\text{lin}}^{(t)})$.

First, let us consider the training data $\boldsymbol{X}$. Denote $\Delta_t = g_{\text{lin}}(\theta^{(t)}) - g(\theta^{(t)})$. We have

$$\begin{cases}
g_{\text{lin}}(\theta^{(t+1)}) - g_{\text{lin}}(\theta^{(t)}) = -\eta J(\theta^{(0)})^\top J(\theta^{(0)})\boldsymbol{Q}^{(t)}g_{\text{lin}}(\theta^{(t)}) \\
g(\theta^{(t+1)}) - g(\theta^{(t)}) = -\eta J(\tilde{\theta}^{(t)})^\top J(\theta^{(t)})\boldsymbol{Q}^{(t)}g(\theta^{(t)})
\end{cases} \tag{69}$$

where $\tilde{\theta}^{(t)}$ is some linear interpolation between $\theta^{(t)}$ and $\theta^{(t+1)}$. Thus,

$$\begin{aligned}
\Delta_{t+1} - \Delta_t = &\eta\left[J(\tilde{\theta}^{(t)})^\top J(\theta^{(t)}) - J(\theta^{(0)})^\top J(\theta^{(0)})\right]\boldsymbol{Q}^{(t)}g(\theta^{(t)}) \\
&- \eta J(\theta^{(0)})^\top J(\theta^{(0)})\boldsymbol{Q}^{(t)}\Delta_t
\end{aligned} \tag{70}$$

By Lemma 13, we have

$$\begin{aligned}
&\left\|J(\tilde{\theta}^{(t)})^\top J(\theta^{(t)}) - J(\theta^{(0)})^\top J(\theta^{(0)})\right\|_F \\
&\leq \left\|\left(J(\tilde{\theta}^{(t)}) - J(\theta^{(0)})\right)^\top J(\theta^{(t)})\right\|_F + \left\|J(\theta^{(0)})^\top \left(J(\theta^{(t)}) - J(\theta^{(0)})\right)\right\|_F \\
&\leq 2M^2 C_0\tilde{d}^{-1/4}
\end{aligned} \tag{71}$$

which implies that for all $t < t_\epsilon$,

$$
\begin{aligned}
\|\Delta_{t+1}\|_2 &\le \left\| \left[ \boldsymbol{I} - \eta J(\theta^{(0)})^\top J(\theta^{(0)}) \boldsymbol{Q}^{(t)} \right] \Delta_t \right\|_2 + \left\| \eta \left[ J(\tilde{\theta}^{(t)})^\top J(\theta^{(t)}) - J(\theta^{(0)})^\top J(\theta^{(0)}) \right] \boldsymbol{Q}^{(t)} g(\theta^{(t)}) \right\|_2 \\
&\le \left\| \boldsymbol{I} - \eta J(\theta^{(0)})^\top J(\theta^{(0)}) \boldsymbol{Q}^{(t)} \right\|_F \|\Delta_t\|_2 + \eta \left\| J(\tilde{\theta}^{(t)})^\top J(\theta^{(t)}) - J(\theta^{(0)})^\top J(\theta^{(0)}) \right\|_F \left\| g(\theta^{(t)}) \right\|_2 \\
&\le (1 + \eta M^2) \|\Delta_t\|_2 + 2\eta M^2 C_0 B^t R_0 \tilde{d}^{-1/4} \\
&\le B \|\Delta_t\|_2 + 2\eta M^2 C_0 B^t R_0 \tilde{d}^{-1/4}
\end{aligned}
\tag{72}
$$

Therefore, we have

$$
B^{-(t+1)} \|\Delta_{t+1}\|_2 \le B^{-t} \|\Delta_t\|_2 + 2\eta M^2 C_0 B^{-1} R_0 \tilde{d}^{-1/4}
\tag{73}
$$

Since $\Delta_0 = 0$, it follows that for all $t \le t_\epsilon$,

$$
\|\Delta_t\|_2 \le 2t\eta M^2 C_0 B^{t-1} R_0 \tilde{d}^{-1/4}
\tag{74}
$$

and particularly we have

$$
\left\| \sqrt{\boldsymbol{Q}} \Delta_{t_\epsilon} \right\|_2 \le \|\Delta_{t_\epsilon}\|_2 \le 2t_\epsilon \eta M^2 C_0 B^{t_\epsilon - 1} R_0 \tilde{d}^{-1/4}
\tag{75}
$$

For $t \ge t_\epsilon$, we have the alternative update rule (47). Thus,

$$
\begin{aligned}
\sqrt{\boldsymbol{Q}} \Delta_{t+1} - \sqrt{\boldsymbol{Q}} \Delta_t =&\eta \sqrt{\boldsymbol{Q}} \left[ J(\tilde{\theta}^{(t)})^\top J(\theta^{(t)}) - J(\theta^{(0)})^\top J(\theta^{(0)}) \right] \sqrt{\boldsymbol{Q}_{3\epsilon}^{(t)}} \left[ \sqrt{\boldsymbol{Q}} g(\theta^{(t)}) \right] \\
&- \eta \sqrt{\boldsymbol{Q}} J(\theta^{(0)})^\top J(\theta^{(0)}) \sqrt{\boldsymbol{Q}_{3\epsilon}^{(t)}} \left[ \sqrt{\boldsymbol{Q}} \Delta_t \right]
\end{aligned}
\tag{76}
$$

Let $\boldsymbol{A} = \boldsymbol{I} - \eta \sqrt{\boldsymbol{Q}} J(\theta^{(0)})^\top J(\theta^{(0)}) \sqrt{\boldsymbol{Q}_{3\epsilon}^{(t)}} = \boldsymbol{I} - \eta \sqrt{\boldsymbol{Q}} \Theta^{(0)} \sqrt{\boldsymbol{Q}_{3\epsilon}^{(t)}}$. Then, we have

$$
\sqrt{\boldsymbol{Q}} \Delta_{t+1} = \boldsymbol{A} \sqrt{\boldsymbol{Q}} \Delta_t + \eta \sqrt{\boldsymbol{Q}} \left[ J(\tilde{\theta}^{(t)})^\top J(\theta^{(t)}) - J(\theta^{(0)})^\top J(\theta^{(0)}) \right] \sqrt{\boldsymbol{Q}_{3\epsilon}^{(t)}} \left( \sqrt{\boldsymbol{Q}} g(\theta^{(t)}) \right)
\tag{77}
$$

Let $\gamma = 1 - \frac{\eta q^* \lambda^{\min}}{3} < 1$. Combining with Theorem 12 and (67), the above leads to

$$
\begin{aligned}
\left\| \sqrt{\boldsymbol{Q}} \Delta_{t+1} \right\|_2 &\le \|\boldsymbol{A}\|_2 \left\| \sqrt{\boldsymbol{Q}} \Delta_t \right\|_2 + \eta \left\| \sqrt{\boldsymbol{Q}} \left[ J(\tilde{\theta}^{(t)})^\top J(\theta^{(t)}) - J(\theta^{(0)})^\top J(\theta^{(0)}) \right] \sqrt{\boldsymbol{Q}_{3\epsilon}^{(t)}} \right\|_2 \left\| \sqrt{\boldsymbol{Q}} g(\theta^{(t)}) \right\|_2 \\
&\le \gamma \left\| \sqrt{\boldsymbol{Q}} \Delta_t \right\|_2 + \eta \left\| J(\tilde{\theta}^{(t)})^\top J(\theta^{(t)}) - J(\theta^{(0)})^\top J(\theta^{(0)}) \right\|_F \sqrt{1 + 3\epsilon} \gamma^{t - t_\epsilon} B^{t_\epsilon} R_0 \\
&\le \gamma \left\| \sqrt{\boldsymbol{Q}} \Delta_t \right\|_2 + 2\eta M^2 C_0 \sqrt{1 + 3\epsilon} \gamma^{t - t_\epsilon} B^{t_\epsilon} R_0 \tilde{d}^{-1/4}
\end{aligned}
\tag{78}
$$

This implies that

$$
\gamma^{-(t+1)} \left\| \sqrt{\boldsymbol{Q}} \Delta_{t+1} \right\|_2 \le \gamma^{-t} \left\| \sqrt{\boldsymbol{Q}} \Delta_t \right\|_2 + 2\eta M^2 C_0 \sqrt{1 + 3\epsilon} \gamma^{-1 - t_\epsilon} B^{t_\epsilon} R_0 \tilde{d}^{-1/4}
\tag{79}
$$

Combining with (75), it implies that for all $t \ge t_\epsilon$,

$$
\left\| \sqrt{\boldsymbol{Q}} \Delta_t \right\|_2 \le 2\gamma^{t - t_\epsilon} \eta M^2 C_0 B^{t_\epsilon} R_0 \left[ t_\epsilon B^{-1} + \sqrt{1 + 3\epsilon} \gamma^{-1} (t - t_\epsilon) \right] \tilde{d}^{-1/4}
\tag{80}
$$

Next, we consider an arbitrary test point $\boldsymbol{x}$ such that $\|\boldsymbol{x}\|_2 \le 1$. Denote $\delta_t = f_{\text{lin}}^{(t)}(\boldsymbol{x}) - f^{(t)}(\boldsymbol{x})$. Then we have

$$
\begin{cases}
f_{\text{lin}}^{(t+1)}(\boldsymbol{x}) - f_{\text{lin}}^{(t)}(\boldsymbol{x}) = -\eta \nabla_\theta f(\boldsymbol{x}; \theta^{(0)})^\top J(\theta^{(0)}) \boldsymbol{Q}^{(t)} g_{\text{lin}}(\theta^{(t)}) \\
f^{(t+1)}(\boldsymbol{x}) - f^{(t)}(\boldsymbol{x}) = -\eta \nabla_\theta f(\boldsymbol{x}; \tilde{\theta}^{(t)})^\top J(\theta^{(t)}) \boldsymbol{Q}^{(t)} g(\theta^{(t)})
\end{cases}
\tag{81}
$$

which yields

$$
\begin{aligned}
\delta_{t+1} - \delta_t =&\eta \left[ \nabla_\theta f(\boldsymbol{x}; \tilde{\theta}^{(t)})^\top J(\theta^{(t)}) - \nabla_\theta f(\boldsymbol{x}; \theta^{(0)})^\top J(\theta^{(0)}) \right] \boldsymbol{Q}^{(t)} g(\theta^{(t)}) \\
&- \eta \nabla_\theta f(\boldsymbol{x}; \theta^{(0)})^\top J(\theta^{(0)}) \boldsymbol{Q}^{(t)} \Delta_t
\end{aligned}
\tag{82}
$$

For $t \leq t_\epsilon$, we have

$$
\begin{aligned}
\|\delta_t\|_2 \leq & \eta \sum_{s=0}^{t-1} \left\| \left[ \nabla_\theta f(\boldsymbol{x}; \tilde{\theta}^{(s)})^\top J(\theta^{(s)}) - \nabla_\theta f(\boldsymbol{x}; \theta^{(0)})^\top J(\theta^{(0)}) \right] \boldsymbol{Q}^{(s)} \right\|_2 \left\| g(\theta^{(s)}) \right\|_2 \\
& + \eta \sum_{s=0}^{t-1} \left\| \nabla_\theta f(\boldsymbol{x}; \theta^{(0)})^\top J(\theta^{(0)}) \boldsymbol{Q}^{(s)} \right\|_2 \|\Delta_s\|_2 \\
\leq & \eta \sum_{s=0}^{t-1} \left\| \nabla_\theta f(\boldsymbol{x}; \tilde{\theta}^{(s)})^\top J(\theta^{(s)}) - \nabla_\theta f(\boldsymbol{x}; \theta^{(0)})^\top J(\theta^{(0)}) \right\|_F \left\| g(\theta^{(s)}) \right\|_2 \qquad (83) \\
& + \eta \sum_{s=0}^{t-1} \left\| \nabla_\theta f(\boldsymbol{x}; \theta^{(0)}) \right\|_2 \left\| J(\theta^{(0)}) \right\|_F \|\Delta_s\|_2 \\
\leq & 2\eta M^2 C_0 \tilde{d}^{-1/4} \sum_{s=0}^{t-1} B^s R_0 + \eta M^2 \sum_{s=0}^{t-1} (2s\eta M^2 C_0 B^{s-1} R_0 \tilde{d}^{-1/4})
\end{aligned}
$$

So we can see that there exists a constant $C_1$ such that $\|\delta_{t_\epsilon}\|_2 \leq C_1 \tilde{d}^{-1/4}$. Then, for $t > t_\epsilon$, we have

$$
\begin{aligned}
\|\delta_t\|_2 - \|\delta_{t_\epsilon}\|_2 \leq & \eta \sum_{s=t_\epsilon}^{t-1} \left\| \left[ \nabla_\theta f(\boldsymbol{x}; \tilde{\theta}^{(s)})^\top J(\theta^{(s)}) - \nabla_\theta f(\boldsymbol{x}; \theta^{(0)})^\top J(\theta^{(0)}) \right] \sqrt{\boldsymbol{Q}_{3\epsilon}^{(s)}} \right\|_2 \left\| \sqrt{\boldsymbol{Q}} g(\theta^{(s)}) \right\|_2 \\
& + \eta \sum_{s=t_\epsilon}^{t-1} \left\| \nabla_\theta f(\boldsymbol{x}; \theta^{(0)})^\top J(\theta^{(0)}) \sqrt{\boldsymbol{Q}_{3\epsilon}^{(s)}} \right\|_2 \left\| \sqrt{\boldsymbol{Q}} \Delta_s \right\|_2 \\
\leq & 2\eta M^2 C_0 \tilde{d}^{-1/4} \sqrt{1+3\epsilon} \sum_{s=t_\epsilon}^{t-1} \gamma^{s-t_\epsilon} B^{t_\epsilon} R_0 \\
& + \eta M^2 \sqrt{1+3\epsilon} \sum_{s=t_\epsilon}^{t-1} \left( 2\gamma^{s-t_\epsilon} \eta M^2 C_0 B^{t_\epsilon} R_0 \left[ t_\epsilon B^{-1} + \sqrt{1+3\epsilon} \gamma^{-1}(s-t_\epsilon) \right] \tilde{d}^{-1/4} \right)
\end{aligned}
$$

$$(84)$$

Note that $\sum_{t=0}^\infty t\gamma^t$ is finite as long as $\gamma \in (0,1)$. Therefore, there is a constant $C$ such that for any $t$, $\|\delta_t\|_2 \leq C\tilde{d}^{-1/4}$ with probability at least $(1-\delta)$ for any $\tilde{d} \geq \tilde{D}$. $\qquad \square$

### D.3.3    PROOF OF LEMMA 13

We will use the following theorem regarding the eigenvalues of random Gaussian matrices:

**Theorem 14** (Corollary 5.35 in Vershynin (2010))**.** *If* $\boldsymbol{A} \in \mathbb{R}^{p \times q}$ *is a random matrix whose entries are independent standard normal random variables, then for every* $t \geq 0$*, with probability at least* $1 - 2\exp(-t^2/2)$*,*

$$\sqrt{p} - \sqrt{q} - t \leq \lambda^{\min}(\boldsymbol{A}) \leq \lambda^{\max}(\boldsymbol{A}) \leq \sqrt{p} + \sqrt{q} + t \qquad (85)$$

By this theorem, and also note that $W^L$ is a vector, we can see that for any $\delta$, there exist $\tilde{D} > 0$ and $M_1 > 0$ such that if $\tilde{d} \geq \tilde{D}$, then with probability at least $(1-\delta)$, for all $\theta \in B(\theta^{(0)}, C_0)$, we have

$$\left\| W^l \right\|_2 \leq 3\sqrt{\tilde{d}} \quad (\forall 0 \leq l \leq L-1) \qquad \text{and} \qquad \left\| W^L \right\|_2 \leq C_0 \leq 3\sqrt[4]{\tilde{d}} \qquad (86)$$

as well as

$$\left\| \beta \boldsymbol{b}^l \right\|_2 \leq M_1 \sqrt{\tilde{d}} \qquad (\forall l = 0, \cdots, L) \qquad (87)$$

Now we assume that (86) and (87) are true. Then, for any $x$ such that $\|x\|_2 \le 1$,

$$
\begin{aligned}
\left\|h^1\right\|_2 &= \left\|\frac{1}{\sqrt{d_0}}W^0 x + \beta b^0\right\|_2 \le \frac{1}{\sqrt{d_0}}\left\|W^0\right\|_2 \|x\|_2 + \left\|\beta b^0\right\|_2 \le (\frac{3}{\sqrt{d_0}} + M_1)\sqrt{\tilde{d}} \\
\left\|h^{l+1}\right\|_2 &= \left\|\frac{1}{\sqrt{\tilde{d}}}W^l x^l + \beta b^l\right\|_2 \le \frac{1}{\sqrt{\tilde{d}}}\left\|W^l\right\|_2 \left\|x^l\right\|_2 + \left\|\beta b^l\right\|_2 \qquad (\forall l \ge 1) \\
\left\|x^l\right\|_2 &= \left\|\sigma(h^l) - \sigma(0^l) + \sigma(0^l)\right\|_2 \le L_0 \left\|h^l\right\|_2 + \sigma(0)\sqrt{\tilde{d}} \qquad (\forall l \ge 1)
\end{aligned}
\tag{88}
$$

where $L_0$ is the Lipschitz constant of $\sigma$ and $\sigma(0^l) = (\sigma(0), \cdots, \sigma(0)) \in \mathbb{R}^{d_l}$. By induction, there exists an $M_2 > 0$ such that $\left\|x^l\right\|_2 \le M_2\sqrt{\tilde{d}}$ and $\left\|h^l\right\|_2 \le M_2\sqrt{\tilde{d}}$ for all $l = 1, \cdots, L$.

Denote $\alpha^l = \nabla_{h^l} f(x) = \nabla_{h^l} h^{L+1}$. For all $l = 1, \cdots, L$, we have $\alpha^l = \operatorname{diag}(\dot\sigma(h^l))\frac{W^{l\top}}{\sqrt{\tilde{d}}}\alpha^{l+1}$ where $\dot\sigma(x) \le L_0$ for all $x \in \mathbb{R}$ since $\sigma$ is $L_0$-Lipschitz, $\alpha^{L+1} = 1$ and $\left\|\alpha^L\right\|_2 = \left\|\operatorname{diag}(\dot\sigma(h^L))\frac{W^{L\top}}{\sqrt{\tilde{d}}}\right\|_2 \le \frac{3}{\sqrt[4]{\tilde{d}}}L_0$. Then, we can easily prove by induction that there exists an $M_3 > 1$ such that $\left\|\alpha^l\right\|_2 \le M_3/\sqrt[4]{\tilde{d}}$ for all $l = 1, \cdots, L$ (note that this is not true for $L+1$ because $\alpha^{L+1} = 1$).

For $l = 0$, $\nabla_{W^0} f(x) = \frac{1}{\sqrt{d_0}}x^0\alpha^{1\top}$, so $\|\nabla_{W^l} f(x)\|_2 \le \frac{1}{\sqrt{d_0}}\left\|x^0\right\|_2 \left\|\alpha^1\right\|_2 \le \frac{1}{\sqrt{d_0}}M_3/\sqrt[4]{\tilde{d}}$. And for any $l = 1, \cdots, L$, $\nabla_{W^l} f(x) = \frac{1}{\sqrt{\tilde{d}}}x^l\alpha^{l+1}$, so $\|\nabla_{W^l} f(x)\|_2 \le \frac{1}{\sqrt{\tilde{d}}}\left\|x^l\right\|_2 \left\|\alpha^{l+1}\right\|_2 \le M_2 M_3$. (Note that if $M_3 > 1$, then $\left\|\alpha^{L+1}\right\|_2 \le M_3$; and since $\tilde{d} \ge 1$, there is $\left\|\alpha^l\right\|_2 \le M_3$ for $l \le L$.) Moreover, for $l = 0, \cdots, L$, $\nabla_{b^l} f(x) = \beta\alpha^{l+1}$, so $\|\nabla_{b^l} f(x)\|_2 \le \beta M_3$. Thus, if (86) and (87) are true, then there exists an $M_4 > 0$, such that $\|\nabla_\theta f(x)\|_2 \le M_4/\sqrt{n}$. And since $\|x_i\|_2 \le 1$ for all $i$, so $\|J(\theta)\|_F \le M_4$.

Next, we consider the difference in $\nabla_\theta f(x)$ between $\theta$ and $\tilde\theta$. Let $\tilde{f}, \tilde{W}, \tilde{b}, \tilde{x}, \tilde{h}, \tilde{\alpha}$ be the function and the values corresponding to $\tilde\theta$. There is

$$
\begin{aligned}
\left\|h^1 - \tilde{h}^1\right\|_2 &= \left\|\frac{1}{\sqrt{d_0}}(W^0 - \tilde{W}^0)x + \beta(b^0 - \tilde{b}^0)\right\|_2 \\
&\le \frac{1}{\sqrt{d_0}}\left\|W^0 - \tilde{W}^0\right\|_2 \|x\|_2 + \beta\left\|b^0 - \tilde{b}^0\right\|_2 \le \left(\frac{1}{\sqrt{d_0}} + \beta\right)\left\|\theta - \tilde\theta\right\|_2 \\
\left\|h^{l+1} - \tilde{h}^{l+1}\right\|_2 &= \left\|\frac{1}{\sqrt{\tilde{d}}}W^l(x^l - \tilde{x}^l) + \frac{1}{\sqrt{\tilde{d}}}(W^l - \tilde{W}^l)\tilde{x}^l + \beta(b^l - \tilde{b}^l)\right\|_2 \\
&\le \frac{1}{\sqrt{\tilde{d}}}\left\|W^l\right\|_2 \left\|x^l - \tilde{x}^l\right\|_2 + \frac{1}{\sqrt{\tilde{d}}}\left\|W^l - \tilde{W}^l\right\|_2 \left\|\tilde{x}^l\right\|_2 + \beta\left\|b^l - \tilde{b}^l\right\|_2 \\
&\le 3\left\|x^l - \tilde{x}^l\right\|_2 + (M_2 + \beta)\left\|\theta - \tilde\theta\right\|_2 \qquad (\forall l \ge 1) \\
\left\|x^l - \tilde{x}^l\right\|_2 &= \left\|\sigma(h^l) - \sigma(\tilde{h}^l)\right\|_2 \le L_0\left\|h^l - \tilde{h}^l\right\|_2 \qquad (\forall l \ge 1)
\end{aligned}
\tag{89}
$$

By induction, there exists an $M_5 > 0$ such that $\left\|x^l - \tilde{x}^l\right\|_2 \le M_5\left\|\theta - \tilde\theta\right\|_2$ for all $l$.

For $\boldsymbol{\alpha}^l$, we have $\boldsymbol{\alpha}^{L+1} = \tilde{\boldsymbol{\alpha}}^{L+1} = 1$, and for all $l \geq 1$,

$$\left\| \boldsymbol{\alpha}^l - \tilde{\boldsymbol{\alpha}}^l \right\|_2 = \left\| \mathrm{diag}(\dot{\sigma}(\boldsymbol{h}^l)) \frac{W^{l\top}}{\sqrt{\tilde{d}}} \boldsymbol{\alpha}^{l+1} - \mathrm{diag}(\dot{\sigma}(\tilde{\boldsymbol{h}}^l)) \frac{\tilde{W}^{l\top}}{\sqrt{\tilde{d}}} \tilde{\boldsymbol{\alpha}}^{l+1} \right\|_2$$

$$\leq \left\| \mathrm{diag}(\dot{\sigma}(\boldsymbol{h}^l)) \frac{W^{l\top}}{\sqrt{\tilde{d}}} (\boldsymbol{\alpha}^{l+1} - \tilde{\boldsymbol{\alpha}}^{l+1}) \right\|_2 + \left\| \mathrm{diag}(\dot{\sigma}(\boldsymbol{h}^l)) \frac{(W^l - \tilde{W}^l)^\top}{\sqrt{\tilde{d}}} \tilde{\boldsymbol{\alpha}}^{l+1} \right\|_2$$

$$+ \left\| \mathrm{diag}((\dot{\sigma}(\boldsymbol{h}^l) - \dot{\sigma}(\tilde{\boldsymbol{h}}^l))) \frac{\tilde{W}^{l\top}}{\sqrt{\tilde{d}}} \tilde{\boldsymbol{\alpha}}^{l+1} \right\|_2$$

$$\leq 3L_0 \left\| \boldsymbol{\alpha}^{l+1} - \tilde{\boldsymbol{\alpha}}^{l+1} \right\|_2 + \left( M_3 L_0 \tilde{d}^{-1/2} + 3M_3 M_5 L_1 \tilde{d}^{-1/4} \right) \left\| \theta - \tilde{\theta} \right\|_2$$

$$(90)$$

where $L_1$ is the Lipschitz constant of $\dot{\sigma}$. Particularly, for $l = L$, though $\tilde{\boldsymbol{\alpha}}^{L+1} = 1$, since $\left\| \tilde{W}^L \right\|_2 \leq 3\tilde{d}^{1/4}$, (90) is still true. By induction, there exists an $M_6 > 0$ such that $\left\| \boldsymbol{\alpha}^l - \tilde{\boldsymbol{\alpha}}^l \right\|_2 \leq \frac{M_6}{\sqrt[4]{\tilde{d}}} \left\| \theta - \tilde{\theta} \right\|_2$ for all $l \geq 1$ (note that this is also true for $l = L + 1$).

Thus, if (86) and (87) are true, then for all $\theta, \tilde{\theta} \in B(\theta^{(0)}, C_0)$, any $\boldsymbol{x}$ such that $\|\boldsymbol{x}\|_2 \leq 1$, we have

$$\left\| \nabla_{W^0} f(\boldsymbol{x}) - \nabla_{\tilde{W}^0} \tilde{f}(\boldsymbol{x}) \right\|_2 = \frac{1}{\sqrt{d_0}} \left\| \boldsymbol{x} \boldsymbol{\alpha}^{1\top} - \boldsymbol{x} \tilde{\boldsymbol{\alpha}}^{1\top} \right\|_2$$

$$\leq \frac{1}{\sqrt{d_0}} \left\| \boldsymbol{\alpha}^1 - \tilde{\boldsymbol{\alpha}}^1 \right\|_2 \qquad (91)$$

$$\leq \frac{1}{\sqrt{d_0}} \frac{M_6}{\sqrt[4]{\tilde{d}}} \left\| \theta - \tilde{\theta} \right\|_2$$

and for $l = 1, \cdots, L$, we have

$$\left\| \nabla_{W^l} f(\boldsymbol{x}) - \nabla_{\tilde{W}^l} \tilde{f}(\boldsymbol{x}) \right\|_2 = \frac{1}{\sqrt{\tilde{d}}} \left\| \boldsymbol{x}^l \boldsymbol{\alpha}^{l+1\top} - \tilde{\boldsymbol{x}}^l \tilde{\boldsymbol{\alpha}}^{l+1\top} \right\|_2$$

$$\leq \frac{1}{\sqrt{\tilde{d}}} \left( \left\| \boldsymbol{x}^l \right\|_2 \left\| \boldsymbol{\alpha}^{l+1} - \tilde{\boldsymbol{\alpha}}^{l+1} \right\|_2 + \left\| \boldsymbol{x}^l - \tilde{\boldsymbol{x}}^l \right\|_2 \left\| \tilde{\boldsymbol{\alpha}}^{l+1} \right\|_2 \right) \qquad (92)$$

$$\leq \left( \frac{M_2 M_6}{\sqrt[4]{\tilde{d}}} + \frac{M_5 M_3}{\sqrt{\tilde{d}}} \right) \left\| \theta - \tilde{\theta} \right\|_2$$

Moreover, for any $l = 0, \cdots, L$, there is

$$\left\| \nabla_{b^l} f(\boldsymbol{x}) - \nabla_{\tilde{b}^l} \tilde{f}(\boldsymbol{x}) \right\|_2 = \beta \left\| \boldsymbol{\alpha}^{l+1} - \tilde{\boldsymbol{\alpha}}^{l+1} \right\|_2 \leq \frac{\beta M_6}{\sqrt[4]{\tilde{d}}} \left\| \theta - \tilde{\theta} \right\|_2 \qquad (93)$$

Overall, we can see that there exists a constant $M_7 > 0$ such that $\left\| \nabla_\theta f(\boldsymbol{x}) - \nabla_{\tilde{\theta}} \tilde{f}(\boldsymbol{x}) \right\|_2 \leq \frac{M_7}{\sqrt{n} \cdot \sqrt[4]{\tilde{d}}} \left\| \theta - \tilde{\theta} \right\|_2$, so that $\left\| J(\theta) - J(\tilde{\theta}) \right\|_F \leq \frac{M_7}{\sqrt[4]{\tilde{d}}} \left\| \theta - \tilde{\theta} \right\|_2$. $\qquad \square$

### D.3.4 PROOF OF THEOREM 4

First of all, for a linearized neural network (11), if we view $\{\nabla_\theta f^{(0)}(\boldsymbol{x}_i)\}_{i=1}^n$ as the inputs and $\{y_i - f^{(0)}(\boldsymbol{x}_i) + \langle \theta^{(0)}, \nabla_\theta f^{(0)}(\boldsymbol{x}_i) \rangle\}_{i=1}^n$ as the targets, then the model becomes a linear model. So by Theorem 2 we have the following corollary:

**Corollary 15.** *If $\nabla_\theta f^{(0)}(\boldsymbol{x}_1), \cdots, \nabla_\theta f^{(0)}(\boldsymbol{x}_n)$ are linearly independent, then there exists $\eta_0 > 0$ such that for any GRW satisfying Assumption 1, and any $\eta \leq \eta_0$, $\theta^{(t)}$ converges to the same interpolator $\theta^*$ that does not depend on $q_i$.*

Let $\eta_1 = \min\{\eta_0, \eta^*\}$, where $\eta_0$ is defined in Corollary 15 and $\eta^*$ is defined in Lemma 5. Let $f_{\text{lin}}^{(t)}(\boldsymbol{x})$ and $f_{\text{linERM}}^{(t)}(\boldsymbol{x})$ be the linearized neural networks of $f^{(t)}(\boldsymbol{x})$ and $f_{\text{ERM}}^{(t)}(\boldsymbol{x})$, respectively. By Lemma 5, for any $\delta > 0$, there exists $\tilde{D} > 0$ and a constant $C$ such that

$$
\begin{cases}
\sup_{t \geq 0} \left| f_{\text{lin}}^{(t)}(\boldsymbol{x}) - f^{(t)}(\boldsymbol{x}) \right| \leq C \tilde{d}^{-1/4} \\
\sup_{t \geq 0} \left| f_{\text{linERM}}^{(t)}(\boldsymbol{x}) - f_{\text{ERM}}^{(t)}(\boldsymbol{x}) \right| \leq C \tilde{d}^{-1/4}
\end{cases}
\tag{94}
$$

By Corollary 15, we have

$$
\lim_{t \to \infty} \left| f_{\text{lin}}^{(t)}(\boldsymbol{x}) - f_{\text{linERM}}^{(t)}(\boldsymbol{x}) \right| = 0
\tag{95}
$$

Summing the above yields

$$
\limsup_{t \to \infty} \left| f^{(t)}(\boldsymbol{x}) - f_{\text{ERM}}^{(t)}(\boldsymbol{x}) \right| \leq 2C \tilde{d}^{-1/4}
\tag{96}
$$

which is the result we want. $\qquad\square$

### D.4 PROOFS FOR SUBSECTION 4.3

#### D.4.1 A NEW APPROXIMATION THEOREM

**Lemma 16** (Approximation Theorem for Regularized GRW). *For a wide fully-connected neural network $f$, denote $J(\theta) = \nabla_\theta f(\boldsymbol{X}; \theta) \in \mathbb{R}^{p \times n}$ and $g(\theta) = \nabla_{\hat{y}} \ell(f(\boldsymbol{X}; \theta), \boldsymbol{Y}) \in \mathbb{R}^n$. Given that the loss function $\ell$ satisfies: $\nabla_\theta g(\theta) = J(\theta) U(\theta)$ for any $\theta$, and $U(\theta)$ is a positive semi-definite diagonal matrix whose elements are uniformly bounded, we have: for any GRW that minimizes the regularized weighted empirical risk (13) with a sufficiently small learning rate $\eta$, there is: for a sufficiently large $\tilde{d}$, with high probability over random initialization, on any test point $\boldsymbol{x}$ such that $\|\boldsymbol{x}\|_2 \leq 1$,*

$$
\sup_{t \geq 0} \left| f_{\text{linreg}}^{(t)}(\boldsymbol{x}) - f_{\text{reg}}^{(t)}(\boldsymbol{x}) \right| \leq C \tilde{d}^{-1/4}
\tag{97}
$$

*where both $f_{\text{linreg}}^{(t)}$ and $f_{\text{reg}}^{(t)}$ are trained by the same regularized GRW and start from the same initial point.*

First of all, with some simple linear algebra analysis, we can prove the following proposition:

**Proposition 17.** *For any positive definite symmetric matrix $\boldsymbol{H} \in \mathbb{R}^{n \times n}$, denote its largest and smallest eigenvalues by $\lambda^{\max}$ and $\lambda^{\min}$. Then, for any positive semi-definite diagonal matrix $\boldsymbol{Q} = \text{diag}(q_1, \cdots, q_n)$, $\boldsymbol{HQ}$ has $n$ eigenvalues that all lie in $[\min_i q_i \cdot \lambda^{\min}, \max_i q_i \cdot \lambda^{\max}]$.*

*Proof.* $\boldsymbol{H}$ is a positive definite symmetric matrix, so there exists $\boldsymbol{A} \in \mathbb{R}^{n \times n}$ such that $\boldsymbol{H} = \boldsymbol{A}^\top \boldsymbol{A}$, and $\boldsymbol{A}$ is full-rank. First, any eigenvalue of $\boldsymbol{AQA}^\top$ is also an eigenvalue of $\boldsymbol{A}^\top \boldsymbol{AQ}$, because for any eigenvalue $\lambda$ of $\boldsymbol{AQA}^\top$ we have some $\boldsymbol{v} \neq 0$ such that $\boldsymbol{AQA}^\top \boldsymbol{v} = \lambda \boldsymbol{v}$. Multiplying both sides by $\boldsymbol{A}^\top$ on the left yields $\boldsymbol{A}^\top \boldsymbol{AQ}(\boldsymbol{A}^\top \boldsymbol{v}) = \lambda(\boldsymbol{A}^\top \boldsymbol{v})$ which implies that $\lambda$ is also an eigenvalue of $\boldsymbol{A}^\top \boldsymbol{AQ}$ because $\boldsymbol{A}^\top \boldsymbol{v} \neq 0$ as $\lambda \boldsymbol{v} \neq 0$.

Second, by condition we know that the eigenvalues of $\boldsymbol{A}^\top \boldsymbol{A}$ are all in $[\lambda^{\min}, \lambda^{\max}]$ where $\lambda^{\min} > 0$, which implies for any unit vector $\boldsymbol{v}$, $\boldsymbol{v}^\top \boldsymbol{A}^\top \boldsymbol{A} \boldsymbol{v} \in [\lambda^{\min}, \lambda^{\max}]$, which is equivalent to $\|\boldsymbol{A}\boldsymbol{v}\|_2 \in [\sqrt{\lambda^{\min}}, \sqrt{\lambda^{\max}}]$. Thus, we have $\boldsymbol{v}^\top \boldsymbol{A}^\top \boldsymbol{Q} \boldsymbol{A} \boldsymbol{v} \in [\lambda^{\min} \min_i q_i, \lambda^{\max} \max_i q_i]$, which implies that the eigenvalues of $\boldsymbol{A}^\top \boldsymbol{Q} \boldsymbol{A}$ are all in $[\lambda^{\min} \min_i q_i, \lambda^{\max} \max_i q_i]$.

Thus, the eigenvalues of $\boldsymbol{HQ} = \boldsymbol{A}^\top \boldsymbol{AQ}$ are all in $[\lambda^{\min} \min_i q_i, \lambda^{\max} \max_i q_i]$. $\qquad\square$

**Proof of Lemma 16** By the condition $\ell$ satisfies, without loss of generality, assume that the elements of $U(\theta)$ are in $[0, 1]$ for all $\theta$. Then, let $\eta \leq (\mu + \lambda^{\min} + \lambda^{\max})^{-1}$. (If the elements of $U(\theta)$ are bounded by $[0, C]$, then we can let $\eta \leq (\mu + C\lambda^{\min} + C\lambda^{\max})^{-1}$ and prove the result in the same way.)

With $L_2$ penalty, the update rule of the GRW for the neural network is:

$$\theta^{(t+1)} = \theta^{(t)} - \eta J(\theta^{(t)})\boldsymbol{Q}^{(t)}g(\theta^{(t)}) - \eta\mu(\theta^{(t)} - \theta^{(0)}) \tag{98}$$

And the update rule for the linearized neural network is:

$$\theta_{\text{lin}}^{(t+1)} = \theta_{\text{lin}}^{(t)} - \eta J(\theta^{(0)})\boldsymbol{Q}^{(t)}g(\theta_{\text{lin}}^{(t)}) - \eta\mu(\theta_{\text{lin}}^{(t)} - \theta^{(0)}) \tag{99}$$

By Proposition 11, $f(\boldsymbol{x};\theta)$ converges in probability to a zero-mean Gaussian process. Thus, for any $\delta > 0$, there exists a constant $R_0 > 0$ such that with probability at least $(1-\delta/3)$, $\left\|g(\theta^{(0)})\right\|_2 < R_0$. Let $M$ be as defined in Lemma 13. Denote $A = \eta M R_0$, and let $C_0 = \frac{4A}{\eta\mu}$ in Lemma 13[6]. By Lemma 13, there exists $D_1$ such that for all $\tilde{d} \geq D_1$, with probability at least $(1 - \delta/3)$, (52) is true.

Similar to the proof of Proposition 17, we can show that for arbitrary $\tilde{\theta}$, all non-zero eigenvalues of $J(\theta^{(0)})\boldsymbol{Q}^{(t)}U(\tilde{\theta})J(\theta^{(0)})^\top$ are eigenvalues of $J(\theta^{(0)})^\top J(\theta^{(0)})\boldsymbol{Q}^{(t)}U(\tilde{\theta})$. This is because for any $\lambda \neq 0$, if $J(\theta^{(0)})\boldsymbol{Q}^{(t)}U(\tilde{\theta})J(\theta^{(0)})^\top\boldsymbol{v} = \lambda\boldsymbol{v}$, then $J(\theta^{(0)})^\top J(\theta^{(0)})\boldsymbol{Q}^{(t)}U(\tilde{\theta})(J(\theta^{(0)})^\top\boldsymbol{v}) = \lambda(J(\theta^{(0)})^\top\boldsymbol{v})$, and $J(\theta^{(0)})^\top\boldsymbol{v} \neq 0$ since $\lambda\boldsymbol{v} \neq 0$, so $\lambda$ is also an eigenvalue of $J(\theta^{(0)})^\top J(\theta^{(0)})\boldsymbol{Q}^{(t)}U(\tilde{\theta})$. On the other hand, by Proposition 3, $J(\theta^{(0)})^\top J(\theta^{(0)})\boldsymbol{Q}^{(t)}U(\tilde{\theta})$ converges in probability to $\Theta\boldsymbol{Q}^{(t)}U(\tilde{\theta})$ whose eigenvalues are all in $[0, \lambda^{\max}]$ by Proposition 17. So there exists $D_2$ such that for all $\tilde{d} \geq D_2$, with probability at least $(1 - \delta/3)$, the eigenvalues of $J(\theta^{(0)})\boldsymbol{Q}^{(t)}U(\tilde{\theta})J(\theta^{(0)})^\top$ are all in $[0, \lambda^{\max} + \lambda^{\min}]$ for all $t$.

By union bound, with probability at least $(1 - \delta)$, all three above are true, which we will assume in the rest of this proof.

First, we need to prove that there exists $D_0$ such that for all $\tilde{d} \geq D_0$, $\sup_{t\geq0}\left\|\theta^{(t)} - \theta^{(0)}\right\|_2$ is bounded with high probability. Denote $a_t = \theta^{(t)} - \theta^{(0)}$. By (98) we have

$$\begin{aligned} a_{t+1} =&(1 - \eta\mu)a_t - \eta[J(\theta^{(t)}) - J(\theta^{(0)})]\boldsymbol{Q}^{(t)}g(\theta^{(t)}) \\ &- \eta J(\theta^{(0)})\boldsymbol{Q}^{(t)}[g(\theta^{(t)}) - g(\theta^{(0)})] - \eta J(\theta^{(0)})\boldsymbol{Q}^{(t)}g(\theta^{(0)}) \end{aligned} \tag{100}$$

which implies

$$\begin{aligned} \left\|a_{t+1}\right\|_2 \leq& \left\|(1 - \eta\mu)\boldsymbol{I} - \eta J(\theta^{(0)})\boldsymbol{Q}^{(t)}U(\tilde{\theta}^{(t)})J(\tilde{\theta}^{(t)})^\top\right\|_2 \left\|a_t\right\|_2 \\ &+ \eta\left\|J(\theta^{(t)}) - J(\theta^{(0)})\right\|_F\left\|g(\theta^{(t)})\right\|_2 + \eta\left\|J(\theta^{(0)})\right\|_F\left\|g(\theta^{(0)})\right\|_2 \end{aligned} \tag{101}$$

where $\tilde{\theta}^{(t)}$ is some linear interpolation between $\theta^{(t)}$ and $\theta^{(0)}$. Our choice of $\eta$ ensures that $\eta\mu < 1$.

Now we prove by induction that $\left\|a_t\right\|_2 < C_0$. It is true for $t = 0$, so we need to prove that if $\left\|a_t\right\|_2 < C_0$, then $\left\|a_{t+1}\right\|_2 < C_0$.

For the first term on the right-hand side of (101), we have

$$\begin{aligned} \left\|(1 - \eta\mu)\boldsymbol{I} - \eta J(\theta^{(0)})\boldsymbol{Q}^{(t)}U(\tilde{\theta}^{(t)})J(\tilde{\theta}^{(t)})^\top\right\|_2 \leq&(1 - \eta\mu)\left\|\boldsymbol{I} - \frac{\eta}{1 - \eta\mu}J(\theta^{(0)})\boldsymbol{Q}^{(t)}U(\tilde{\theta}^{(t)})J(\theta^{(0)})^\top\right\|_2 \\ &+ \eta\left\|J(\theta^{(0)})\right\|_F\left\|J(\tilde{\theta}^{(t)}) - J(\theta^{(0)})\right\|_F \end{aligned} \tag{102}$$

Since $\eta/(1 - \eta\mu) \leq (\lambda^{\min} + \lambda^{\max})^{-1}$ by our choice of $\eta$, we have

$$\left\|\boldsymbol{I} - \frac{\eta}{1 - \eta\mu}J(\theta^{(0)})\boldsymbol{Q}^{(t)}U(\tilde{\theta}^{(t)})J(\theta^{(0)})^\top\right\|_2 \leq 1 \tag{103}$$

On the other hand, we can use (52) since $\left\|a_t\right\|_2 < C_0$, so $\left\|J(\theta^{(0)})\right\|_F\left\|J(\tilde{\theta}^{(t)}) - J(\theta^{(0)})\right\|_F \leq \frac{M^2}{\sqrt[4]{\tilde{d}}}C_0$. Therefore, there exists $D_3$ such that for all $\tilde{d} \geq D_3$,

$$\left\|(1 - \eta\mu)\boldsymbol{I} - \eta J(\theta^{(0)})\boldsymbol{Q}^{(t)}U(\tilde{\theta}^{(t)})J(\tilde{\theta}^{(t)})^\top\right\|_2 \leq 1 - \frac{\eta\mu}{2} \tag{104}$$

---

[6]Note that Lemma 13 only depends on the network structure and does not depend on the update rule, so we can use this lemma here.

For the second term, we have

$$
\begin{aligned}
\left\| g(\theta^{(t)}) \right\|_2 &\leq \left\| g(\theta^{(t)}) - g(\theta^{(0)}) \right\|_2 + \left\| g(\theta^{(0)}) \right\|_2 \\
&\leq \left\| J(\tilde{\theta}^{(t)}) \right\|_2 \left\| U(\tilde{\theta}^{(t)}) \right\|_2 \left\| \theta^{(t)} - \theta^{(0)} \right\|_2 + R_0 \leq M C_0 + R_0
\end{aligned}
\tag{105}
$$

And for the third term, we have

$$
\eta \left\| J(\theta^{(0)}) \right\|_F \left\| g(\theta^{(0)}) \right\|_2 \leq \eta M R_0 = A
\tag{106}
$$

Thus, we have

$$
\| a_{t+1} \|_2 \leq \left( 1 - \frac{\eta\mu}{2} \right) \| a_t \|_2 + \frac{\eta M (M C_0 + R_0)}{\sqrt[4]{\tilde{d}}} + A
\tag{107}
$$

So there exists $D_4$ such that for all $\tilde{d} \geq D_4$, $\| a_{t+1} \|_2 \leq \left( 1 - \frac{\eta\mu}{2} \right) \| a_t \|_2 + 2A$. This shows that if $\| a_t \|_2 < C_0$ is true, then $\| a_{t+1} \|_2 < C_0$ will also be true.

In conclusion, for all $\tilde{d} \geq D_0 = \max\{D_1, D_2, D_3, D_4\}$, $\left\| \theta^{(t)} - \theta^{(0)} \right\|_2 < C_0$ is true for all $t$. This also implies that for $C_1 = M C_0 + R_0$, we have $\left\| g(\theta^{(t)}) \right\|_2 \leq C_1$ for all $t$ by (105). Similarly, we can prove that $\| \theta_{\mathrm{lin}}^{(t)} - \theta^{(0)} \|_2 < C_0$ for all $t$.

Second, let $\Delta_t = \theta_{\mathrm{lin}}^{(t)} - \theta^{(t)}$. Then we have

$$
\Delta_{t+1} - \Delta_t = \eta(J(\theta^{(t)}) \boldsymbol{Q}^{(t)} g(\theta^{(t)}) - J(\theta^{(0)}) \boldsymbol{Q}^{(t)} g(\theta_{\mathrm{lin}}^{(t)}) - \mu \Delta_t)
\tag{108}
$$

which implies

$$
\Delta_{t+1} = \left[ (1 - \eta\mu)\boldsymbol{I} - \eta J(\theta^{(0)}) \boldsymbol{Q}^{(t)} U(\tilde{\theta}^{(t)}) J(\tilde{\theta}^{(t)})^\top \right] \Delta_t + \eta(J(\theta^{(t)}) - J(\theta^{(0)})) \boldsymbol{Q}^{(t)} g(\theta^{(t)})
\tag{109}
$$

where $\tilde{\theta}^{(t)}$ is some linear interpolation between $\theta^{(t)}$ and $\theta_{\mathrm{lin}}^{(t)}$. By (104), with probability at least $(1 - \delta)$ for all $\tilde{d} \geq D_0$, we have

$$
\begin{aligned}
\| \Delta_{t+1} \|_2 &\leq \left\| (1 - \eta\mu)\boldsymbol{I} - \eta J(\theta^{(0)}) \boldsymbol{Q}^{(t)} U(\tilde{\theta}^{(t)}) J(\tilde{\theta}^{(t)})^\top \right\|_2 \| \Delta_t \|_2 + \eta \left\| J(\theta^{(t)}) - J(\theta^{(0)}) \right\|_F \left\| g(\theta^{(t)}) \right\|_2 \\
&\leq \left( 1 - \frac{\eta\mu}{2} \right) \| \Delta_t \|_2 + \eta \frac{M}{\sqrt[4]{\tilde{d}}} C_0 C_1
\end{aligned}
\tag{110}
$$

Again, as $\Delta_0 = 0$, we can prove by induction that for all $t$,

$$
\| \Delta_t \|_2 < \frac{2 M C_0 C_1}{\mu} \tilde{d}^{-1/4}
\tag{111}
$$

For any test point $\boldsymbol{x}$ such that $\| \boldsymbol{x} \|_2 \leq 1$, we have

$$
\begin{aligned}
\left| f_{\mathrm{reg}}^{(t)}(\boldsymbol{x}) - f_{\mathrm{linreg}}^{(t)}(\boldsymbol{x}) \right| &= \left| f(\boldsymbol{x}; \theta^{(t)}) - f_{\mathrm{lin}}(\boldsymbol{x}; \theta_{\mathrm{lin}}^{(t)}) \right| \\
&\leq \left| f(\boldsymbol{x}; \theta^{(t)}) - f_{\mathrm{lin}}(\boldsymbol{x}; \theta^{(t)}) \right| + \left| f_{\mathrm{lin}}(\boldsymbol{x}; \theta^{(t)}) - f_{\mathrm{lin}}(\boldsymbol{x}; \theta_{\mathrm{lin}}^{(t)}) \right| \\
&\leq \left| f(\boldsymbol{x}; \theta^{(t)}) - f_{\mathrm{lin}}(\boldsymbol{x}; \theta^{(t)}) \right| + \left\| \nabla_\theta f(\boldsymbol{x}; \theta^{(0)}) \right\|_2 \left\| \theta^{(t)} - \theta_{\mathrm{lin}}^{(t)} \right\|_2 \\
&\leq \left| f(\boldsymbol{x}; \theta^{(t)}) - f_{\mathrm{lin}}(\boldsymbol{x}; \theta^{(t)}) \right| + M \| \Delta_t \|_2
\end{aligned}
\tag{112}
$$

For the first term, note that

$$
\begin{cases}
f(\boldsymbol{x}; \theta^{(t)}) - f(\boldsymbol{x}; \theta^{(0)}) = \nabla_\theta f(\boldsymbol{x}; \tilde{\theta}^{(t)})(\theta^{(t)} - \theta^{(0)}) \\
f_{\mathrm{lin}}(\boldsymbol{x}; \theta^{(t)}) - f_{\mathrm{lin}}(\boldsymbol{x}; \theta^{(0)}) = \nabla_\theta f(\boldsymbol{x}; \theta^{(0)})(\theta^{(t)} - \theta^{(0)})
\end{cases}
\tag{113}
$$

where $\tilde{\theta}^{(t)}$ is some linear interpolation between $\theta^{(t)}$ and $\theta^{(0)}$. Since $f(\boldsymbol{x}; \theta^{(0)}) = f_{\text{lin}}(\boldsymbol{x}; \theta^{(0)})$,

$$\left| f(\boldsymbol{x}; \theta^{(t)}) - f_{\text{lin}}(\boldsymbol{x}; \theta^{(t)}) \right| \leq \left\| \nabla_\theta f(\boldsymbol{x}; \tilde{\theta}^{(t)}) - \nabla_\theta f(\boldsymbol{x}; \theta^{(0)}) \right\|_2 \left\| \theta^{(t)} - \theta^{(0)} \right\|_2 \leq \frac{M}{\sqrt[4]{\tilde{d}}} C_0^2 \quad (114)$$

Thus, we have shown that for all $\tilde{d} \geq D_0$, with probability at least $(1 - \delta)$ for all $t$ and all $\boldsymbol{x}$,

$$\left| f_{\text{reg}}^{(t)}(\boldsymbol{x}) - f_{\text{linreg}}^{(t)}(\boldsymbol{x}) \right| \leq \left( MC_0^2 + \frac{2M^2 C_0 C_1}{\mu} \right) \tilde{d}^{-1/4} = O(\tilde{d}^{-1/4}) \quad (115)$$

which is the result we need. $\qquad\square$

### D.4.2 Result for Linearized Neural Networks

**Lemma 18.** *Suppose there exists $M_0 > 0$ such that $\left\| \nabla_\theta f^{(0)}(\boldsymbol{x}) \right\|_2 \leq M_0$ for all test point $\boldsymbol{x}$. If the gradients $\nabla_\theta f^{(0)}(\boldsymbol{x}_1), \cdots, \nabla_\theta f^{(0)}(\boldsymbol{x}_n)$ are linearly independent, and the empirical training risk of $f_{\text{linreg}}^{(t)}$ satisfies*

$$\limsup_{t \to \infty} \hat{\mathcal{R}}(f_{\text{linreg}}^{(t)}) < \epsilon, \quad (116)$$

*for some $\epsilon > 0$, then for $\boldsymbol{x}$ such that $\|\boldsymbol{x}\|_2 \leq 1$ we have*

$$\limsup_{t \to \infty} \left| f_{\text{linreg}}^{(t)}(\boldsymbol{x}) - f_{\text{linERM}}^{(t)}(\boldsymbol{x}) \right| = O(\sqrt{\epsilon}). \quad (117)$$

First, we can see that under the new weight update rule, $\theta^{(t)} - \theta^{(0)} \in \text{span}\{\nabla_\theta f^{(0)}(\boldsymbol{x}_1), \cdots, \nabla_\theta f^{(0)}(\boldsymbol{x}_n)\}$ is still true for all $t$. Let $\theta^*$ be the interpolator in $\text{span}(\nabla_\theta f^{(0)}(\boldsymbol{x}_1), \cdots, \nabla_\theta f^{(0)}(\boldsymbol{x}_n))$, then the empirical risk of $\theta$ is $\frac{1}{2n} \sum_{i=1}^n \langle \theta - \theta^*, \nabla_\theta f^{(0)}(\boldsymbol{x}_i) \rangle^2 = \frac{1}{2n} \left\| \nabla_\theta f^{(0)}(\boldsymbol{X})^\top (\theta - \theta^*) \right\|_2^2$. Thus, there exists $T > 0$ such that for any $t \geq T$,

$$\left\| \nabla_\theta f^{(0)}(\boldsymbol{X})^\top (\theta^{(t)} - \theta^*) \right\|_2^2 \leq 2n\epsilon \quad (118)$$

Let the smallest singular value of $\frac{1}{\sqrt{n}} \nabla_\theta f^{(0)}(\boldsymbol{X})$ be $s^{\min}$, and we have $s^{\min} > 0$. Note that the column space of $\nabla_\theta f^{(0)}(\boldsymbol{X})$ is exactly $\text{span}(\nabla_\theta f^{(0)}(\boldsymbol{x}_1), \cdots, \nabla_\theta f^{(0)}(\boldsymbol{x}_n))$. Define $\boldsymbol{H} \in \mathbb{R}^{p \times n}$ such that its columns form an orthonormal basis of this subspace, then there exists $\boldsymbol{G} \in R^{n \times n}$ such that $\nabla_\theta f^{(0)}(\boldsymbol{X}) = \boldsymbol{H}\boldsymbol{G}$, and the smallest singular value of $\frac{1}{\sqrt{n}} \boldsymbol{G}$ is also $s^{\min}$. Since $\theta^{(t)} - \theta^{(0)}$ is also in this subspace, there exists $\boldsymbol{v} \in \mathbb{R}^n$ such that $\theta^{(t)} - \theta^* = \boldsymbol{H}\boldsymbol{v}$. Then we have $\sqrt{2n\epsilon} \geq \left\| \boldsymbol{G}^\top \boldsymbol{H}^\top \boldsymbol{H}\boldsymbol{v} \right\|_2 = \left\| \boldsymbol{G}^\top \boldsymbol{v} \right\|_2$. Thus, $\|\boldsymbol{v}\|_2 \leq \frac{\sqrt{2\epsilon}}{s^{\min}}$, which implies

$$\left\| \theta^{(t)} - \theta^* \right\|_2 \leq \frac{\sqrt{2\epsilon}}{s^{\min}} \quad (119)$$

We have already proved in previous results that if we minimize the unregularized risk with ERM, then $\theta$ always converges to the interpolator $\theta^*$. So for any $t \geq T$ and any test point $\boldsymbol{x}$ such that $\|\boldsymbol{x}\|_2 \leq 1$, we have

$$|f_{\text{linreg}}^{(t)}(\boldsymbol{x}) - f_{\text{linERM}}^{(t)}(\boldsymbol{x})| = |\langle \theta^{(t)} - \theta^*, \nabla_\theta f^{(0)}(\boldsymbol{x}) \rangle| \leq \frac{M_0 \sqrt{2\epsilon}}{s^{\min}} \quad (120)$$

which implies (117). $\qquad\square$

### D.4.3 Proof of Theorem 6

Given that $\hat{\mathcal{R}}(f_{\text{linreg}}^{(t)}) < \epsilon$ for sufficiently large $t$, Lemma 16 implies that

$$\left| \hat{\mathcal{R}}(f_{\text{linreg}}^{(t)}) - \hat{\mathcal{R}}(f_{\text{reg}}^{(t)}) \right| = O(\tilde{d}^{-1/4} \sqrt{\epsilon} + \tilde{d}^{-1/2}) \quad (121)$$

So for a fixed $\epsilon$, there exists $D > 0$ such that for all $d \geq D$, for sufficiently large $t$,

$$\hat{\mathcal{R}}(f_{\text{reg}}^{(t)}) < \epsilon \Rightarrow \hat{\mathcal{R}}(f_{\text{linreg}}^{(t)}) < 2\epsilon \tag{122}$$

By Lemma 5 and Lemma 16, we have

$$\begin{cases} \sup\limits_{t \geq 0} \left| f_{\text{linERM}}^{(t)}(\boldsymbol{x}) - f_{\text{ERM}}^{(t)}(\boldsymbol{x}) \right| = O(\tilde{d}^{-1/4}) \\ \sup\limits_{t \geq 0} \left| f_{\text{linreg}}^{(t)}(\boldsymbol{x}) - f_{\text{reg}}^{(t)}(\boldsymbol{x}) \right| = O(\tilde{d}^{-1/4}) \end{cases} \tag{123}$$

Combining Lemma 18 with (123) derives

$$\limsup_{t \to \infty} \left| f_{\text{reg}}^{(t)}(\boldsymbol{x}) - f_{\text{ERM}}^{(t)}(\boldsymbol{x}) \right| = O(\tilde{d}^{-1/4} + \sqrt{\epsilon}) \tag{124}$$

Letting $\tilde{d} \to \infty$ leads to the result we need. $\qquad\square$

**Remark.** One might wonder whether $\|\nabla_\theta f^{(0)}(\boldsymbol{x})\|_2$ will diverge as $\tilde{d} \to \infty$. In fact, in Lemma 13, we have proved that there exists a constant $M$ such that with high probability, for any $\tilde{d}$ there is $\|\nabla_\theta f^{(0)}(\boldsymbol{x})\|_2 \leq M$ for any $\boldsymbol{x}$ such that $\|\boldsymbol{x}\|_2 \leq 1$. Therefore, it is fine to suppose that there exists such an $M_0$.

## D.5 PROOFS FOR SUBSECTION 5.1

### D.5.1 PROOF OF THEOREM 7

First we need to show that $\hat{\theta}_{\text{MM}}$ is unique. Suppose both $\theta_1$ and $\theta_2$ maximize $\min_{i=1,\cdots,n} y_i \cdot \langle \theta, \boldsymbol{x}_i \rangle$ and $\theta_1 \neq \theta_2$, $\|\theta_1\|_2 = \|\theta_2\|_2 = 1$. Then consider $\theta_0 = \theta/\|\theta\|_2$ where $\theta = (\theta_1 + \theta_2)/2$. Obviously, $\|\theta\|_2 < 1$, and for any $i$, $y_i \cdot \langle \theta, \boldsymbol{x}_i \rangle = (y_i \cdot \langle \theta_1, \boldsymbol{x}_i \rangle + y_i \cdot \langle \theta_2, \boldsymbol{x}_i \rangle)/2$, so $y_i \cdot \langle \theta_0, \boldsymbol{x}_i \rangle > \min\{y_i \cdot \langle \theta_1, \boldsymbol{x}_i \rangle, y_i \cdot \langle \theta_2, \boldsymbol{x}_i \rangle\}$, which implies that $\min_{i=1,\cdots,n} y_i \cdot \langle \theta_0, \boldsymbol{x}_i \rangle > \min\{\min_{i=1,\cdots,n} y_i \cdot \langle \theta_1, \boldsymbol{x}_i \rangle, \min_{i=1,\cdots,n} y_i \cdot \langle \theta_2, \boldsymbol{x}_i \rangle\}$, contradiction!

Now we start proving the result. Without loss of generality, let $(\boldsymbol{x}_1, y_1), \cdots, (\boldsymbol{x}_m, y_m)$ be the samples with the smallest margin to $\boldsymbol{u}$, i.e.

$$\arg\min_{1 \leq i \leq n} y_i \cdot \langle \boldsymbol{u}, \boldsymbol{x}_i \rangle = \{1, \cdots, m\} \tag{125}$$

And denote $y_1 \cdot \langle \boldsymbol{u}, \boldsymbol{x}_1 \rangle = \cdots = y_m \cdot \langle \boldsymbol{u}, \boldsymbol{x}_m \rangle = \gamma_{\boldsymbol{u}}$. Since the training error converges to 0, $\gamma_{\boldsymbol{u}} > 0$. Note that for the logistic loss, if $y_i \cdot \langle \theta, \boldsymbol{x}_i \rangle < y_j \cdot \langle \theta, \boldsymbol{x}_j \rangle$, then for any $M > 0$, there exists an $R_M > 0$ such that for all $R \geq R_M$,

$$\frac{\nabla_\theta \ell(\langle R\theta, \boldsymbol{x}_i \rangle, y_i)}{\nabla_\theta \ell(\langle R\theta, \boldsymbol{x}_j \rangle, y_j)} > M \tag{126}$$

which can be shown with some simple calculation. And because the training error converges to 0, we must have $\|\theta^{(t)}\| \to \infty$. Then, by Assumption 3 this means that when $t$ gets sufficiently large, the impact of $(\boldsymbol{x}_j, y_j)$ to $\theta^{(t)}$ where $j > m$ is an infinitesimal compared to $(\boldsymbol{x}_i, y_i)$ where $i \leq m$ (because there exists a positive constant $\delta$ such that $q_i^{(t)} > \delta$ for all sufficiently large $t$ by Assumption 3). Thus, we must have $\boldsymbol{u} \in \text{span}\{\boldsymbol{x}_1, \cdots, \boldsymbol{x}_m\}$.

Let $\boldsymbol{u} = \alpha_1 y_1 \boldsymbol{x}_1 + \cdots + \alpha_m y_m \boldsymbol{x}_m$. Now we show that $\alpha_i \geq 0$ for all $i = 1, \cdots, m$. This is because when $t$ is sufficiently large such that the impact of $(\boldsymbol{x}_j, y_j)$ to $\theta^{(t)}$ where $j > m$ becomes infinitesimal, we have

$$\theta^{(t+1)} - \theta^{(t)} \approx \eta \frac{q_i^{(t)} \exp(y_i \cdot \langle \theta^{(t)}, \boldsymbol{x}_i \rangle)}{1 + \exp(y_i \cdot \langle \theta^{(t)}, \boldsymbol{x}_i \rangle)} y_i \boldsymbol{x}_i \tag{127}$$

and since $\|\theta^{(t)}\| \to \infty$ as $t \to \infty$, we have

$$\alpha_i \propto \lim_{T \to \infty} \sum_{t=T_0}^{T} \frac{q_i^{(t)} \exp(y_i \cdot \langle \theta^{(t)}, \boldsymbol{x}_i \rangle)}{1 + \exp(y_i \cdot \langle \theta^{(t)}, \boldsymbol{x}_i \rangle)} := \lim_{T \to \infty} \alpha_i(T) \tag{128}$$

where $T_0$ is sufficiently large. Here the notion $\alpha_i \propto \lim_{T\to\infty}\alpha_i(T)$ means that $\lim_{T\to\infty}\frac{\alpha_i(T)}{\alpha_j(T)} = \frac{\alpha_i}{\alpha_j}$ for any pair of $i, j$ and $\alpha_j \neq 0$. Note that each term in the sum is non-negative. This implies that all $\alpha_1, \cdots, \alpha_m$ have the same sign (or equal to 0). On the other hand,

$$\sum_{i=1}^{m} \alpha_i \gamma_{\boldsymbol{u}} = \sum_{i=1}^{m} \alpha_i y_i \cdot \langle \boldsymbol{u}, \boldsymbol{x}_i \rangle = \langle \boldsymbol{u}, \boldsymbol{u} \rangle > 0 \tag{129}$$

Thus, $\alpha_i \geq 0$ for all $i$ and at least one of them is positive. Now suppose $\boldsymbol{u} \neq \hat{\theta}_{\text{MM}}$, which means that $\gamma_{\boldsymbol{u}}$ is smaller than the margin of $\hat{\theta}_{\text{MM}}$. Then, for all $i = 1, \cdots, m$, there is $y_i \cdot \langle \boldsymbol{u}, \boldsymbol{x}_i \rangle < y_i \cdot \langle \hat{\theta}_{\text{MM}}, \boldsymbol{x}_i \rangle$. This implies that

$$\langle \boldsymbol{u}, \boldsymbol{u} \rangle = \sum_{i=1}^{m} \alpha_i y_i \cdot \langle \boldsymbol{u}, \boldsymbol{x}_i \rangle < \sum_{i=1}^{m} \alpha_i y_i \cdot \langle \hat{\theta}_{\text{MM}}, \boldsymbol{x}_i \rangle = \langle \hat{\theta}_{\text{MM}}, \boldsymbol{u} \rangle \tag{130}$$

which is a contradiction. Thus, we must have $\boldsymbol{u} = \hat{\theta}_{\text{MM}}$. $\qquad \square$

### D.5.2 PROOF OF THEOREM 8

Denote the largest and smallest eigenvalues of $\boldsymbol{X}^\top \boldsymbol{X}$ by $\lambda^{\max}$ and $\lambda^{\min}$, and by condition we have $\lambda^{\min} > 0$. Let $\epsilon = \min\{\frac{q^*}{3}, \frac{(q^*\lambda^{\min})^2}{192\lambda^{\max 2}}\}$. Then similar to the proof in Appendix D.2.2, there exists $t_\epsilon$ such that for all $t \geq t_\epsilon$ and all $i$, $q_i^{(t)} \in (q_i - \epsilon, q_i + \epsilon)$. Denote $\boldsymbol{Q} = \text{diag}(q_1, \cdots, q_n)$, then for all $t \geq t_\epsilon$, $\boldsymbol{Q}^{(t)} := \boldsymbol{Q}_\epsilon^{(t)} = \sqrt{\boldsymbol{Q}}\sqrt{\boldsymbol{Q}_{3\epsilon}^{(t)}}$, where we use the subscript $\epsilon$ to indicate that $\left\|\boldsymbol{Q}_\epsilon^{(t)} - \boldsymbol{Q}\right\|_2 < \epsilon$.

First, we prove that $F(\theta)$ is $L$-smooth as long as $\|\boldsymbol{x}_i\|_2 \leq 1$ for all $i$. The gradient of $F$ is

$$\nabla F(\theta) = \sum_{i=1}^{n} q_i \nabla_{\hat{y}} \ell(\langle \theta, \boldsymbol{x}_i \rangle, y_i) \boldsymbol{x}_i \tag{131}$$

Since $\ell(\hat{y}, y)$ is $L$-smooth in $\hat{y}$, we have for any $\theta_1, \theta_2$ and any $i$,

$$\begin{aligned}
&\ell(\langle \theta_2, \boldsymbol{x}_i \rangle, y_i) - \ell(\langle \theta_1, \boldsymbol{x}_i \rangle, y_i) \\
&\leq \nabla_{\hat{y}} \ell(\langle \theta_1, \boldsymbol{x}_i \rangle, y_i) \cdot (\langle \theta_2, \boldsymbol{x}_i \rangle - \langle \theta_1, \boldsymbol{x}_i \rangle) + \frac{L}{2}(\langle \theta_2, \boldsymbol{x}_i \rangle - \langle \theta_1, \boldsymbol{x}_i \rangle)^2 \\
&= \langle \nabla_{\hat{y}} \ell(\langle \theta_1, \boldsymbol{x}_i \rangle, y_i) \cdot \boldsymbol{x}_i, \theta_2 - \theta_1 \rangle + \frac{L}{2}(\langle \theta_2 - \theta_1, \boldsymbol{x}_i \rangle)^2 \\
&\leq \langle \nabla_{\hat{y}} \ell(\langle \theta_1, \boldsymbol{x}_i \rangle, y_i) \cdot \boldsymbol{x}_i, \theta_2 - \theta_1 \rangle + \frac{L}{2} \|\theta_2 - \theta_1\|_2^2
\end{aligned} \tag{132}$$

Thus, we have

$$\begin{aligned}
F(\theta_2) - F(\theta_1) &= \sum_{i=1}^{n} q_i \left[ \ell(\langle \theta_2, \boldsymbol{x}_i \rangle, y_i) - \ell(\langle \theta_1, \boldsymbol{x}_i \rangle, y_i) \right] \\
&\leq \sum_{i=1}^{n} q_i \langle \nabla_{\hat{y}} \ell(\langle \theta_1, \boldsymbol{x}_i \rangle, y_i) \cdot \boldsymbol{x}_i, \theta_2 - \theta_1 \rangle + \frac{L}{2} \sum_{i=1}^{n} q_i \|\theta_2 - \theta_1\|_2^2 \\
&= \langle \nabla F(\theta_1), \theta_2 - \theta_1 \rangle + \frac{L}{2} \|\theta_2 - \theta_1\|_2^2
\end{aligned} \tag{133}$$

which implies that $F(\theta)$ is $L$-smooth.

Denote $\tilde{g}(\theta) = \nabla_{\hat{y}} \ell(f(\boldsymbol{X}; \theta), \boldsymbol{Y}) \in \mathbb{R}^n$, then $\nabla F(\theta^{(t)}) = \boldsymbol{X}\boldsymbol{Q}\tilde{g}(\theta^{(t)})$, and the update rule is

$$\theta^{(t+1)} = \theta^{(t)} - \eta \boldsymbol{X}\boldsymbol{Q}^{(t)}\tilde{g}(\theta^{(t)}) \tag{134}$$

So by the upper quadratic bound, we have

$$F(\theta^{(t+1)}) \leq F(\theta^{(t)}) - \eta \langle \boldsymbol{X}\boldsymbol{Q}\tilde{g}(\theta^{(t)}), \boldsymbol{X}\boldsymbol{Q}^{(t)}\tilde{g}(\theta^{(t)}) \rangle + \frac{\eta^2 L}{2} \left\| \boldsymbol{X}\boldsymbol{Q}^{(t)}\tilde{g}(\theta^{(t)}) \right\|_2^2 \tag{135}$$

Let $\eta_1 = \frac{q^*\lambda^{\min}}{2L(1+3\epsilon)\lambda^{\max}}$. Similar to what we did in Appendix D.2.2 (Eqn. (40)), we can prove that for all $\eta \le \eta_1$, (135) implies that for all $t \ge t_\epsilon$, there is

$$
\begin{aligned}
F(\theta^{(t+1)}) &\le F(\theta^{(t)}) - \frac{\eta q^*\lambda^{\min}}{2}\left\|\sqrt{Q}\tilde{g}(\theta^{(t)})\right\|_2^2 + \frac{\eta^2 L}{2}\left\|X\sqrt{Q_{3\epsilon}^{(t)}}\right\|_2^2\left\|\sqrt{Q}\tilde{g}(\theta^{(t)})\right\|_2^2 \\
&\le F(\theta^{(t)}) - \frac{\eta q^*\lambda^{\min}}{2}\left\|\sqrt{Q}\tilde{g}(\theta^{(t)})\right\|_2^2 + \frac{\eta^2 L}{2}\|X\|_2^2(1+3\epsilon)\left\|\sqrt{Q}\tilde{g}(\theta^{(t)})\right\|_2^2 \\
&\le F(\theta^{(t)}) - \frac{\eta q^*\lambda^{\min}}{4}\left\|\sqrt{Q}\tilde{g}(\theta^{(t)})\right\|_2^2 \\
&\le F(\theta^{(t)}) - \frac{\eta q^{*2}\lambda^{\min}}{4}\left\|\tilde{g}(\theta^{(t)})\right\|_2^2
\end{aligned}
\tag{136}
$$

This shows that $F(\theta^{(t)})$ is monotonically non-increasing. Since $F(\theta) \ge 0$, $F(\theta^{(t)})$ must converge as $t \to \infty$, and we need to prove that it converges to 0. Suppose that $F(\theta^{(t)})$ does not converge to 0, then there exists a constant $C > 0$ such that $F(\theta^{(t)}) \ge 2C$ for all $t$. On the other hand, it is easy to see that there exists $\theta^*$ such that $\ell(\langle\theta^*, x_i\rangle, y_i) < C$ for all $i$. (136) also implies that $\left\|\tilde{g}(\theta^{(t)})\right\|_2 \to 0$ as $t \to \infty$ because we must have $F(\theta^{(t)}) - F(\theta^{(t+1)}) \to 0$.

Note that from (134) we have

$$
\left\|\theta^{(t+1)} - \theta^*\right\|_2^2 = \left\|\theta^{(t)} - \theta^*\right\|_2^2 + 2\eta\langle XQ^{(t)}\tilde{g}(\theta^{(t)}), \theta^* - \theta^{(t)}\rangle + \eta^2\left\|XQ^{(t)}\tilde{g}(\theta^{(t)})\right\|_2^2 \tag{137}
$$

Denote

$$
F_t(\theta) = \sum_{i=1}^n q_i^{(t)}\ell(\langle\theta, x_i\rangle, y_i) \tag{138}
$$

Then $F_t$ is convex because $\ell$ is convex and $q_i^{(t)}$ are non-negative, and $\nabla F_t(\theta^{(t)}) = XQ^{(t)}\tilde{g}(\theta^{(t)})$. By the lower linear bound $F_t(y) \ge F_t(x) + \langle\nabla F_t(x), y - x\rangle$, we have for all $t$,

$$
\langle XQ^{(t)}\tilde{g}(\theta^{(t)}), \theta^* - \theta^{(t)}\rangle \le F_t(\theta^*) - F_t(\theta^{(t)}) \le F_t(\theta^*) - \frac{2}{3}F(\theta^{(t)}) \le C - \frac{4C}{3} = -\frac{C}{3} \tag{139}
$$

because $q_i^{(t)} \ge q_i - \epsilon \ge \frac{2}{3}q_i$ and $\sum_{i=1}^n q_i^{(t)} = 1$. Since $\left\|\tilde{g}(\theta^{(t)})\right\|_2 \to 0$, there exists $T > 0$ such that for all $t \ge T$ and all $\eta \le \eta_0$,

$$
\left\|\theta^{(t+1)} - \theta^*\right\|_2^2 \le \left\|\theta^{(t)} - \theta^*\right\|_2^2 - \frac{\eta C}{3} \tag{140}
$$

which means that $\left\|\theta^{(t)} - \theta^*\right\|_2^2 \to -\infty$ because $\frac{\eta C}{3}$ is a positive constant. This is a contradiction! Thus, $F(\theta^{(t)})$ must converge to 0, which is result (i).

(i) immediately implies (ii) because $\ell$ is strictly decreasing to 0 by condition.

Now let's prove (iii). First of all, the uniqueness of $\theta_R$ can be easily proved from the convexity of $F(\theta)$. The condition implies that $y_i\langle\theta_R, x_i\rangle > 0$, i.e. $\theta_R$ must classify all training samples correctly. If there are two different minimizers $\theta_R$ and $\theta'_R$ in whose norm is at most $R$, then consider $\theta''_R = \frac{1}{2}(\theta_R + \theta'_R)$. By the convexity of $F$, we know that $\theta''_R$ must also be a minimizer, and $\|\theta''_R\|_2 < R$. Thus, $F(\frac{R}{\|\theta''_R\|_2}\theta''_R) < F(\theta''_R)$ and $\|\frac{R}{\|\theta''_R\|_2}\theta''_R\|_2 = R$, which contradicts with the fact that $\theta''_R$ is a minimizer.

To prove the rest of (iii), the key is to consider (135). On one hand, similar to (36) we can prove that for all $t \ge t_\epsilon$, there is

$$
\left|\langle XQ^{(t)}\tilde{g}(\theta^{(t)}), X(Q^{(t)} - Q)\tilde{g}(\theta^{(t)})\rangle\right| \le \lambda^{\max}\sqrt{3\epsilon}\left\|\sqrt{Q^{(t)}}\tilde{g}(\theta^{(t)})\right\|_2^2 \tag{141}
$$

Since we choose $\epsilon = \min\{\frac{q^*}{3}, \frac{(q^*\lambda^{\min})^2}{192\lambda^{\max 2}}\}$, this inequality implies that

$$
\begin{aligned}
\left\|\nabla F_t(\theta^{(t)})\right\|_2^2 &= \left\|XQ^{(t)}\tilde{g}(\theta^{(t)})\right\|_2^2 \ge \lambda^{\min}\left\|Q^{(t)}\tilde{g}(\theta^{(t)})\right\|_2^2 \ge \lambda^{\min}(q^* - \epsilon)\left\|\sqrt{Q^{(t)}}\tilde{g}(\theta^{(t)})\right\|_2^2 \\
&\ge \frac{\lambda^{\min}q^*}{2}\left\|\sqrt{Q^{(t)}}\tilde{g}(\theta^{(t)})\right\|_2^2 \ge 4\left|\langle XQ^{(t)}\tilde{g}(\theta^{(t)}), X(Q^{(t)} - Q)\tilde{g}(\theta^{(t)})\rangle\right|
\end{aligned}
\tag{142}
$$

On the other hand, if $\eta \le \eta_2 = \frac{1}{2L}$, we will have

$$\frac{\eta^2 L}{2} \left\| \boldsymbol{X}\boldsymbol{Q}^{(t)}\tilde{g}(\theta^{(t)}) \right\|_2^2 \le \frac{\eta}{4} \left\| \nabla F_t(\theta^{(t)}) \right\|_2^2 \tag{143}$$

Combining all the above with (135) yields

$$F(\theta^{(t+1)}) - F(\theta^{(t)}) \le -\frac{\eta}{2} \left\| \nabla F_t(\theta^{(t)}) \right\|_2^2 \tag{144}$$

Denote $\boldsymbol{u} = \lim_{R \to \infty} \frac{\theta_R}{\|\theta_R\|_2}$. Similar to Lemma 9 in Ji et al. (2020), we can prove that: for any $\alpha > 0$, there exists a constant $\rho(\alpha) > 0$ such that for any $\theta$ subject to $\|\theta\|_2 \ge \rho(\alpha)$, there is

$$F_t((1+\alpha)\|\theta\|_2\boldsymbol{u}) \le F_t(\theta) \tag{145}$$

for any $t$. Let $t_\alpha \ge t_\epsilon$ satisfy that for all $t \ge t_\alpha$, $\|\theta^{(t)}\|_2 \ge \max\{\rho(\alpha), 1\}$. By the convexity of $F_t$, for all $t \ge t_\alpha$,

$$\langle \nabla F_t(\theta^{(t)}), \theta^{(t)} - (1+\alpha)\|\theta^{(t)}\|_2\boldsymbol{u} \rangle \ge F_t(\theta^{(t)}) - F_t((1+\alpha)\|\theta^{(t)}\|_2\boldsymbol{u}) \ge 0 \tag{146}$$

Thus, we have

$$\begin{aligned}
\langle \theta^{(t+1)} - \theta^{(t)}, \boldsymbol{u} \rangle &= \langle -\eta\nabla F_t(\theta^{(t)}), \boldsymbol{u} \rangle \\
&\ge \langle -\eta\nabla F_t(\theta^{(t)}), \theta^{(t)} \rangle \frac{1}{(1+\alpha)\|\theta^{(t)}\|_2} \\
&= \langle \theta^{(t+1)} - \theta^{(t)}, \theta^{(t)} \rangle \frac{1}{(1+\alpha)\|\theta^{(t)}\|_2} \\
&= \left( \frac{1}{2} \left\| \theta^{(t+1)} \right\|_2^2 - \frac{1}{2} \left\| \theta^{(t)} \right\|_2^2 - \frac{1}{2} \left\| \theta^{(t+1)} - \theta^{(t)} \right\|_2^2 \right) \frac{1}{(1+\alpha)\|\theta^{(t)}\|_2}
\end{aligned} \tag{147}$$

By $\frac{1}{2}(\|\theta^{(t+1)}\|_2 - \|\theta^{(t)}\|_2)^2 \ge 0$, we have $(\frac{1}{2}\|\theta^{(t+1)}\|_2^2 - \frac{1}{2}\|\theta^{(t)}\|_2^2)/\|\theta^{(t)}\|_2 \ge \|\theta^{(t+1)}\|_2 - \|\theta^{(t)}\|_2$. Moreover, by (144) we have

$$\frac{\left\| \theta^{(t+1)} - \theta^{(t)} \right\|_2^2}{2(1+\alpha)\|\theta^{(t)}\|_2} \le \frac{\left\| \theta^{(t+1)} - \theta^{(t)} \right\|_2^2}{2} = \frac{\eta^2 \left\| \nabla F_t(\theta^{(t)}) \right\|_2^2}{2} \le \eta \left( F(\theta^{(t)}) - F(\theta^{(t+1)}) \right) \tag{148}$$

Summing up (147) from $t = t_\alpha$ to $t - 1$, we have

$$\langle \theta^{(t)} - \theta^{(t_\alpha)}, \boldsymbol{u} \rangle \ge \frac{\|\theta^{(t)}\|_2 - \|\theta^{(t_\alpha)}\|_2}{1+\alpha} + \eta \left( F(\theta^{(t)}) - F(\theta^{(t_\alpha)}) \right) \ge \frac{\|\theta^{(t)}\|_2 - \|\theta^{(t_\alpha)}\|_2}{1+\alpha} - \eta F(\theta^{(t_\alpha)}) \tag{149}$$

which implies that

$$\left\langle \frac{\theta^{(t)}}{\left\|\theta^{(t)}\right\|_2}, \boldsymbol{u} \right\rangle \ge \frac{1}{1+\alpha} + \frac{1}{\left\|\theta^{(t)}\right\|_2} \left( \langle \theta^{(t_\alpha)}, \boldsymbol{u} \rangle - \frac{\|\theta^{(t_\alpha)}\|_2}{1+\alpha} - \eta F(\theta^{(t_\alpha)}) \right) \tag{150}$$

Since $\lim_{t \to \infty} \|\theta^{(t)}\|_2 = \infty$, we have

$$\liminf_{t \to \infty} \left\langle \frac{\theta^{(t)}}{\left\|\theta^{(t)}\right\|_2}, \boldsymbol{u} \right\rangle \ge \frac{1}{1+\alpha} \tag{151}$$

Since $\alpha$ is arbitrary, we must have $\lim_{t \to \infty} \frac{\theta^{(t)}}{\|\theta^{(t)}\|_2} = \boldsymbol{u}$ as long as $\eta \le \min\{\eta_1, \eta_2\}$.  □

### D.5.3 COROLLARY OF THEOREM 8

We can show that for the logistic loss, it satisfies all conditions of Theorem 8 and $\lim_{R \to \infty} \frac{\theta_R}{R} = \hat{\theta}_{\text{MM}}$. First of all, for the logistic loss we have $\nabla_{\hat{y}}^2 \ell(\hat{y}, y) = \frac{y^2}{e^{y\hat{y}} + e^{-y\hat{y}} + 2} \le \max_i \frac{y_i^2}{4}$, so $\ell$ is smooth.

Then, we prove that $\lim_{R \to \infty} \frac{\theta_R}{R}$ exists and is equal to $\hat{\theta}_{\text{MM}}$. For the logistic loss, it is easy to show that for any $\hat{\theta}' \neq \hat{\theta}_{\text{MM}}$, there exists an $R(\hat{\theta}') > 0$ and an $\delta(\hat{\theta}') > 0$ such that $F(R \cdot \theta) > F(R \cdot \hat{\theta}_{\text{MM}})$ for all $R \geq R(\hat{\theta}')$ and $\theta \in B(\hat{\theta}', \delta(\hat{\theta}'))$.

Let $S = \{\theta : \|\theta\|_2 = 1\}$. For any $\epsilon > 0$, $S - B(\hat{\theta}_{\text{MM}}, \epsilon)$ is a compact set. And for any $\theta \in S - B(\hat{\theta}_{\text{MM}}, \epsilon)$, there exist $R(\theta)$ and $\delta(\theta)$ as defined above. Thus, there must exist $\theta_1, \cdots, \theta_m \in S - B(\hat{\theta}_{\text{MM}}, \epsilon)$ such that $S - B(\hat{\theta}_{\text{MM}}, \epsilon) \subseteq \cup_{i=1}^m B(\theta_i, \delta(\theta_i))$. Let $R(\epsilon) = \max\{R(\theta_1), \cdots, R(\theta_m)\}$, then for all $R \geq R(\epsilon)$ and all $\theta \in S - B(\hat{\theta}_{\text{MM}}, \epsilon)$, $F(R \cdot \theta) > F(R \cdot \hat{\theta}_{\text{MM}})$, which means that $\frac{\theta_R}{R} \in B(\hat{\theta}_{\text{MM}}, \epsilon)$ for all $R \geq R(\epsilon)$. Therefore, $\lim_{R \to \infty} \frac{\theta_R}{R}$ exists and is equal to $\hat{\theta}_{\text{MM}}$.

Therefore, by Theorem 8, any GRW satisfying Assumption 1 makes a linear model converge to the max-margin classifier under the logistic loss.

### D.6 PROOF OF THEOREM 9

We first consider the regularized linearized neural network $f_{\text{linreg}}^{(t)}$. Since by Proposition 11 $f^{(0)}(\boldsymbol{x})$ is sampled from a zero-mean Gaussian process, there exists a constant $M > 0$ such that $|f^{(0)}(\boldsymbol{x}_i)| < M$ for all $i$ with high probability. Define

$$F(\theta) = \sum_{i=1}^n q_i \ell(\langle \theta, \nabla_\theta f^{(0)}(\boldsymbol{x}_i)\rangle + f^{(0)}(\boldsymbol{x}_i), y_i) \tag{152}$$

Denote $\tilde{\theta}_R = \arg\min_\theta \{F(R \cdot \theta) : \|\theta\|_2 \leq 1\}$. when the linearized neural network is trained by a GRW satisfying Assumption 1 with regularization, since this is convex optimization and the objective function is smooth, we can prove that with a sufficiently small learning rate, as $t \to \infty$, $\theta^{(t)} \to R \cdot \tilde{\theta}_R + \theta^{(0)}$ where $R = \lim_{t \to \infty} \|\theta^{(t)} - \theta^{(0)}\|_2$ (which is the minimizer). And define

$$\gamma = \min_{i=1,\cdots,n} y_i \cdot \langle \hat{\theta}_{\text{MM}}, \nabla_\theta f^{(0)}(\boldsymbol{x}_i)\rangle \tag{153}$$

First, we derive the lower bound of $R$. By Theorem 16, with a sufficiently large $\tilde{d}$, with high probability $\hat{\mathcal{R}}(f_{\text{reg}}^{(t)}) < \epsilon$ implies $\hat{\mathcal{R}}(f_{\text{linreg}}^{(t)}) < 2\epsilon$. By the convexity of $\ell$, we have

$$2\epsilon > \frac{1}{n}\sum_{i=1}^n \ell(\langle R\tilde{\theta}_R, \boldsymbol{x}_i\rangle + f^{(0)}(\boldsymbol{x}_i), y_i) \geq \log\left(1 + \exp\left(-\frac{1}{n}\sum_{i=1}^n (\langle R\tilde{\theta}_R, \boldsymbol{x}_i\rangle + f^{(0)}(\boldsymbol{x}_i))y_i\right)\right)$$

$$\geq \log\left(1 + \exp\left(-\frac{1}{n}\sum_{i=1}^n R\langle \tilde{\theta}_R, \boldsymbol{x}_i\rangle y_i - M\right)\right) \tag{154}$$

which implies that $R = \Omega(-\log 2\epsilon)$ for all $\epsilon \in (0, \frac{1}{4})$.

Denote $\delta = \|\hat{\theta}_{\text{MM}} - \tilde{\theta}_R\|_2$. Let $\theta' = \frac{\hat{\theta}_{\text{MM}} + \tilde{\theta}_R}{2}$, then we can see that $\|\theta'\|_2 = \sqrt{1 - \frac{\delta^2}{4}}$. Let $\tilde{\theta}' = \frac{\theta'}{\|\theta'\|_2}$. By the definition of $\hat{\theta}_{\text{MM}}$, there exists $j$ such that $y_j \cdot \langle \tilde{\theta}', \nabla_\theta f^{(0)}(\boldsymbol{x}_j)\rangle \leq \gamma$, which implies

$$y_j \cdot \left\langle \frac{\hat{\theta}_{\text{MM}} + \tilde{\theta}_R}{2} \frac{1}{\sqrt{1 - \frac{\delta^2}{4}}}, \nabla_\theta f^{(0)}(\boldsymbol{x}_j)\right\rangle \leq \gamma \tag{155}$$

Thus, we have

$$\begin{aligned} y_j \cdot \langle \tilde{\theta}_R, \nabla_\theta f^{(0)}(\boldsymbol{x}_j)\rangle &\leq 2\sqrt{1 - \frac{\delta^2}{4}}\gamma - y_j \cdot \langle \hat{\theta}_{\text{MM}}, \nabla_\theta f^{(0)}(\boldsymbol{x}_j)\rangle \\ &\leq \left(2\sqrt{1 - \frac{\delta^2}{4}} - 1\right)\gamma \\ &\leq \left(2(1 - \frac{\delta^2}{8}) - 1\right)\gamma \qquad (\text{since } \sqrt{1-x} \leq 1 - \frac{x}{2}) \\ &= (1 - \frac{\delta^2}{4})\gamma \end{aligned} \tag{156}$$

On the other hand, we have

$$q_j \log(1 + \exp(-y_j \cdot \langle R \cdot \tilde{\theta}_R, \nabla_\theta f^{(0)}(\boldsymbol{x}_j)\rangle - M)) \leq F(R \cdot \tilde{\theta}_R)$$
$$\leq F(R \cdot \hat{\theta}_{\mathrm{MM}}) \leq \log(1 + \exp(-R\gamma + M)) \tag{157}$$

which implies that

$$q^* \log\left(1 + \exp\left(-(1 - \frac{\delta^2}{4})R\gamma - M\right)\right) \leq \log(1 + \exp(-R\gamma + M)) \tag{158}$$

and this leads to

$$1 + \exp(-R\gamma + M) \geq \left(1 + \exp\left(-(1 - \frac{\delta^2}{4})R\gamma - M\right)\right)^{q^*} \geq 1 + q^* \exp\left(-(1 - \frac{\delta^2}{4})R\gamma - M\right) \tag{159}$$

which is equivalent to

$$-R\gamma + M \geq -(1 - \frac{\delta^2}{4})R\gamma - M + \log(q^*) \tag{160}$$

Thus, we have

$$\delta = O(R^{-1/2}) = O((-\log 2\epsilon)^{-1/2}) \tag{161}$$

So for any test point $\boldsymbol{x}$, since $\|\nabla_\theta f^{(0)}(\boldsymbol{x})\|_2 \leq M_0$, we have

$$|\langle \hat{\theta}_{\mathrm{MM}} - \tilde{\theta}_R, \nabla_\theta f^{(0)}(\boldsymbol{x})\rangle| \leq \delta M_0 = O((-\log 2\epsilon)^{-1/2}) \tag{162}$$

Combined with Theorem 16, we have: with high probability,

$$\limsup_{t\to\infty} |R \cdot f_{\mathrm{MM}}(\boldsymbol{x}) - f_{\mathrm{reg}}^{(t)}(\boldsymbol{x})| = O(R \cdot (-\log 2\epsilon)^{-1/2} + \tilde{d}^{-1/4}) \tag{163}$$

So there exists a constant $C > 0$ such that: As $\tilde{d} \to \infty$, with high probability, for all $\epsilon \in (0, \frac{1}{4})$, if $|f_{\mathrm{MM}}(\boldsymbol{x})| > C \cdot (-\log 2\epsilon)^{-1/2}$, then $f_{\mathrm{reg}}^{(t)}(\boldsymbol{x})$ will have the same sign as $f_{\mathrm{MM}}(\boldsymbol{x})$ for a sufficiently large $t$. Note that this $C$ only depends on $n$, $q^*$, $\gamma$, $M$ and $M_0$, so it is a constant independent of $\epsilon$. $\qquad\square$

**Remark.** Note that Theorem 9 requires Assumption 1 while Theorem 6 does not due to the fundamental difference between the classification and regression. In regression the model converges to a finite point. However, in classification, the training loss converging to zero implies that either (i) The direction of the weight is close to the max-margin classifier or (ii) The norm of the weight is very large. Assumption 1 is used to eliminate the possibility of (ii). If the regularization parameter $\mu$ is sufficiently large, then a small empirical risk could imply a small weight norm. However, in our theorem we do not assume anything on $\mu$, so Assumption 1 is necessary.

## E  A NOTE ON THE PROOFS IN LEE ET AL. (2019)

We have mentioned that the proofs in Lee et al. (2019), particularly the proofs of their Theorem 2.1 and Lemma 1 in their Appendix G, are flawed. In order to fix their proof, we change the network initialization to (9). In this section, we will demonstrate what goes wrong in the proofs in Lee et al. (2019), and how we manage to fix the proof. For clarity, we are referring to the following version of the paper: `https://arxiv.org/pdf/1902.06720v4.pdf`.

To avoid confusion, in this section we will still use the notations used in our paper.

### E.1  THEIR PROBLEMS

Lee et al. (2019) claimed in their Theorem 2.1 that under the conditions of our Lemma 5, for any $\delta > 0$, there exist $\tilde{D} > 0$ and a constant $C$ such that for any $\tilde{d} \geq \tilde{D}$, with probability at least $(1 - \delta)$,

the gap between the output of a sufficiently wide fully-connected neural network and the output of its linearized neural network at any test point $\boldsymbol{x}$ can be uniformly bounded by

$$\sup_{t \geq 0} \left| f^{(t)}(\boldsymbol{x}) - f_{\text{lin}}^{(t)}(\boldsymbol{x}) \right| \leq C\tilde{d}^{-1/2} \qquad \text{(claimed)} \tag{164}$$

where they used the original NTK formulation and initialization in Jacot et al. (2018):

$$\begin{cases} \boldsymbol{h}^{l+1} = \dfrac{W^l}{\sqrt{d_l}} \boldsymbol{x}^l + \beta \boldsymbol{b}^l \\ \boldsymbol{x}^{l+1} = \sigma(\boldsymbol{h}^{l+1}) \end{cases} \quad \text{and} \quad \begin{cases} W_{i,j}^{l(0)} \sim \mathcal{N}(0, 1) \\ b_i^{l(0)} \sim \mathcal{N}(0, 1) \end{cases} \quad (\forall l = 0, \cdots, L) \tag{165}$$

where $\boldsymbol{x}_0 = \boldsymbol{x}$ and $f(\boldsymbol{x}) = h^{L+1}$. However, in their proof in their Appendix G, they did not directly prove their result for the NTK formulation, but instead they proved another result for the following formulation which they called the *standard formulation*:

$$\begin{cases} \boldsymbol{h}^{l+1} = W^l \boldsymbol{x}^l + \beta \boldsymbol{b}^l \\ \boldsymbol{x}^{l+1} = \sigma(\boldsymbol{h}^{l+1}) \end{cases} \quad \text{and} \quad \begin{cases} W_{i,j}^{l(0)} \sim \mathcal{N}(0, \dfrac{1}{d_l}) \\ b_i^{l(0)} \sim \mathcal{N}(0, 1) \end{cases} \quad (\forall l = 0, \cdots, L) \tag{166}$$

See their Appendix F for the definition of their standard formulation. In the original formulation, they also included two constants $\sigma_w$ and $\sigma_b$ for standard deviations, and for simplicity we omit these constants here. Note that the outputs of the NTK formulation and the standard formulation at initialization are actually the same. The only difference is that the norm of the weight $W^l$ and the gradient of the model output with respect to $W^l$ are different for all $l$.

In their Appendix G, they claimed that if a network with the standard formulation is trained by minimizing the squared loss with gradient descent and learning rate $\eta' = \eta/\tilde{d}$, where $\eta$ is our learning rate in Lemma 5 and also their learning rate in their Theorem 2.1, then (164) is true for this network, so it is also true for a network with the NTK formulation because the two formulations have the same network output. And then they claimed in their equation (S37) that applying learning rate $\eta'$ to the standard formulation is equivalent to applying the following learning rates

$$\eta_W^l = \frac{d_l}{d_{\max}} \eta \qquad \text{and} \qquad \eta_{\boldsymbol{b}}^l = \frac{1}{d_{\max}} \eta \tag{167}$$

to $W^l$ and $\boldsymbol{b}^l$ of the NTK formulation, where $d_{\max} = \max\{d_0, \cdots, d_L\}$.

To avoid confusion, in the following discussions we will still use the NTK formulation and initialization if not stated otherwise.

**Problem 1.** Claim (167) is true, but it leads to two problems. The first problem is that $\eta_{\boldsymbol{b}}^l = O(d_{\max}^{-1})$ since $\eta = O(1)$, while their Theorem 2.1 needs the learning rate to be $O(1)$. Nevertheless, this problem can be simply fixed by modifying their standard formulation as $\boldsymbol{h}^{l+1} = W^l \boldsymbol{x}^l + \beta\sqrt{d_l}\boldsymbol{b}^l$ where $b_i^{l(0)} \sim \mathcal{N}(0, d_l^{-1})$. The real problem that is non-trivial to fix is that by (167), there is $\eta_W^0 = \frac{d_0}{d_{\max}}\eta$. However, note that $d_0$ is a constant since it is the dimension of the input space, while $d_{\max}$ goes to infinity. Consequently, in (167) they were essentially using a very small learning rate for the first layer $W^0$ but a normal learning rate for the rest of the layers, which definitely does not match with their claim in their Theorem 2.1.

**Problem 2.** Another big problem is that the proof of their Lemma 1 in their Appendix G is erroneous, and consequently their Theorem 2.1 is unsound as it heavily depends on their Lemma 1. In their Lemma 1, they claimed that for some constant $M > 0$, for any two models with the parameters $\theta$ and $\tilde{\theta}$ such that $\theta, \tilde{\theta} \in B(\theta^{(0)}, C_0)$ for some constant $C_0$, there is

$$\left\| J(\theta) - J(\tilde{\theta}) \right\|_F \leq \frac{M}{\sqrt{\tilde{d}}} \left\| \theta - \tilde{\theta} \right\|_2 \qquad \text{(claimed)} \tag{168}$$

Note that the original claim in their paper was $\left\| J(\theta) - J(\tilde{\theta}) \right\|_F \leq M\sqrt{\tilde{d}} \left\| \theta - \tilde{\theta} \right\|_2$. This is because they were proving this result for their standard formulation. Compared to the standard formulation,

in the NTK formulation $\theta$ is $\sqrt{\tilde{d}}$ times larger, while the Jacobian $J(\theta)$ is $\sqrt{\tilde{d}}$ times smaller. This is also why here we have $\theta, \tilde{\theta} \in B(\theta^{(0)}, C_0)$ instead of $\theta, \tilde{\theta} \in B(\theta^{(0)}, C_0 \tilde{d}^{-1/2})$ for the NTK formulation. Therefore, equivalently they were claiming (168) for the NTK formulation.

However, their proof of (168) in incorrect. Specifically, the right-hand side of their inequality (S86) is incorrect. Using the notations in our Appendix D.3.3, their (S86) essentially claimed that

$$\left\| \boldsymbol{\alpha}^l - \tilde{\boldsymbol{\alpha}}^l \right\|_2 \le \frac{M}{\sqrt{\tilde{d}}} \left\| \theta - \tilde{\theta} \right\|_2 \qquad \text{(claimed)} \qquad (169)$$

for any $\theta, \tilde{\theta} \in B(\theta^{(0)}, C_0)$, where $\boldsymbol{\alpha}^l = \nabla_{\boldsymbol{h}^l} \boldsymbol{h}^{L+1}$ and $\tilde{\boldsymbol{\alpha}}^l$ is the same gradient for the second model. Note that their (S86) does not have the $\sqrt{\tilde{d}}$ in the denominator which appears in (169). This is because for their standard formulation, $\theta$ is $\sqrt{\tilde{d}}$ times smaller than the original NTK formulation, while $\left\| \boldsymbol{\alpha}^l \right\|_2$ has the same order in the two formulations because all $\boldsymbol{h}^l$ are the same.

However, it is actually impossible to prove (169). Consider the following counterexample: Since $\theta$ and $\tilde{\theta}$ are arbitrarily chosen, we can choose them such that they only differ in $b_1^l$ for some $1 \le l < L$. Then, $\left\| \theta - \tilde{\theta} \right\|_2 = \left| b_1^l - \tilde{b}_1^l \right|$. We can see that $\boldsymbol{h}^{l+1}$ and $\tilde{\boldsymbol{h}}^{l+1}$ only differ in the first element, and $\left| h_1^{l+1} - \tilde{h}_1^{l+1} \right| = \left| \beta(b_1^l - \tilde{b}_1^l) \right|$. Moreover, we have $W^{l+1} = \tilde{W}^{l+1}$, so there is

$$\begin{aligned} \boldsymbol{\alpha}^{l+1} - \tilde{\boldsymbol{\alpha}}^{l+1} =& \text{diag}(\dot{\sigma}(\boldsymbol{h}^{l+1})) \frac{W^{l+1\top}}{\sqrt{\tilde{d}}} \boldsymbol{\alpha}^{l+2} - \text{diag}(\dot{\sigma}(\tilde{\boldsymbol{h}}^{l+1})) \frac{\tilde{W}^{l+1\top}}{\sqrt{\tilde{d}}} \tilde{\boldsymbol{\alpha}}^{l+2} \\ =& \left[ \text{diag}(\dot{\sigma}(\boldsymbol{h}^{l+1})) - \text{diag}(\dot{\sigma}(\tilde{\boldsymbol{h}}^{l+1})) \right] \frac{W^{l+1\top}}{\sqrt{\tilde{d}}} \boldsymbol{\alpha}^{l+2} \\ & + \text{diag}(\dot{\sigma}(\tilde{\boldsymbol{h}}^{l+1})) \frac{W^{l+1\top}}{\sqrt{\tilde{d}}} (\boldsymbol{\alpha}^{l+2} - \tilde{\boldsymbol{\alpha}}^{l+2}) \end{aligned} \qquad (170)$$

Then we can lower bound $\left\| \boldsymbol{\alpha}^{l+1} - \tilde{\boldsymbol{\alpha}}^{l+1} \right\|_2$ by

$$\begin{aligned} \left\| \boldsymbol{\alpha}^{l+1} - \tilde{\boldsymbol{\alpha}}^{l+1} \right\|_2 \ge & \left\| \left[ \text{diag}(\dot{\sigma}(\boldsymbol{h}^{l+1})) - \text{diag}(\dot{\sigma}(\tilde{\boldsymbol{h}}^{l+1})) \right] \frac{W^{l+1\top}}{\sqrt{\tilde{d}}} \boldsymbol{\alpha}^{l+2} \right\|_2 \\ & - \left\| \text{diag}(\dot{\sigma}(\tilde{\boldsymbol{h}}^{l+1})) \frac{W^{l+1\top}}{\sqrt{\tilde{d}}} (\boldsymbol{\alpha}^{l+2} - \tilde{\boldsymbol{\alpha}}^{l+2}) \right\|_2 \end{aligned} \qquad (171)$$

The first term on the right-hand side is equal to $\left| \left[ \dot{\sigma}(h_1^{l+1}) - \dot{\sigma}(\tilde{h}_1^{l+1}) \right] \langle W_1^{l+1}/\sqrt{\tilde{d}}, \boldsymbol{\alpha}^{l+2} \rangle \right|$ where $W_1^{l+1}$ is the first row of $W^{l+1}$. We know that $\left\| W_1^{l+1} \right\|_2 = \Theta\left( \sqrt{\tilde{d}} \right)$ with high probability as its elements are sampled from $\mathcal{N}(0, 1)$, and in their (S85) they claimed that $\left\| \boldsymbol{\alpha}^{l+2} \right\|_2 = O(1)$, which is true. In addition, they assumed that $\dot{\sigma}$ is Lipschitz. Hence, we can see that

$$\left\| \left[ \text{diag}(\dot{\sigma}(\boldsymbol{h}^{l+1})) - \text{diag}(\dot{\sigma}(\tilde{\boldsymbol{h}}^{l+1})) \right] \frac{W^{l+1\top}}{\sqrt{\tilde{d}}} \boldsymbol{\alpha}^{l+2} \right\|_2 = O\left( \left| h_1^{l+1} - \tilde{h}_1^{l+1} \right| \right) = O\left( \left\| \theta - \tilde{\theta} \right\|_2 \right) \qquad (172)$$

On the other hand, suppose that claim (169) is true, then $\left\| \boldsymbol{\alpha}^{l+2} - \tilde{\boldsymbol{\alpha}}^{l+2} \right\|_2 = O\left( \tilde{d}^{-1/2} \left\| \theta - \tilde{\theta} \right\|_2 \right)$. Then we can see that the second term on the right-hand side is $O\left( \tilde{d}^{-1/2} \left\| \theta - \tilde{\theta} \right\|_2 \right)$ because $\left\| W^{l+1} \right\|_2 = O(\sqrt{\tilde{d}})$ and $\dot{\sigma}(x)$ is bounded by a constant as $\sigma$ is Lipschitz. Thus, for a very large $\tilde{d}$, the second-term is an infinitesimal compared to the first term, so we can only prove that

$$\left\| \boldsymbol{\alpha}^{l+1} - \tilde{\boldsymbol{\alpha}}^{l+1} \right\|_2 = O\left( \left\| \theta - \tilde{\theta} \right\|_2 \right) \qquad (173)$$

which is different from (169) because it lacks a critical $\tilde{d}^{-1/2}$ and thus leads to a contradiction. Hence, we cannot prove (169) with the $\tilde{d}^{-1/2}$ factor, and consequently we cannot prove (168) with

the $\sqrt{\tilde{d}}$ in the denominator on the right-hand side. As a result, their Lemma 1 and Theorem 2.1 cannot be proved without this critical $\tilde{d}^{-1/2}$. Similarly, we can also construct a counterexample where $\theta$ and $\tilde{\theta}$ only differ in the first row of some $W^l$.

### E.2 OUR FIXES

Regarding Problem 1, we can still use an $O(1)$ learning rate for the first layer in the NTK formulation given that $\|\boldsymbol{x}\|_2 \leq 1$. This is because for the first layer, we have

$$\nabla_{W^0} f(\boldsymbol{x}) = \frac{1}{\sqrt{d_0}} \boldsymbol{x}^0 \alpha^{1\top} = \frac{1}{\sqrt{d_0}} \boldsymbol{x} \alpha^{1\top} \tag{174}$$

For all $l \geq 1$, we have $\|\boldsymbol{x}^l\|_2 = O(\tilde{d}^{1/2})$. However, for $l = 0$, we instead have $\|\boldsymbol{x}^0\|_2 = O(1)$. Thus, we can prove that the norm of $\nabla_{W^0} f(\boldsymbol{x})$ has the same order as the gradient with respect to any other layer, so there is no need to use a smaller learning rate for the first layer.

Regarding Problem 2, in our formulation (8) and initialization (9), the initialization of the last layer of the NTK formulation is changed from the Gaussian initialization $W_{i,j}^{L(0)} \sim \mathcal{N}(0, 1)$ to the zero initialization $W_{i,j}^{L(0)} = 0$. Now we show how this modification solves Problem 2.

The main consequence of changing the initialization of the last layer is that (86) becomes different: instead of $\|W^L\|_2 \leq 3\sqrt{\tilde{d}}$, we now have $\|W^L\|_2 \leq C_0 \leq 3\sqrt[4]{\tilde{d}}$. In fact, for any $r \in (0, 1/2)$, we can prove that $\|W^L\|_2 \leq 3\tilde{d}^r$ for sufficiently large $\tilde{d}$. In our proof we choose $r = 1/4$.

Consequently, instead of $\|\boldsymbol{\alpha}^l\|_2 \leq M_3$, we can now prove that $\|\boldsymbol{\alpha}^l\|_2 \leq M_3 \tilde{d}^{r-1/2}$ for all $l \leq L$ by induction. So now we can prove $\|\boldsymbol{\alpha}^l - \tilde{\boldsymbol{\alpha}}^l\|_2 = O\left(\tilde{d}^{r-1/2} \|\theta - \tilde{\theta}\|_2\right)$ instead of $O\left(\|\theta - \tilde{\theta}\|_2\right)$, because

- For $l < L$, we now have $\|\boldsymbol{\alpha}^{l+1}\|_2 = O(\tilde{d}^{r-1/2})$ instead of $O(1)$, so we can have the additional $\tilde{d}^{r-1/2}$ factor in the bound.
- For $l = L$, although $\|\boldsymbol{\alpha}^{L+1}\|_2 = 1$, note that $\|W^L\|_2$ now becomes $O(\tilde{d}^r)$ instead of $O(\tilde{d}^{1/2})$, so again we can decrease the bound by a factor of $\tilde{d}^{r-1/2}$.

Then, with this critical $\tilde{d}^{r-1/2}$, we can prove the approximation theorem with the form

$$\sup_{t \geq 0} \left| f^{(t)}(\boldsymbol{x}) - f_{\text{lin}}^{(t)}(\boldsymbol{x}) \right| \leq C \tilde{d}^{r-1/2} \tag{175}$$

for any $r \in (0, 1/2)$, though we cannot really prove the $O(\tilde{d}^{-1/2})$ bound as originally claimed in (164). So this is how we solve Problem 2.

One caveat of changing the initialization to zero initialization is whether we can still safely assume that $\lambda^{\min} > 0$ where $\lambda^{\min}$ is the smallest eigenvalue of $\Theta$, the kernel matrix of our new formulation. The answer is yes. In fact, in our Proposition 3 we proved that $\Theta$ is non-degenerated (which means that $\Theta(\boldsymbol{x}, \boldsymbol{x}')$ still depends on $\boldsymbol{x}$ and $\boldsymbol{x}'$), and under the overparameterized setting where $d_L \gg n$, chances are high that $\Theta$ is full-rank. Hence, we can still assume that $\lambda^{\min} > 0$.

As a final remark, one key reason why we need to initialize $W^L$ as zero is that the dimension of the output space (i.e. the dimension of $\boldsymbol{h}^{L+1}$) is finite, and in our case it is 1. Suppose we allow the dimension of $\boldsymbol{h}^{L+1}$ to be $\tilde{d}$ which goes to infinity, then using the same proof techniques, for the NTK formulation we can prove that $\sup_t \left\| \boldsymbol{h}^{L+1(t)} - \boldsymbol{h}_{\text{lin}}^{L+1(t)} \right\|_2 \leq C$, i.e. the gap between two vectors of infinite dimension is always bounded by a finite constant. This is the approximation theorem we need for the infinite-dimensional output space. However, when the dimension of the output space is finite, $\sup_t \left\| \boldsymbol{h}^{L+1(t)} - \boldsymbol{h}_{\text{lin}}^{L+1(t)} \right\|_2 \leq C$ no longer suffices, so we need to decrease the order of the norm of $W^L$ in order to obtain a smaller upper bound.

