# OpenReview forum: "Understanding Why Generalized Reweighting Does Not Improve Over ERM"
_ICLR.cc/2023/Conference — ICLR 2023 poster_

### Official Review · Reviewer_Zc4T · 2022-10-24

**Confidence:** 4
**Correctness:** 4
**Technical Novelty And Significance:** 3
**Empirical Novelty And Significance:** 3
**Recommendation:** 6

**Clarity, Quality, Novelty And Reproducibility:**

This paper is well organized and easy to follow. This work is of good quality with convinced theoretical results. The novelty seems to be incremental since a related reference is omitted (See weaknesses). Considering that this is a theory work and there is no new proposed method, reproducibility is not appliable.

**Strength And Weaknesses:**

Strength: The theoretical analysis is comprehensive and convincing, which I believe could bring new insights to the community in designing DRO methods. The authors also conduct experiments to support their theoretical results.
Weaknesses: As the authors have pointed out, the analysis relies on some strong assumptions, which could be the major weakness of this work. Besides, previous work [1] has the similar results that minimizing the weighted empirical risk is equivalent to minimizing the vanilla empirical risk. The author should make comparison and explain the differences in detail. I also wonder whether the linearly independent gradients assumption in Theorem 4 could be satisfied in real applications.

[1] Does Distributionally Robust Supervised Learning Give Robust Classifiers? Weihua Hu, Gang Niu, Issei Sato and Masashi Sugiyama.


**Summary Of The Paper:**

This paper provides a detailed and comprehensive theoretical analysis to the problem that existing DRO approaches do not bring significant performance gain over ERM. The analysis focus on the GRW framework that minimizes the weighted empirical risk. The authors show that for linear models and wide neural networks, the resulting model yielded by GRW is closed to that obtained by ERM on both regression and classification tasks, namely, GRW does not improve over ERM. The authors further show that the effect of adding small regularization is of no help on alleviating this problem. With the theoretical results, principal ways are presented to improve DRG.

**Summary Of The Review:**

This paper studies a vital problem that existing DRO approaches do not bring significant performance gain over ERM from theory perspective. The authors provide detailed analysis of two representative models, linear models and wide neural networks, on regression and classification tasks. Experiments are also provided to support the results. My major concern is the relation between this work and previous work (See weaknesses). The authors are encouraged to provide detailed comparison and explain the differences.

---

> ### Author Response · Authors · 2022-11-10
> **Response to Reviewer Zc4T**
>
> Thank you for your very detailed comments! We would like to address your concerns as follows:
> ### Regarding the assumptions we made
> As we have mentioned in the last paragraph of Section 1 and in Section 6.2, our results do rely on some strong assumptions. Here, we would like to provide three justifications for these assumptions:
> 1. These assumptions (linear or wide NN and without early stopping) are very common and have been widely adopted in existing literature, and the results previously derived from these assumptions (such as the analysis on the loss landscape and training dynamics) have been widely acknowledged to be useful for general deep learning.
> 2. These assumptions are necessary, because at this point, for a general DNN, we cannot even prove basic things like it can reach the global optima under gradient descent or it can generalize well. In order to prove anything meaningful, we have to make these assumptions.
> 3. The results we prove have significant implications: Our results suggest that a deeper understanding in DRG is crucial for developing better algorithms for distribution shift, which pinpoints a critical bottleneck of current distribution shift research, and aims to open new directions for distributionally robust training algorithms such as the ones we listed in Section 6.1.
>
> ---
>
> ### Regarding the difference from Hu et al. [1]
> We would like to clarify that our results are **substantially different** from the results in [1]. [1] proved that in classification that uses the zero-one loss, GRW methods such as DRSL are equivalent to ERM, in the sense that the minimizer of the DRSL risk is also the minimizer of the average risk. However, this does not mean that DRSL and ERM will always converge to the *same point*, as there could be multiple minimizers. Their result relies on the zero-one loss, which leads to a monotonic linear relationship between the DRSL risk and the average risk. Moreover, their result is only about the relationship between two minimizers, and they did not prove that DRSL and ERM can actually converge to these global minima.
>
> On the other hand, in our results, we first show that without regularization, GRW and ERM will *converge to the exact same point*, so that they have equivalent implicit biases, which is a much stronger result. Then we show that even with regularization, if the regularization is not large enough, GRW will still converge to a point that is very close to the point ERM converges to. Our results do not depend on the loss function, and work for both the squared loss for regression and the logistic loss for classification (and can be extended to other losses). Instead, our results depend on the optimization method (must be first-order or gradient-based) as well as the model architecture (linear or wide NN), since we need to explicitly prove that both GRW and ERM can reach the global minima if trained under a small learning rate for sufficiently long.
>
> In a word, [1] proves the equivalence between GRW and ERM under the zero-one loss with a monotonic relationship between the two risk functions, while our results focus on the optimization and training dynamics, and prove that GRW and ERM have almost equivalent implicit biases.
>
> We sincerely apologize for not citing [1] which is indeed a related work, and thank the reviewer for bringing this point up. We have added this comparison with [1] in Appendix A in the revision.
>
> ---
>
> ### Regarding the linear independence of gradients in Theorem 4
> This assumption can be very easily satisfied for an overparameterized model. Let p be the number of parameters in the model, then each $\nabla_\theta f^{(0)}(x_i)$ is p-dimensional. If the model is a non-linear neural network, $p \gg n$ and $x_1,\cdots,x_n$ are i.i.d., then $\nabla_\theta f^{(0)}(x_1), \cdots, \nabla_\theta f^{(0)}(x_n)$ are almost surely linearly independent. We have added this clarification in the revision.
>
> ---
>
> We again thank the reviewer for this very detailed and useful feedback, and sincerely hope that our response can address your concerns. We are looking forward to having further discussions with you during the discussion period.
>
> [1] Hu et al., Does distributionally robust supervised learning give robust classifiers? ICML 2018.

---

### Official Review · Reviewer_ZGFE · 2022-10-25

**Confidence:** 2
**Clarity, Quality, Novelty And Reproducibility:** see above
**Correctness:** 3
**Technical Novelty And Significance:** 2
**Empirical Novelty And Significance:** Not applicable
**Recommendation:** 5

**Strength And Weaknesses:**

Strength

- This is a very important and interesting problem.

- the introduction is well-written.

- The paper clearly states what it contributes and how it relates to existing papers.s.

- The paper clearly states its limitations.

Main weakness

- After reading this paper, I don’t think I gained an intuitive understanding of why generalized reweighting does not improve over erm.
It would make the paper more accessible to a wider audience if parts of the paper were devoted to explaining these technical descriptions in plain English.

Minor points

- Currently, the related work section appears to be an afterthought. It appears in the appendix and does not explain how this work fits into existing literature.
As an example, in the section on group fairness, it is unclear how the subpopulation shift problem in this paper is related to group fairness. It does not specify how this work relates to these papers.

- Furthermore, there is a large literature on distribution generalization, which is not included in this paper. Of course, there is no need for the authors to cite every paper in a  growing field. However, I would like to see some examples of what constitutes a generalized reweighing algorithm and what does not.

- Some jargons are not well defined (e.g., sub-population shift, implicit bias)

Overall, I like the motivating problem and think this could be a high-impact piece of work. The paper could be improved if some of the issues outlined above were addressed.

I am willing to change my score if the issues mentioned are addressed.

**Summary Of The Paper:**

This paper studies whether generalized reweighing algorithms can improve out-of-distribution generalization.

First, it defines "generalized reweighting", and then shows that generalized reweighting algorithms have the same implicit bias as ERM, meaning the model converges to the same parameters given the same initial training point.

This paper proves this result for both regression and classification with overparameterized linear models (including L2 regularization) and wide neural networks.

For linear models, the theorems are more straightforward.

The theorems for the wide neural nets are based on the theoretical results of neural tangent kernels.

**Summary Of The Review:**

This paper provides a theoretical analysis of why some DRO algorithms do not improve over ERM. There is a good motivation for the problem and it is an important one;  the main paper is somewhat clear.  The main problem is that the paper does not provide intuitive explanations but instead relies only on technical theorems to answer the motivating questions.

---

> ### Author Response · Authors · 2022-11-10
> **Response to Reviewer ZGFE**
>
> Thank you for your very helpful comments! We would like to address your concerns as follows:
>
> ### Regarding making the paper more accessible to a wider audience
> This is a very good point and we thank the reviewer for this suggestion. Intuitively the main result we proved in this work, in plain English, is the following: **The points where GRW and ERM converge to are very close to each other, so GRW cannot do better than ERM because it does not yield a significantly different model, i.e. the GRW model and the ERM model produce the same predictions on almost all samples so their performances should be almost the same**. We have added this intuitive explanation in Section 1 in the revision. We have also added intuitive explanations to many of our results throughout the paper, so that the readers can have a better knowledge of what each result is talking about.
>
> ---
>
> ### Regarding improving the related work section
> This is also a very good suggestion. We have updated the related work section in the revision and invite the reviewer to check it out. The relationship between the subpopulation shift problem and group fairness is the following: In both problems, the data is divided into several groups and the goal in general is to train a model that can do well on majority as well as minority groups. The difference is how “fairness” is defined. A lot of fairness notions, such as equal opportunity and statistical parity, have been proposed. As mentioned in Appendix A, the subpopulation shift problem uses the Rawlsian max-min fairness notion. In this sense, the subpopulation shift problem is a subset of the group fairness problem.
>
> ---
>
> ### Other comments
> 1. Regarding related work on domain generalization: We have added a section introducing recent work on domain generalization in the new related work section in Appendix A.
> 2. Regarding some terms used in the paper not being well defined: We have changed the writing in several places in the revision - We either avoid using these terms, or add additional definitions or explanations alongside these terms.
>
> ---
> We again thank the reviewer for these very constructive comments - They really help us improve this paper. We sincerely hope that our response and revision address your concerns, and are looking forward to having further discussions with you during the discussion period.

---

### Official Review · Reviewer_NJFN · 2022-10-25

**Confidence:** 3
**Correctness:** 3
**Technical Novelty And Significance:** 3
**Empirical Novelty And Significance:** 2
**Recommendation:** 5

**Clarity, Quality, Novelty And Reproducibility:**

- Clarity: this paper is well-structured and clearly written. Discussions on the future direction and the limitation are provided.

- Quality: the results provided by the paper seems correct, but the assumptions are strong to me.

- Novelty: a similar conclusion that DRO performs like ERM has been shown in the previous work. But the paper has shown more general results for general reweighting algorithms and wide NN.

- Reproducibility: proofs for the theorems are provided.


**Strength And Weaknesses:**

### Strengths:
- Distribution shift is common in real-world applications and reweighting is a popular method to deal with this problem. Thus, understanding whether GRW is useful and why is of great importance.
- This work extends the previous theoretical results to the settings: (1) wide neural networks. (2) general reweighted algorithm (3) both regression and classification tasks. The extensions are non-trivial.
- The results are well-supported and instructive, which deepen our understanding of GRW.

### Weaknesses:
- About the strong assumption: as the authors have discussed in Section 6.2, the results of this paper are established on assumptions that the models are linear or wide NN. Based on the assumption, GWR is shown to share similar implicit bias with ERM. I am concerned that these assumptions are too strong to explain the empirical observations for DNN. For example, the proposed GRW also takes the importance weighting algorithm as an example, but with the DNN model, the method is shown to be empirically effective for distribution shift problems by updating the weight dynamically [1]. In this sense, the significance of the proposed theorems seems limited to me.

- About the GRW with L2 regularization: I am not sure whether Theorem 6 can support the claim that GRW does not achieve better DRG than ERM. Then theorem has shown that with a small regularization, the GWR will have a similar testing performance as ERM. But, when the regularization is large, the performance GWR could be different from ERM. Although the training error of GWR (with large regularization) could be worse than ERM, the comparison of their testing performance is still unclear.

- A similar conclusion that DRO will perform just like ERM has also been drawn by the previous work [2]. Their analysis is independent of the model. I think it would be important to provide a discussion on the difference between the two works.

[1] Rethinking Importance Weighting for Deep Learning under Distribution Shift. T Fang, N Lu, G Niu, M Sugiyama.
[2] Does Distributionally Robust Supervised Learning Give Robust Classifiers? W Hu, G Niu, I Sato and M Sugiyama.



**Summary Of The Paper:**

This paper studies the phenomenon that Generalized Reweighting (GRW) and distributionally robust optimization (DRO) do not significantly improve over ERM in real applications with distribution shift. This paper shows for a linear model or an NTK
neural network trained with GRW, the resulting models are close to that obtained by ERM. This paper also shows that L2 regularization only works when it is large enough to lower the training performance. The empirical results verify the arguments in this paper.


**Summary Of The Review:**

This paper has shown that the general reweighting algorithm does not necessarily achieve better robustness than ERM. The analysis of GRW framework is novel to me, and the results are solid. My main concern is that the paper has made strong assumptions about the linear or wide NN model. It is unclear whether the results still hold when extended to the DNN (please see the first point of the weaknesses for more details). Besides, the significance of Theorem 6 is unclear to me (second point of the weakness). I will raise my score if the authors can provide a convincing response to the concerns.

---

> ### Author Response · Authors · 2022-11-10
> **Response to Reviewer NJFN (1/2)**
>
> Thank you for your very detailed comments! We would like to address your concerns as follows:
> ### Regarding GRW being observed to be better than ERM in some real tasks
> This is a good point and we would like to clarify. In some real tasks, GRW can indeed perform better than ERM, but it requires strong regularization and early stopping. However, we would like to argue that studying DRG is still crucial because regularization and early stopping are not the universal solution for all situations, and provide two justifications for this argument:
> 1. [1] empirically showed that GRW methods like Group DRO overfit very easily, and thus require a large regularization (much larger than standard training). This large regularization, however, could impair the model’s average training performance. The results in [1] already showed on some datasets that with a large regularization, GRW achieves a significantly lower average training accuracy than GRW without regularization (e.g. 5% drop on CelebA). Moreover, in our experiments on larger datasets, we find that this gap can be even larger. In real practice, usually people do not want to apply an overly large regularization that significantly lowers the average training performance.
> 2. The domain-oblivious subpopulation shift problem, where the group labels are not provided during training, is still an unsolved problem though there is a large body of work. The critical bottleneck here is model selection: All existing GRW methods, such as DRO [2] and JTT [3], require group labels during validation to select the best model, which should not really be allowed in the domain-oblivious setting. Without a validation set with group labels, no method (even with regularization and early stopping) is guaranteed to perform better than ERM, as shown in Appendix B.2 in [4]. This clearly shows that these GRW methods cannot achieve a better DRG: Should their generalization be better, some simple strategies such as training for a fixed number of epochs would do the job. Thus, the domain-oblivious problem is a concrete example where regularization and early stopping fail.
>
> Given these reasons, we firmly believe that a deeper understanding of DRG is essential for developing better algorithms for tackling subpopulation shift. And in Section 6.1, we list three possible ways to achieve better DRG. We have added the citation the reviewer mentioned.
>
> ---
>
> ### Regarding the assumptions we made
> As we have mentioned in the last paragraph of Section 1 and in Section 6.2, our results do rely on some strong assumptions. Here, we would like to provide three justifications for these assumptions:
> 1. These assumptions (linear or wide NN and without early stopping) are very common and have been widely adopted in existing literature, and the results previously derived from these assumptions (such as the analysis on the loss landscape and training dynamics) have been widely acknowledged to be useful for general deep learning.
> 2. These assumptions are necessary, because at this point, for a general DNN, we cannot even prove basic things like it can converge to the global optima under gradient descent or it can generalize well. In order to prove anything meaningful, we have to make these assumptions.
> 3. The results we prove have significant implications: Our results suggest that a deeper understanding in DRG is crucial for developing better algorithms for distribution shift, which pinpoints a critical bottleneck of current distribution shift research, and aims to open new directions for distributionally robust training algorithms such as the ones we listed in Section 6.1.
>
> ---
>
> ### Regarding the implication of Theorem 6
> Theorem 6 implies that GRW requires a regularization large enough to significantly lower the average training performance in order to achieve a better worst-group test performance. As we mentioned in the first point in the first paragraph, in real practice, usually people do not want to apply an overly large regularization that significantly lowers the average training performance. We agree with the reviewer that in practice, even if the training performance drops significantly, the test performance might not drop as much. However, in general the test performance aligns with the training performance, so a huge drop in the training performance is almost never desirable.

---

> > ### Author Response · Authors · 2022-11-10
> > **Response to Reviewer NJFN (2/2)**
> >
> >
> > ### Regarding the difference from Hu et al. [5]
> > We would like to clarify that our results are **substantially different** from the results in [5]. [5] proved that in classification that uses the zero-one loss, GRW methods such as DRSL are equivalent to ERM, in the sense that the minimizer of the DRSL risk is also the minimizer of the average risk. However, this does not mean that DRSL and ERM will always converge to the *same point*, as there could be multiple minimizers. Their result relies on the zero-one loss, which leads to a monotonic linear relationship between the DRSL risk and the average risk. Moreover, their result is only about the relationship between two minimizers, and they did not prove that DRSL and ERM can actually converge to these global minima.
> >
> > On the other hand, in our results, we first show that without regularization, GRW and ERM will *converge to the exact same point*, so that they have equivalent implicit biases, which is a much stronger result. Then we show that even with regularization, if the regularization is not large enough, GRW will still converge to a point that is very close to the point ERM converges to. Our results do not depend on the loss function, and work for both the squared loss for regression and the logistic loss for classification (and can be extended to other losses). Instead, our results depend on the optimization method (must be first-order or gradient-based) as well as the model architecture (linear or wide NN), since we need to explicitly prove that both GRW and ERM can reach the global minima if trained under a small learning rate for sufficiently long.
> >
> > In a word, [5] proves the equivalence between GRW and ERM under the zero-one loss with a monotonic relationship between the two risk functions, while our results focus on the optimization and training dynamics, and prove that GRW and ERM have almost equivalent implicit biases.
> >
> > We thank the reviewer for bringing this point up, and have added this comparison with [5] in Appendix A in the revision.
> >
> > ---
> >
> > We again thank the reviewer for this very detailed and useful feedback, and sincerely hope that our response can address your concerns. We are looking forward to having further discussions with you during the discussion period.
> >
> > [1] Sagawa et al., Distributionally robust neural networks for group shifts: On the importance of regularization for worst-case generalization, ICLR 2020.
> > [2] Hashimoto et al., Fairness without demographics in repeated loss minimization, ICML 2018.
> > [3] Liu et al.,  Just train twice: Improving group robustness without training group information, ICML 2021.
> > [4] Zhai et al., Doro: Distributional and outlier robust optimization, ICML 2021.
> > [5] Hu et al., Does distributionally robust supervised learning give robust classifiers? ICML 2018.

---

> > > ### Comment · Reviewer_NJFN · 2022-11-16
> > > **discussion**
> > >
> > > Thank you for the feedback and the detailed discussion about the relationship with Hu et al 2016. I do believe that the DRO method does not necessarily achieve better DRG than ERM method without early stopping or strong regularization. But, the proposed GRW is an even more general framework to take many other reweighing algorithms as an example. For instance, it seems to cover the re-weighting algorithms [1] for general distribution shift, where the algorithm is shown to outperform the ERM algorithm without early stopping or strong regularization (but just standard tools like weighted decay is enough). In such a case, I find the claim that "GRW and ERM have (almost) equivalent implicit biases" is too strong to explain some empirical observations outside the DRG research.
> > >
> > > [1] Ren, M., Zeng, W., Yang, B., & Urtasun, R. Learning to reweight examples for robust deep learning. In ICML'2018.

---

> > > > ### Author Response · Authors · 2022-11-16
> > > > **Discussion with Reviewer NJFN**
> > > >
> > > > Thank you for your discussion! We are very happy to address your new concern. First we would like to discuss the specific paper the reviewer talked about, and then we will discuss more about the implication and significance of this work.
> > > >
> > > > ## Regarding [1]
> > > > The reviewer is asking why the method proposed in [1] could achieve good empirical performance despite the fact that we proved that GRW and ERM have (almost) equivalent implicit biases. There are two major reasons:
> > > > ### Reason 1: In [1], the sample weight $w_i$ (denoted by $q_i^{(t)}$ in our work) can be zero.
> > > > One very important assumption we made in this work is that the weight of any sample cannot be zero (e.g. Assumption 1). This is a necessary assumption: If we allow the sample weight to be zero, then one can simply throw away some training samples and train on the remaining samples with ERM, and apparently the result won't be the same as training on all samples with ERM.
> > > >
> > > > On the other hand, the method proposed in [1] has this explicit step: $\tilde{w}_{i,t} = \max ( u_i   , 0)   $, which means that the sample weight can be zero. This is especially important in their noisy label experiment: Their method can identify noisy samples whose weights will go to zero, and our result does not hold in this case.
> > > >
> > > > ### Reason 2: [1] did not train the model for an infinitely long time.
> > > > Our result studies the "implicit biases", meaning that the model needs to be train for an infinitely long time. For example, for a classification task that uses the cross entropy loss, we need to push the cross entropy loss to zero, and consequently the logits will be extremely large. This is not the case in [1]: For example, on MNIST they trained "for a total of 8000 steps".
> > > >
> > > > Now that we are clear about why the method in [1] can achieve good empirical performance despite our results, we can proceed to a more general discussion about our work.
> > > >
> > > > ---
> > > >
> > > > ### 1. We say that "GRW cannot improve over ERM" was observed in a lot of previous empirical work, but we don't say that there is no way to fix it, nor do we say that we should not use GRW at all.
> > > > GRW cannot improve over ERM, or GRW can easily overfit, was observed in [2,3,4,5], so this is a widely observed phenomenon. However, there are a number of ways to fix this issue for specific tasks, such as strong regularization, early stopping [2] and allowing the sample weights to be zero [1]. We never try to argue that we should not use GRW at all, but **we are saying that there is a risk**: A risk that even with fixes such as strong regularization and early stopping, GRW is still not better than ERM. Example: As we mentioned in the first point in the original response: These fixes do not work in the domain-oblivious fairness problem.
> > > > ### 2. We believe that a deeper understanding of DRG is important for better algorithms.
> > > > The reason is that **if we know that an algorithm has better DRG, then we can have much more confidence that the algorithm can work on a wider variety of tasks**. [3,4] showed that lots of distributionally robust methods that were previously shown to have good empirical performance cannot improve over ERM on some large-scale realistic tasks, and one critical reason is that these method do not have better DRG. Going back to [1]: Can we be really confident that the method proposed by [1] can work well on tasks whose scales are much larger than MNIST and CIFAR? We are not so sure. However, if we can somehow improve the DRG of the method in [1], then we can be much more confident since the method is theoretically proved to be good.
> > > > ### 3. The DRG research has just begun and there are some fruitful results.
> > > > As we mentioned in Section 6.1 in the paper, currently there are three lines of research on DRG, where we have already got some fruitful results [6,7,8]. We sincerely hope that this paper can arouse people's (such as the reviewer's) awareness of the importance of DRG, so that more people can join us in developing methods with provable better DRG. Thus, we firmly believe that this paper has high significance.
> > > >
> > > > We hope that this response addresses your new concern, and we are more than happy to have further discussions.
> > > >
> > > > ---
> > > > [1] Ren et al., Learning to reweight examples for robust deep learning, ICML 2018.
> > > > [2] Sagawa et al., . Distributionally robust neural networks for group shifts: On the importance of regularization for worst-case generalization, ICLR 2020.
> > > > [3] Gulrajani et al., In search of lost domain generalization, ICLR 2021.
> > > > [4] Koh et al., Wilds: A benchmark of in-the-wild distribution shifts, ICML 2021.
> > > > [5] Byrd & Lipton, What is the effect of importance weighting in deep learning?, ICML 2019.
> > > > [6] Cao et al., Learning imbalanced datasets with label-distribution-aware margin loss, NeurIPS 2019.
> > > > [7] Kini et al., Label imbalanced and group-sensitive classification under overparameterization, NeurIPS 2021.
> > > > [8] Wang et al., Is importance weighting incompatible with interpolating classifiers?, ICLR 2022.

---

> ### Author Response · Authors · 2022-11-16
> **Examples of How to Improve the DRG of GRW Methods**
>
> It might help the reviewer understand how the DRG of existing GRW methods can be improved with some concrete examples. In these improved methods, **we are still using GRW or some kind of reweighting**, but we make some small modifications and the DRG can be provably better.
>
> ### Example 1: Changing the loss function.
> [1] showed that the DRG of GRW can be simply improved by changing the loss function. Note that in our paper, for classification we proved that under an exponentially-tailed loss such as the logistic loss or the cross entropy loss, GRW and ERM have almost equivalent implicit biases. [1] proved that by using a polynomially-tailed loss instead, the implicit bias of GRW will be different. And for some simple data model, they showed that the DRG of this method can be provably better.
>
> ### Example 2: Applying sample weights to the logits.
> [2] showed that we can improve DRG if we multiple the sample weights to the logits of a classifier instead of multiplying the sample weights to the loss. Previous GRW minimizes $\sum_i q_i \cdot \ell(f(x_i), y_i)$, while [2] proposed to minimize $\sum_i \ell(q_i \cdot f(x_i), y_i)$ and proved that it enjoys better DRG.
>
> Through these two examples, we want to demonstrate that the DRG of GRW can be improved with some very simple techniques. Therefore, **we can still use GRW, but we just need to use them with these DRG improving techniques which can lead to provable better performances**, instead of strong regularization or early stopping that are not guaranteed to work in all scenarios. This is the core argument of this paper, and we firmly believe that this is very important for distributional robustness research.
>
> As always, we are more than happy to have further discussions with the reviewer.
>
> [1] Wang et al., Is importance weighting incompatible with interpolating classifiers?, ICLR 2022.
> [2] Kini et al., Label imbalanced and group-sensitive classification under overparameterization, NeurIPS 2021.

---

### Official Review · Reviewer_cgwJ · 2022-10-27

**Confidence:** 4
**Correctness:** 4
**Technical Novelty And Significance:** 3
**Empirical Novelty And Significance:** Not applicable
**Recommendation:** 8

**Clarity, Quality, Novelty And Reproducibility:**

The paper is well-organized and well presented. The established theoretical results are novel, and the contributions are solid.

**Strength And Weaknesses:**

Strength:

1. The authors conducted a systematic study to theoretical understand why generalized reweighting does not improve over ERM. The results may apply to various existing reweighting schemes.
2. The conclusion of the paper is useful and inspiring. It may lead to more future work on addressing the related robust learning problems.


Weaknesses:
None identified.

**Summary Of The Paper:**

From a theoretical viewpoint, this paper studied Generalized Reweighting (GRW) algorithms that include importance weighting and Distributionally Robust Optimization (DRO) variants, which were designed for learning with a distributional shift. The authors showed that when used to train overparameterized linear models or wide NN models, GRW algorithms do not improve over ERM due to the existence of implicit biases. It is also shown that adding small regularization which does not greatly affect the empirical training accuracy cannot help. Based on the theoretical results, it is concluded that to pursue distributionally robust generalization, one needs to develop non-GRW approaches or devise novel classification/regression loss functions that are adapted to GRW approaches.

**Summary Of The Review:**

See above.

---

> ### Author Response · Authors · 2022-11-10
> **Response to Reviewer cgwJ**
>
> Thank you for your comments! Just like the reviewer said, this paper pinpoints a critical issue in existing methods for distribution shift, that is they cannot improve the DRG over ERM. Our results suggest that a deeper understanding in DRG is crucial for developing better algorithms for distribution shift, and aims to open new directions for distributionally robust training algorithms such as the ones we listed in Section 6.1. If the reviewer would like to discuss more about the paper with us during the discussion period, we will be more than happy to do so.

---

### Author Response · Authors · 2022-11-10
**General Response to Reviewers**

We would like to thank all reviewers for their valuable and detailed comments. We have revised the paper according to the feedback and have uploaded the revision, with edits marked in red. We are looking forward to having further discussions with the reviewers before Nov 18.

Thanks,
Paper964 Authors

---

> ### Author Response · Authors · 2022-11-15
> **Discussion Period Ending Soon**
>
> Just a kindly reminder that the discussion period ends on Nov 18. We are looking forward to discussing with the reviewers and are happy to answer any further questions.

---

> ### Author Response · Authors · 2022-12-08
> **Rebuttal Read Acknowledgement**
>
> Dear Reviewers,
>
> The discussion period is ending on Dec 12, but 3 of the 4 reviewers have yet to respond to our rebuttal. Could you please acknowledge that you have read the rebuttal? And we are happy to answer any further questions.
>
> Thanks,
> Paper964 Authors

---

### Decision · Program_Chairs · 2023-01-20

**Decision:**

Accept: poster

**Justification For Why Not Higher Score:**

Reading the paper, it really boils down to the fact that the implicit bias of GD means that the solution is the minimal norm solution in the span of the examples, a fact that is not changed by the reweighting of the examples. As such, I find the results hardly surprising.

**Justification For Why Not Lower Score:**

The reviewers liked it and there was very little engagement after the authors' rebuttal. I could go against them (and I'm still tempted to do so) but, given the complete lack of discussion between the reviewers and the authors, this might be tricky. Please advise.

**Metareview: Summary, Strengths And Weaknesses:**

This paper shows that reweighting examples, even dynamically during optimization, does not change the final model under some linearity and interpolation assumption. I find this hardly surprising for two reasons:
- the results on the implicit bias of GD show that the final model is the minimal norm solution in the span of the examples (or of the features in the case of the linearized assumption of a very large net). The norm and the span are independent of the weight of each example and thus  the result emerges.
- this result echoes the vast literature in tabular RL and bandits where the same solution is achieved regardless of the sampling of states and actions, a property that disappears as soon as the capacity is reduced.

That being said, the reviewers appreciated the result being laid out and it might provide a reference to those aiming to understand when reweighting does and does not work.

**Note From Pc:**

if the above contains the word "oral" or "spotlight" please see: "oral" presentation means -> notable-top-5% and "spotlight" means -> notable-top-25%. As stated in our emails, we are disassociating presentation type from AC recommendations

**Summary Of Ac-Reviewer Meeting:**

The reviewer did not reply to any of my messages. This was not a good basis to have a call.